# Does the Data Processing Inequality Reflect Practice? On the Utility of Low-Level Tasks

**Roy Turgeman**
Faculty of Engineering
Bar-Ilan University
Ramat Gan, Israel

**Tom Tirer**
Faculty of Engineering
Bar-Ilan University
Ramat Gan, Israel

## Abstract

The data processing inequality is an information-theoretic principle stating that the information content of a signal cannot be increased by processing the observations. In particular, it suggests that there is no benefit in enhancing the signal or encoding it before addressing a classification problem. This assertion can be proven to be true for the case of the optimal Bayes classifier. However, in practice, it is common to perform "low-level" tasks before "high-level" downstream tasks despite the overwhelming capabilities of modern deep neural networks. In this paper, we aim to understand when and why low-level processing can be beneficial for classification. We present a comprehensive theoretical study of a binary classification setup, where we consider a classifier that is tightly connected to the optimal Bayes classifier and converges to it as the number of training samples increases. We prove that for any finite number of training samples, there exists a pre-classification processing that improves the classification accuracy. We also explore the effect of class separation, training set size, and class balance on the relative gain from this procedure. We support our theory with an empirical investigation of the theoretical setup. Finally, we conduct an empirical study where we investigate the effect of denoising and encoding on the performance of practical deep classifiers on benchmark datasets. Specifically, we vary the size and class distribution of the training set, and the noise level, and demonstrate trends that are consistent with our theoretical results.

## 1 Introduction

Deep neural networks (DNNs) have demonstrated remarkable performance across an extensive range of tasks, spanning from image and speech recognition to natural language processing and scientific discovery. When the end goal is to address "high-level" tasks, e.g., classification and detection, a natural approach is to train a DNN to directly solve the task using the raw data/observations as input (Yim & Sohn, 2017; Hendrycks & Dietterich, 2019; Singh et al., 2019). Yet, it is a common practice to begin with addressing a "low-level" task in order to improve the quality of the input for the high-level task. Such low-level tasks include signal/image restorations (Tirer & Giryes, 2018; 2020; Zhang et al., 2021) as considered in (Liu et al., 2018; Dai et al., 2016; Li et al., 2023; Son et al., 2020; Haris et al., 2021; Pei et al., 2018), or encoding to a learned embedding space (Lee et al., 2022; Zhou & Paffenroth, 2017; Wu et al., 2023).

This common pipeline, however, stands in contrast to the data processing inequality, a foundational concept in information theory (Cover, 1999), which states that the information content of a signal cannot be increased by processing the observations. Concretely, consider the Markov chain of three random variables: $y \rightarrow x \rightarrow z$, which denotes that $z$ is independent of $y$ given $x$, i.e., $p_{z|x,y}(z|x,y) = p_{z|x}(z|x)$ in terms of probability distributions. This implies that $p_{x,y,z}(x,y,z) = p_y(y)p_{x|y}(x|y)p_{z|x}(z|x) = p_x(x)p_{y|x}(y|x)p_{z|x}(z|x)$. The data processing inequality reads as

$$I(x,y) \geq I(z,y) \tag{1}$$

where $I(x, y)$ is the mutual information of the random variables $x$ and $y$.[1] In particular, if $y$ is the class of a data sample $x$, this implies that there is no "benefit" in low-level processing of the sample (e.g., obtaining $z$ by denoising $x$) before directly considering the classification problem.

Focusing on classification and referring to better (top-1) accuracy as "benefit", the previous assertion can be proven to be true for the case of the optimal Bayes classifier (more details in Section 2). Clearly, we expect performance gaps between practical classifiers and the optimal Bayes classifier. However, modern DNN-based classifiers reach outstanding classification performance, sometimes even exceeding human capabilities. This raises the question: What can we say about the margin between this implication of the data processing inequality and practical classifiers? To the best of our knowledge, no prior work has attempted to theoretically and systematically investigate this question.

In this paper, we aim to understand when and why low-level processing can be beneficial for classification, even when the classifier is "strong" (e.g., converges to the optimal Bayes classifier when the number of training samples grows). Our main contributions include:

- We present a theoretical study of a binary classification setup, where we consider a classifier that is tightly connected to the optimal Bayes classifier (and converges to it). In the high-dimensional setting, we prove that for any finite number of training samples, there exist a pre-classification processing (specifically, a dimensionality reduction procedure) that improves the classification accuracy.

- We establish theoretical results on the effect of various factors, such as the number of training samples, the level of class separation and training set imbalance, on the relative gain from the data processing procedure that we construct. For example, we show that, non-intuitively, the maximal relative gain increases when the class separation improves.

- We present an empirical investigation of the theoretical model that corroborates our theory and sheds more light on the gains from low-level processing.

- We complement our theoretical work with an empirical study. We investigate the effect of image denoising and self-supervised encoding on the performance of practical deep classifiers on benchmark datasets, where we vary the size of the training set, the class distribution in the training set, and the noise level in the samples. We demonstrate trends that are consistent with our theoretical results (e.g., the one on the maximal gain), highlighting the usefulness of the theoretical setup.

## 2 BACKGROUND AND RELATED WORK

Consider the classification task, where the data $(x, y)$ is distributed on $\mathcal{X} \times [C]$, with $[C] := \{1, \ldots, C\}$ and distribution denoted by $p_{x,y}$. For the binary $0 - 1$ criterion, i.e., $\ell(\hat{y}, y) = \mathbb{I}(\hat{y} \neq y)$, the expected risk is equivalent to the error probability $\mathbb{E}[\ell(\hat{y}(x), y)] = \mathbb{P}(\hat{y}(x) \neq y)$. It is well-known that this objective is minimized by the (optimal) Bayes classifier: $c_{opt}(x) = \text{argmax}_{y \in [C]} \, p_{y|x}(y|x)$, where $p_{y|x}$ is the true conditional probability of $y$ given $x$ (Bishop, 2006; Fukunaga, 2013). In practice, of course, the distributions are unknown and a classifier must be learned from data samples.

Consider a data processing operation $\mathcal{A} : \mathcal{X} \to \mathcal{Z}$. This can be denoising, super-resolution, encoding, etc. Let $z = \mathcal{A}(x)$. Notice that $y \to x \to z$ is a Markov chain because $z$ is a function of $x$ and thus $p_{z|x,y}(z|x, y) = p_{z|x}(z|x)$. Therefore, the data processing inequality in Eq. 1 holds. The optimal Bayes classifier that operates on a processed sample $z = \mathcal{A}(x)$ is given by $\tilde{c}_{opt}(z) = \text{argmax}_{y \in [C]} \, p_{y|z}(y|z)$, where $p_{y|z}$ is the true conditional probability of $y$ given $z$.

Focusing on the case of binary classification ($C = 2$), the following result shows that, similarly to the fact that no $\mathcal{A}$ can increase information, there is also no hope in improving the accuracy of optimal Bayes classifiers via data processing.

**Theorem 1.** *Let $y \to x \to z$ be a Markov chain where $y \in \{1, 2\}$ denotes the sample class. We have*

$$\mathbb{P}(c_{opt}(x) \neq y) \leq \mathbb{P}(\tilde{c}_{opt}(z) \neq y), \tag{2}$$

*where $c_{opt}$ and $\tilde{c}_{opt}$ denote optimal Bayes classifiers.*

---

[1]The mutual information is defined as $I(x, y) = \iint p_{x,y}(x, y) \log \left( \frac{p_{x,y}(x, y)}{p_x(x) p_y(y)} \right) \mathrm{d}x \mathrm{d}y.$

A similar statement and proof can be found in an arXiv version of (Liu et al., 2019). For completeness, we present a clearer proof in Appendix A. Note that (Liu et al., 2019) studies a potential tradeoff between the error of a low-level restoration task and the accuracy of a *fixed* classifier, where only the restoration model is trained using the training data. In contrast, our work focuses on the high-level end goal—the classification performance—and allows training the classifier after the low-level processing, as is done in practice. Therefore, (Liu et al., 2019) does not provide any reason why in practice it is common to address a low-level task before high-level ones, which is the central question of our paper.

Our work is motivated by the contrast between common practice and the information-theoretic concept of the data processing inequality, as well as Theorem 1. There exist works that use information-theoretic concepts or compute approximate metrics to analyze DNNs, e.g., (Tishby & Zaslavsky, 2015; Shwartz-Ziv & Tishby, 2017; Saxe et al., 2019; Gabrié et al., 2018; Jeon & Van Roy, 2022). Interestingly, since a DNN processes data gradually, layer by layer, the features across the layers form a Markov chain, and thus the data processing inequality applies. Yet, avoiding the loss of information relevant to the task being learned can be attributed to penalizing failures in predicting the target labels during training, while discarding task-irrelevant information (akin to compression) may be explained by the information bottleneck principle (Tishby & Zaslavsky, 2015; Shwartz-Ziv & Tishby, 2017; Saxe et al., 2019). The contrast between representation learning and the data processing inequality has also motivated theoretical works (Xu et al., 2020; Goldfeld & Greenewald, 2021) to study variants of the mutual information, incorporating transformations of the signal or line projections. None of the aforementioned works consider a sequence of tasks or explain when and why low-level processing can be beneficial to practical classifiers. Moreover, here we directly analyze the classifier's probability of error, which is more interpretable than the information-theoretic objectives studied before.

Finally, we emphasize that in the case of pre-trained classifier under distribution shift, data processing that 'reduces the gap' between the test data distribution and the training data distribution is trivially expected to improve the classifier performance. However, we focus in this paper on the non-intuitive case where no distribution shift occurs, and the classifier is strong, in the sense that it converges to the optimal Bayes classifier as the training set increases (with good statistical properties).

## 3 THEORY

In this section, we present our theoretical contributions. First, we describe the problem setup, the data distribution, the classifier under study, and a data processing operation. Next, we present our theoretical results demonstrating the benefits of this data processing. Finally, we validate our results through experiments and provide additional insights into the factors that affect the performance gain.

### 3.1 PROBLEM SETUP: DATA MODEL, CLASSIFIER, AND DATA PROCESSING

**Data model.** Similar to a vast body of theoretical work on classifiers (Cao et al., 2021; Deng et al., 2022; Wang & Thrampoulidis, 2022; Kothapalli & Tirer, 2025), we consider binary classification ($C = 2$), where the data is distributed according to a Gaussian Mixture Model (GMM) of order two in $\mathcal{X} = \mathbb{R}^d$, with one mixture component per class. Formally,

$$y \in \{1, 2\}, \qquad \boldsymbol{x} \mid y = j \sim \mathcal{N}(\boldsymbol{\mu}_j, \sigma_j^2 \boldsymbol{I}_d), \qquad \mathbb{P}(y = j) = \pi_j. \tag{3}$$

Similar to previous theoretical works, we further assume that

$$\boldsymbol{\mu}_2 = -\boldsymbol{\mu}_1 = \boldsymbol{\mu}, \qquad \sigma_1^2 = \sigma_2^2 = \sigma^2, \qquad \pi_1 = \pi_2 = 1/2, \tag{4}$$

where the magnitudes of the entries of $\boldsymbol{\mu}$ are bounded by some universal constant, and $\sigma$ is independent of $d$. Let us now define the separation quality factor of the GMM data, which can be understood as the signal-to-noise ratio (SNR):

$$\mathcal{S} := \left( \frac{\|\boldsymbol{\mu}_2 - \boldsymbol{\mu}_1\|}{\sigma_1 + \sigma_2} \right)^2 = \frac{\|\boldsymbol{\mu}\|^2}{\sigma^2}. \tag{5}$$

Note that the considered setup is standard in theoretical works that aim at rigorous mathematical analysis (Cao et al., 2021; Deng et al., 2022; Wang & Thrampoulidis, 2022; Kothapalli & Tirer, 2025). Despite its compactness, the learning problem studied in this paper can be arbitrarily hard because (unlike some of the aforementioned works) our analysis covers SNR arbitrarily close to zero, i.e., nearly indistinguishable classes.

The training data consists of $N_j$ labeled i.i.d. samples per class $j$, denoted by $\mathcal{D} = \{\boldsymbol{x}_{i,j} : j \in \{1,2\}, i = 1, \ldots, N_j\}$. Without loss of generality, we denote $N_1 = N$ and $N_2 = \gamma N$ for some $\gamma \in (0,1]$.

**The classifier.** In the considered setting, the optimal Bayes classifier reads:

$$c_{opt}(\boldsymbol{x}) = \arg\max_{j \in \{1,2\}} \pi_j p_{x|y}(\boldsymbol{x}|j) = \arg\max_{j \in \{1,2\}} \exp\left(-\frac{\|\boldsymbol{x}-\boldsymbol{\mu}_j\|^2}{2\sigma^2}\right) = \arg\min_{j \in \{1,2\}} \|\boldsymbol{x} - \boldsymbol{\mu}_j\|.$$

In practice, the data distribution is unknown and thus a classifier cannot use the class means, $\{\boldsymbol{\mu}_i\}$, but rather estimate them from the training set. We therefore study the classifier:

$$\widehat{c}(\boldsymbol{x}; \mathcal{D}) = \arg\min_{j \in \{1,2\}} \|\boldsymbol{x} - \widehat{\boldsymbol{\mu}}_j\|, \tag{6}$$

where $\widehat{\boldsymbol{\mu}}_j = \frac{1}{N_j} \sum_{i=1}^{N_j} \boldsymbol{x}_{i,j}$ is the maximum likelihood estimate of $\boldsymbol{\mu}_j$ from the $j$-th class's samples.

We want to explore if data processing can be beneficial even for a "strong" classifier. It is easy to see that $\widehat{\boldsymbol{\mu}}_j \sim \mathcal{N}\left(\boldsymbol{\mu}_j, \frac{\sigma^2}{N_j} \boldsymbol{I}_d\right)$. In fact, this is an efficient estimator that attains the Cramér–Rao lower bound on the variance for *any* $N_j$ (Kay, 1993). Therefore, in our setting, not only that $\widehat{c}(\cdot)$ is structurally similar to $c_{opt}(\cdot)$ and converges to it for $N_j \to \infty$, but it also has strong statistical properties for finite $N_j$, making it a natural choice for our study. Demonstrating the benefit of low-level processing for such a classifier, which is "almost optimal" for the considered setup, underscores the potential advantages for weaker classifiers.

**Data processing.** As the pre-classification data processing, we are going to study a certain linear dimensionality reduction to $1 \le k < d$. Specifically, we consider

$$\boldsymbol{z} = \boldsymbol{A}\boldsymbol{x}$$

with $\boldsymbol{A} \in \mathbb{R}^{k \times d}$ that obeys

$$\boldsymbol{A}\boldsymbol{A}^\top = \boldsymbol{I}_k, \qquad \|\boldsymbol{A}\boldsymbol{\mu}\| = \|\boldsymbol{\mu}\|. \tag{7}$$

Note that, for establishing our main theoretical claim on the practical limitation of Eq. 2, we just need the *existence* of a processing for which we can rigorously show improved classification. Nevertheless, in the sequel, we provide a constructive proof that also shows how such $\boldsymbol{A}$ *can be learned from unlabeled data* without prior knowledge of $\boldsymbol{\mu}$. Hence, showing that this procedure improves classification performance in our setup underscores the promise of practical low-level procedures learned from unlabeled data.

**Additional notations.** We will analyze and compare the performance of the classifier in Eq. 6 before and after the data processing procedure, namely, $\widehat{c}(\boldsymbol{x}; \mathcal{D})$ versus $\widehat{c}(\boldsymbol{z}; \mathcal{D}_{\boldsymbol{z}})$, where $\mathcal{D}_{\boldsymbol{z}} = \{\boldsymbol{z}_{i,j} = \boldsymbol{A}\boldsymbol{x}_{i,j} : j \in \{1,2\}, i = 1, \ldots, N_j\}$. We denote the probability of error in these two cases by $p_{\boldsymbol{x}}(\text{error}) := \mathbb{P}(\widehat{c}(\boldsymbol{x}; \mathcal{D}) \neq y)$ and $p_{\boldsymbol{z}}(\text{error}) := \mathbb{P}(\widehat{c}(\boldsymbol{z}; \mathcal{D}_{\boldsymbol{z}}) \neq y)$. Finally, we define the widely-used $\mathcal{Q}$-function, which will be used to characterize the classification error probability:

$$\mathcal{Q}(x) = \mathbb{P}\left(\mathcal{N}(0,1) > x\right) = \frac{1}{\sqrt{2\pi}} \int_x^\infty \exp\left(-\frac{t^2}{2}\right) dt. \tag{8}$$

## 3.2 THEORETICAL RESULTS

In this subsection, we present our theoretical results. In Section 3.2.1, we prove that the error probability decreases due to the data processing. To this end, we establish expressions that accurately approximate the probability of error of the data-driven classifier before and after the processing. We then analyze their relation, where, due to different proof strategies, this is done separately for the balanced and imbalanced training set cases. In Section 3.2.2, we provide a fine-grained analysis of the factors that affect the efficiency of the processing, and, for the balanced training set case, we also establish a connection between the maximal gain and the SNR.

The proofs for all the claims are deferred to Appendix A.

### 3.2.1 PERFORMANCE GAIN DUE TO DATA PROCESSING

We begin with characterizing the probability of error when the classifier is applied without pre-processing. Recall the definitions of $\mathcal{S}$ and $\mathcal{Q}(x)$ in Eq. 5 and Eq. 8, respectively.

**Theorem 2** (The probability of error before the processing). *Consider the setup in Section 3.1. With approximation accuracy $\mathcal{O}(1/\sqrt{d})$ we have $p_{\boldsymbol{x}}(\text{error}) \approx \hat{p}_{\boldsymbol{x}}(\text{error}) = \hat{p}(\mathcal{S}, N, \gamma, d)$, where*

$$
\begin{aligned}
\hat{p}(\mathcal{S}, N, \gamma, d) := \frac{1}{2} \cdot \mathcal{Q}\left( \frac{\sqrt{\mathcal{S}} + \frac{1}{4N} \cdot \frac{1-\gamma}{\gamma} \cdot \frac{d}{\sqrt{\mathcal{S}}}}{\sqrt{\frac{1}{4N} \cdot \frac{1+\gamma}{\gamma} \cdot \frac{d}{\mathcal{S}} + \frac{1}{8N^2} \cdot \frac{1+\gamma^2}{\gamma^2} \cdot \frac{d}{\mathcal{S}} + \frac{1}{\gamma N} + 1}} \right) \\
+ \frac{1}{2} \cdot \mathcal{Q}\left( \frac{\sqrt{\mathcal{S}} - \frac{1}{4N} \cdot \frac{1-\gamma}{\gamma} \cdot \frac{d}{\sqrt{\mathcal{S}}}}{\sqrt{\frac{1}{4N} \cdot \frac{1+\gamma}{\gamma} \cdot \frac{d}{\mathcal{S}} + \frac{1}{8N^2} \cdot \frac{1+\gamma^2}{\gamma^2} \cdot \frac{d}{\mathcal{S}} + \frac{1}{N} + 1}} \right).
\end{aligned}
\tag{9}
$$

**Remark.** The proof is mathematically involved. We express the error event as thresholding a scalar random variable, suitable for an application of a generalized Berry–Esseen theorem. However, this variable depends on the interrelation between the entries of $\widehat{\boldsymbol{\mu}}_1$, $\widehat{\boldsymbol{\mu}}_2$, and computing the required moments is a technical challenge.

**Discussion.** Note that: 1) $\hat{p}$ is symmetric in the following sense: $\hat{p}(\mathcal{S}, N, \gamma, d) = \hat{p}\left(\mathcal{S}, \gamma N, \frac{1}{\gamma}, d\right)$, which is expected because swapping the amount of samples between the classes does not change the problem; 2) As $\mathcal{S} \to 0^+$ we have: $\lim_{\mathcal{S} \to 0^+} \hat{p}(\mathcal{S}, N, \gamma, d) = 1/2$, aligned with uniform guess; 3) As $\mathcal{S} \to \infty$ we have: $\lim_{\mathcal{S} \to \infty} \hat{p}(\mathcal{S}, N, \gamma, d) = 0$, aligned with the classes being deterministically separable; and 4) As $N \to \infty$ we have: $\lim_{N \to \infty} \hat{p}(\mathcal{S}, N, \gamma, d) = \mathcal{Q}(\sqrt{\mathcal{S}})$, which is the probability of error of $c_{opt}$, which knows the exact distribution of the data (Fukunaga, 2013).

Let us explore the result of Theorem 2 for the case of balanced training data, $\gamma = 1$ ($N_2 = N_1 = N$), in which the expression simplifies to:

$$
\hat{p}_{\boldsymbol{x}}(\text{error}) = \mathcal{Q}\left( \frac{\sqrt{\mathcal{S}}}{\sqrt{\left(\frac{d}{2\mathcal{S}} + 1\right) \cdot \frac{1}{N} + \frac{d}{4\mathcal{S}} \cdot \frac{1}{N^2} + 1}} \right).
\tag{10}
$$

Fix $d \gg 1$, which ensures that the approximation is accurate. It is easy to see that when the separation quality factor (SNR), $\mathcal{S}$, decreases (with fixed $d$), the argument of the $\mathcal{Q}$-function decreases, and thus the probability of error increases. In addition, as the number of training samples $N$ increases, the argument increases, and thus the probability of error decreases. These two results are aligned with intuition. Interestingly, the effect of increasing/decreasing $d$ depends on its relation with $\mathcal{S}$. For example, if $d$ increases and $\mathcal{S}$ is fixed, which means that the average entry-wise SNR decreases, then the argument of the $\mathcal{Q}$-function increases. The contrary holds if $\mathcal{S} \propto d$, which means that the average entry-wise SNR is fixed. In the latter case, high-dimensionality is advantageous in terms of the probability of error.

Next, let us establish the existence and learnability of the data processing proposed in Section 3.1.

**Theorem 3** (The existence and learnability of the processing). *For all $1 \leq k < d$, there exists a dimension-reducing matrix $\boldsymbol{A} \in \mathbb{R}^{k \times d}$ with the properties stated in Eq. 7. Furthermore, given sufficiently many unlabeled samples, such a matrix can be learned to arbitrary accuracy.*

**Remark.** The proof of Theorem 3 is constructive. It provides an algorithm for computing such $\boldsymbol{A}$ and efficiently estimating the direction of $\boldsymbol{\mu}$ from *unlabeled* data.

Note that the semi-orthonormality of $\boldsymbol{A}$ implies that it cannot increase the norm of any vector, while the property $\|\boldsymbol{A}\boldsymbol{\mu}\| = \|\boldsymbol{\mu}\|$ ensures that the separation quality remains unchanged (equal to Eq. 5) and is not reduced after the processing. In more detail, this implies that when applying $\boldsymbol{A}\boldsymbol{x}$, the class-dependent component of $\boldsymbol{x}$ (i.e., the projection of $\boldsymbol{x}$ onto $\pm\boldsymbol{\mu}$) is not attenuated. In contrast, the complementary component of $\boldsymbol{x}$, which corresponds to within-class variability, is attenuated as the overall dimension is reduced and the semi-orthonormality of $\boldsymbol{A}$ prevents amplification. Taken

together, this is expected to facilitate classification, as will be rigorously proven below. More details and a graphical illustration of the action of $\boldsymbol{A}$ are presented in Appendix F.

We now turn to characterizing the probability of error when applying the classifier on the processed data $\boldsymbol{z} = \boldsymbol{A}\boldsymbol{x}$.

**Theorem 4** (The probability of error on the processed data). *Consider the setup in Section 3.1. With approximation accuracy $\mathcal{O}(1/\sqrt{k})$ we have $p_{\boldsymbol{z}}(\text{error}) \approx \hat{p}_{\boldsymbol{z}}(\text{error}) = \hat{p}\left(\mathcal{S}, N, \gamma, k\right)$, where $\hat{p}$ is defined in Eq. 9.*

The approximate probability of error of the processed data, $\hat{p}_{\boldsymbol{x}}(\text{error})$, admits an expression similar to the one obtained for the raw data, $\hat{p}_{\boldsymbol{z}}(\text{error})$, but with a different dimension parameter ($k$ instead of $d$). Note that in the high-dimensional case, i.e., $d, k \gg 1$, these estimators are guaranteed to be accurate. Now, let us present the main outcomes of our theoretical study, which build on these expressions.

We start with the case where there is no class imbalance in the training set, i.e., $\gamma = 1$ so $N_2 = N_1 = N$. The next theorem shows that the considered data processing yields a gain for any finite value $N \geq 1$. We assume $\mathcal{S} > 0$, as $\mathcal{S} = 0$ is an uninteresting degenerate case.

**Theorem 5** (Performance gain under balanced training data). *For $\gamma = 1$, and for all $\mathcal{S} > 0$, $1 \leq k < d$, and $N \in \mathbb{N}$, we have*

$$\hat{p}_{\boldsymbol{x}}(\text{error}) > \hat{p}_{\boldsymbol{z}}(\text{error}). \tag{11}$$

Theorem 5 shows that when the training samples are balanced among the classes, the chosen processing always *strictly decreases* the approximated probability of error.

**Discussion.** As shown in Theorems 2 and 4, in the high-dimensional case the true probabilities of error, $p_{\boldsymbol{x}}(\text{error})$ and $p_{\boldsymbol{z}}(\text{error})$, are well approximated by $\hat{p}_{\boldsymbol{x}}(\text{error})$ and $\hat{p}_{\boldsymbol{z}}(\text{error})$. This makes the result significant. Moreover, this result—holding for *any* finite $N$—is also quite surprising, since in the limit of $N \to \infty$ we have that $p_{\boldsymbol{x}}(\text{error})$ and $p_{\boldsymbol{z}}(\text{error})$ converge to $\mathbb{P}(c_{opt}(\boldsymbol{x}) \neq y)$ and $\mathbb{P}(\tilde{c}_{opt}(\boldsymbol{z}) \neq y)$, respectively, which satisfy the opposite relation ($\leq$) as shown in Theorem 1.

We now consider the case of an imbalanced training set. The presence of under-represented classes or groups is of significant interest in the machine learning community, as it raises concerns about generalization and fairness (Chawla et al., 2002; Huang et al., 2016; Li et al., 2021). Specifically, while the classes have equal probability ($\pi_1 = \pi_2 = 0.5$), the number of training samples from each of the classes is assumed to be $N_1 = N$ and $N_2 = \gamma N$ with $0 < \gamma < 1$. The following theorem demonstrates the benefit of the considered data processing in this case as well.

**Theorem 6** (Performance gain under imbalanced training data). *Let $0 < \gamma < 1$, $0 < \mathcal{S} \leq 1$, $1 \leq k < d$. If $N \geq \frac{\gamma^2 - 4\gamma + 1}{2\gamma(1+\gamma)}$, then we have*

$$\hat{p}_{\boldsymbol{x}}(\text{error}) > \hat{p}_{\boldsymbol{z}}(\text{error}). \tag{12}$$

**Remark.** Unlike Theorem 5, which considers $\gamma = 1$ and is smoothly obtained from Theorems 2 and 4, in this case the complexity of the formulas of $\hat{p}_{\boldsymbol{x}}(\text{error})$ and $\hat{p}_{\boldsymbol{z}}(\text{error})$ required us to make technical assumptions on $\mathcal{S}$ and $N$ in order to establish a rigorous statement for $\gamma \in (0, 1)$. Nevertheless, these assumptions are reasonable and still encompass the interesting case of low SNR and a reasonable number of training samples. Note that for $\gamma \geq 0.162$, the requirement $N \geq \frac{\gamma^2 - 4\gamma + 1}{2\gamma(1+\gamma)}$ is vacuous (since $N \geq 1$), so it only matters under severe imbalance ($\gamma < 0.162$).

### 3.2.2 FACTORS THAT AFFECT THE PERFORMANCE GAIN

So far, we have only considered the relation between $\hat{p}_{\boldsymbol{x}}(\text{error})$ and $\hat{p}_{\boldsymbol{z}}(\text{error})$. Let us now discuss the margin between them, which reflects the efficiency of the processing.

**Definition 1.** *We define the theoretical efficiency of the processing as*

$$\eta := \left( \frac{\hat{p}_{\boldsymbol{x}}(\text{error}) - \hat{p}_{\boldsymbol{z}}(\text{error})}{\hat{p}_{\boldsymbol{x}}(\text{error})} \right) \cdot 100. \tag{13}$$

The following theorem establishes an approximation of $\eta$ for $N \gg 1$, making it easier to gain insights into the different factors that affect the efficiency of the processing in the case of a large number of training samples.

**Theorem 7** (Analysis of the asymptotic efficiency). *Let $\mathcal{S} > 0$, $1 \leq k < d$, $0 < \gamma \leq 1$. Denote by $N_T = (1 + \gamma) N$ the total number of training samples. With approximation accuracy $\mathcal{O}(1/N_T^2)$, we have*

$$\eta \approx \frac{25}{2\sqrt{2\pi}} \cdot \frac{\exp\left(-\frac{\mathcal{S}}{2}\right)}{\sqrt{\mathcal{S}} \cdot \mathcal{Q}\left(\sqrt{\mathcal{S}}\right)} \cdot \left(3 + 2\gamma + \frac{1}{\gamma}\right) \cdot (d - k) \cdot \frac{1}{N_T}. \tag{14}$$

*In particular, for $N_T \gg 1$: The efficiency increases when $d - k$ increases or $\gamma$ decreases within $0 < \gamma \leq 1/\sqrt{2}$; The efficiency decreases when $\mathcal{S}$ increases or $N_T$ increases.*

**Remark.** The proof of Theorem 7 is based on first-order analysis, which differs from the proof technique used for Theorem 6. This allows us to reach the conclusion that there exists $N_0 \in \mathbb{N}$ such that for all $N \geq N_0$ we have $\eta > 0$ (since the right-hand side of Eq. 14 is positive) which implies $\hat{p}_{\boldsymbol{x}}(\text{error}) > \hat{p}_{\boldsymbol{z}}(\text{error})$ without a technical assumption on $\mathcal{S}$. On the other hand, Theorem 6 can hold even for small values of $N$, depending on $\gamma$.

**Discussion.** Let us discuss the intuition behind the insights provided in Theorem 7. First, notice that in the considered regime of $N_T \gg 1$ training samples, the processing efficiency $\eta$ monotonically decreases toward zero as $N_T$ increases. This is consistent with the fact that in the limit $N_T \to \infty$ the classifier approaches the optimal Bayes classifier, which cannot be improved by data processing. In this regime, higher class separation $\mathcal{S}$ can be interpreted as equivalent to having more effective samples (akin to larger $N_T$), and hence less improvement through the pre-classification processing. Similarly, larger dimensionality reduction $(d - k)$ can be viewed as greater coverage of the input domain, again, analogous to having more samples. Lastly, lower $\gamma < 1$ indicates that the classifier's training samples are less balanced between the classes and hence differ more from the data distribution. Intuitively, this leaves more room for improvement through pre-classification processing.

In addition, Appendix B provides an approximation of the difference $\Delta := \hat{p}_{\boldsymbol{x}}(\text{error}) - \hat{p}_{\boldsymbol{z}}(\text{error})$ for $N \gg 1$. The insights we obtain are consistent with those reported in Theorem 7.

So far, our theory shows that the processing efficiency $\eta$ is positive for all $N$ for $\gamma = 1$, and under a technical assumption it is positive also for $\gamma \in (0, 1)$. Our formulas also show that $\eta = 0$ at $N = 0$ (where $\hat{p}_{\boldsymbol{z}} = \hat{p}_{\boldsymbol{x}} = 0.5$, i.e., probability of guessing) and that $\eta \to 0$ at $N \to \infty$ (where $\hat{p}_{\boldsymbol{z}} = \hat{p}_{\boldsymbol{x}} = \mathcal{Q}(\sqrt{\mathcal{S}})$, consistent with the classifier converging to the optimal Bayes classifier for which Theorem 1 applies). Together, these imply that there is a maximum point of $\eta(N)$. Our final theorem provides a surprising insight into this maximum efficiency.

**Theorem 8** (Analysis of the maximal efficiency). *Fix $\gamma = 1$, and let $\mathcal{S} > 0, 1 \leq k < d$. Consider the efficiency $\eta = \eta(N)$ as a function of continuous $N \in \mathbb{R}_+$. We have that the maximal efficiency $\eta_{\max} = \max_{N \geq 0} \eta(N)$ increases as a function of $\mathcal{S}$.*

**Discussion.** In the asymptotic regime of $N \to \infty$, as discussed above, a higher SNR corresponds to lower $\eta$, which aligns with intuition. Interestingly, however, the theorem shows that a higher SNR also leads to a larger $\eta_{\max}$, which is somewhat counterintuitive. One might expect that lower noise would reduce efficiency across all sample sizes, since the raw data is already well-separated. This highlights the subtle relationship between $\eta$ and the SNR. An extended version of the theorem can be found in Appendix A.8.

### 3.3 EMPIRICAL VERIFICATION

In this subsection, we simulate the theoretical setup in order to further support our theoretical results and also gain more insights on the model, e.g., factors that affect the efficiency of the data processing for small to moderate values of $N$.

We consider data dimension $d = 2000$, and fix $\sigma = 1$. The SNR values we work with are $\mathcal{S} \in \{0.75^2, 1.5^2\}$, and for each fixed SNR, we use $\gamma \in \{0.25, 0.5, 1\}$. We also consider a wide range of $N_{\text{train}}$, which denotes the total number of given training samples. For each fixed tuple $(\mathcal{S}, \gamma, N_{\text{train}})$,

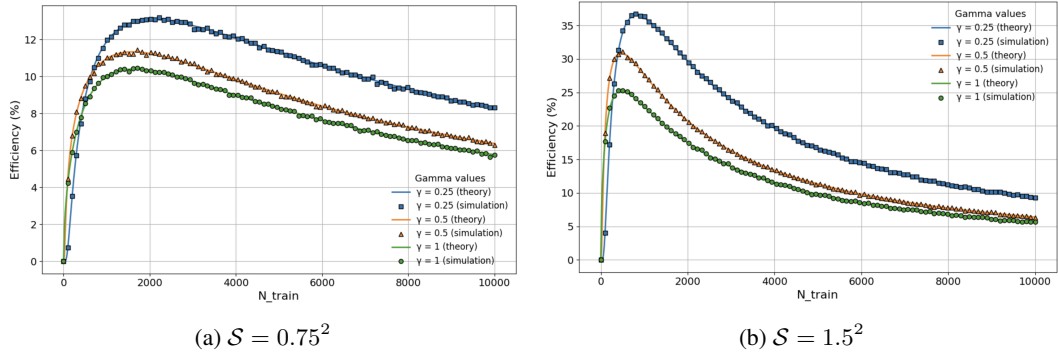

(a) $\mathcal{S} = 0.75^2$  (b) $\mathcal{S} = 1.5^2$

Figure 1: The theoretical setup. Efficiency of the data processing procedure versus the number of training samples $N_{\text{train}}$, for various values of the training imbalance factor, $\gamma$, and the SNR, $\mathcal{S}$.

we randomize $\boldsymbol{\mu} \in \mathbb{R}^d$ with $\|\boldsymbol{\mu}\| = \sigma\sqrt{\mathcal{S}}$, via $\boldsymbol{\mu} = \sigma\sqrt{\mathcal{S}}\frac{\boldsymbol{v}}{\|\boldsymbol{v}\|}$ where $\boldsymbol{v} \sim \mathcal{N}(\boldsymbol{0}, \boldsymbol{I}_d)$. We then construct the data processing matrix $\boldsymbol{A} \in \mathbb{R}^{k \times d}$ that reduces the dimension to $k = 1000$, using the algorithm described in Appendix A.3. Per trial, we sample $N_1 = \text{int}(\frac{N_{\text{train}}}{1+\gamma})$ training points from $\mathcal{N}(-\boldsymbol{\mu}, \sigma^2 \boldsymbol{I}_d)$, and $N_2 = \text{int}(\frac{\gamma N_{\text{train}}}{1+\gamma})$ training points from $\mathcal{N}(\boldsymbol{\mu}, \sigma^2 \boldsymbol{I}_d)$. Before and after the data processing, the per-class means are estimated using the training points, and the classifier defined in Eq. 6 is used on a large amount of fresh test data, sampled with probability $0.5$ from each of the two Gaussians. With a slight abuse of notation, we denote the empirical probabilities of error before and after the data processing by $p_x(\text{error})$ and $p_z(\text{error})$, respectively. In order to compute $p_z(\text{error})$, we use the training and test samples after processing them by multiplying with $\boldsymbol{A}$. We then compute the empirical efficiency of the processing, defined by $\chi = \left(\frac{p_{\boldsymbol{x}}(\text{error}) - p_{\boldsymbol{z}}(\text{error})}{p_{\boldsymbol{x}}(\text{error})}\right) \cdot 100$. We repeat the computation of $\chi$ over 100 independent trials and report the average.

Figure 1 presents both the theoretical efficiency $\eta$, defined in Eq. 13, and the empirical efficiency $\chi$ versus the number of training samples $N_{\text{train}}$, for various values of $\gamma$ and $\mathcal{S}$. Note that the empirical and theoretical efficiencies closely match in all the configurations.

Let us discuss the trends that are observed in Figure 1. First, note the non-monotonic curves depicting the efficiency as a function of $N_{\text{train}}$. When $N_{\text{train}}$ approaches zero or grows to infinity the efficiency tends to zero, aligned with our analytical formulas. Indeed, as discussed above, in the absence of training data the classification is based on guess, and thus there is no effect for the data processing. In the considered setup, as $N_{\text{train}} \to \infty$, the classifiers tend to the optimal Bayes decision rules, which again implies zero efficiency. A major contribution of our paper is providing rigorous theory for the fact that the efficiency remains positive between these two extreme cases.

Let us now focus on $N_{\text{train}} \gg 1$ (the right boundary of each sub-figure). We see that increased $\mathcal{S}$ moderately reduces the efficiency. For example, for $(\mathcal{S}, \gamma, N_{\text{train}}) = (0.75^2, 1, 10K)$ the efficiency is around 6, while for $(\mathcal{S}, \gamma, N_{\text{train}}) = (1.5^2, 1, 10K)$ it is around 5. Moreover, we see that lower values of $\gamma$, corresponding to more imbalanced training data, yield higher efficiency of the data processing. Note that both are aligned with the insights gained in Theorem 7.

Next, note that each of the curves depicts a single maximum point, whose value is aligned with the non-intuitive prediction of Theorem 8. Specifically, the maximal efficiency value increases with $\mathcal{S}$.

Lastly, note that the empirical investigation of our theoretical setup reveals behaviors at relatively small values of $N_{\text{train}}$, which lie beyond the scope of our theoretical analysis. Specifically, we observe that the relation between decrease in $\gamma$ and increase in efficiency emerges already at quite low $N_{\text{train}}$. We also observe dependency between the overall shape of the curves and the value of $\mathcal{S}$.

Additional verification experiments with $\boldsymbol{A}$ that is learned from unlabeled samples, and different values of $\mathcal{S}, k$ are presented in Appendix F. All of them are aligned with our theoretical insights.

## 4 EXPERIMENTS IN PRACTICAL SETTINGS

While our paper focuses on theoretical contributions, in this section, we empirically examine the correlation between the behaviors observed in four practical deep learning settings and the theoretical

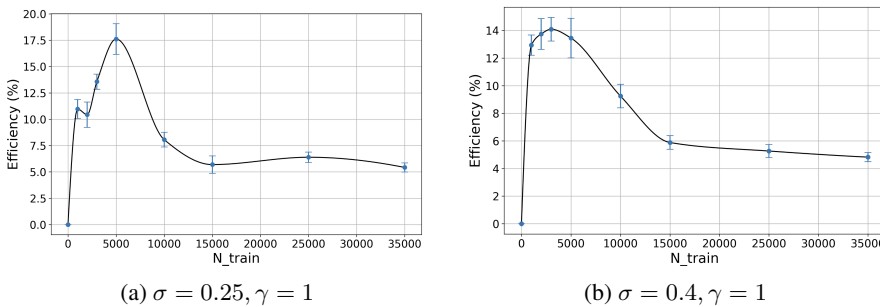

(a) $\sigma = 0.25, \gamma = 1$                                       (b) $\sigma = 0.4, \gamma = 1$

Figure 2: Noisy CIFAR-10 and pre-classification denoising. Efficiency versus $N_{\text{train}}$.

results. Note that such a study, which examines the effects of sample size, SNR, and class balance, requires exhaustive training efforts of both the data-processing module and the classifier.

## 4.1 NOISY CIFAR-10 AND PRE-CLASSIFICATION DENOISING

We consider the CIFAR-10 dataset (Krizhevsky et al., 2009) and the ResNet18 model (He et al., 2016). The train and test sets both experience additive Gaussian noise of the same level (i.e., no distribution shift) with standard deviation $\sigma \in \{0.25, 0.4\}$. A detailed description of the training procedure of the classifier is given in Appendix C. We also note that we verify that the classifier performs well when trained on clean CIFAR-10 data, achieving 90% accuracy.

The data processing step examined here is image denoising, applied to the noisy data, using the DnCNN model (Zhang et al., 2017). The denoiser is trained with the MSE loss on 15,000 clean unlabeled images, which are not part of the classifier's training set. More details on the training procedure of the denoiser are given in Appendix C. Note that, given such a pretrained denoiser, the Markov chain: "label"—"noisy image"—"denoised image" still holds. Thus, the data processing inequality, as well as Theorem 1, suggest that the denoiser will not improve the results.

In Appendix C, we also investigate another setting, where we train the denoiser with SURE loss (i.e., without clean ground truth images) (Stein, 1981; Soltanayev & Chun, 2018), and observe similar results.

We consider various values of $N_{\text{train}}$, the total number of given training samples (across all 10 classes), and examine different training imbalance factors, $\gamma = 1$ here, and $\gamma \in \{0.5, 0.75\}$ in Appendix C. For both the denoised and the noisy case, and for each fixed tuple $(\sigma, \gamma, N_{\text{train}})$, we divide $\frac{N_{\text{train}}}{1+\gamma}$ equally among the first 5 classes, and $\frac{\gamma N_{\text{train}}}{1+\gamma}$ equally among the other 5 classes. We train the classifier 6 times, each time with a different seed, and report the average and standard deviation of the probabilities of error, to obtain a more reliable result. After we have the mean and standard deviation of the probability of error before and after the data processing, we compute the empirical efficiency, i.e., the relative percentage change in the probability of error induced by the denoising step.

Figure 2 presents the efficiency versus $N_{\text{train}}$. We see two main similarities to the theory. First, the non-monotonic behavior (increasing for small $N_{\text{train}}$ and decreasing for large $N_{\text{train}}$) is expected from the same argument in Section 3.3: the efficiency tends to zero as $N_{\text{train}}$ tends to either 0 or $\infty$, while, importantly, it remains positive between these two extreme cases, aligned with our theory. Second, we see that the maximal efficiency value decreases with $\sigma$: its value for $\sigma = 0.25$ is larger than its value for $\sigma = 0.4$. That is, the maximal efficiency increases with the SNR.

## 4.2 NOISY MINI-IMAGENET AND PRE-CLASSIFICATION ENCODING

We turn to investigate a more complex data processing pipeline using the Mini-ImageNet dataset (Vinyals et al., 2016) and the ResNet50 model. Both the training and test sets are subjected to additive Gaussian noise with standard deviations $\sigma \in \{50/255, 100/255\}$. The data processing step examined here is an encoding step, which maps the images from $224 \times 224$ pixels to 256-dimensional embeddings. This encoder model follows (Lu et al., 2025) and is trained from scratch with self-supervision on all noisy unlabeled images for each noise level. Then, for each combination of $(\sigma, \gamma, N_{\text{train}})$, we divide $\frac{N_{\text{train}}}{1+\gamma}$ equally among the first 50 classes, and $\frac{\gamma N_{\text{train}}}{1+\gamma}$ equally among the other 50 classes. Then, across three seeds, we train a ResNet50 model on the noisy images and, in parallel,

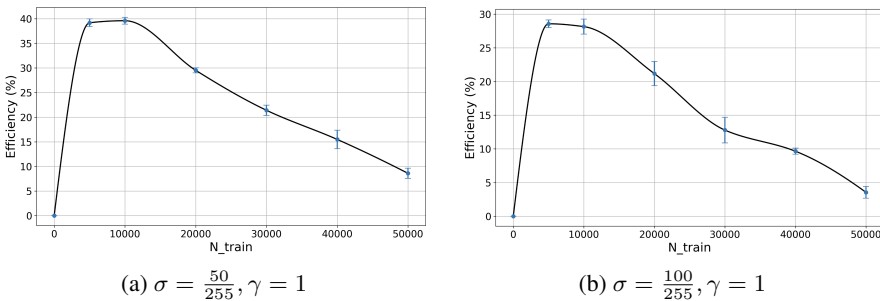

(a) $\sigma = \frac{50}{255}, \gamma = 1$        (b) $\sigma = \frac{100}{255}, \gamma = 1$

Figure 3: Noisy Mini-ImageNet and pre-classification encoding. Efficiency versus $N_{\text{train}}$.

a small MLP on the corresponding embeddings. After we have the mean and standard deviation of the probability of error before and after the data processing, we compute the empirical efficiency, i.e., the relative percentage change in the probability of error induced by the encoding step. Details of the training procedures for both the ResNet50 and the MLP are provided in Appendix D.

Figure 3 presents the efficiency versus $N_{\text{train}}$, for $\gamma = 1$. Experiments for $\gamma \in \{0.5, 0.75\}$ appear in Appendix D. We see the same trends that are aligned with our theory as before: 1) similar non-monotonicity of the curve while remaining positive, and 2) the maximal efficiency increases with the SNR. A message to practitioners is that when labeled samples are scarce, data processing can be especially advantageous for 'high quality' data.

### 4.3 Noisy CIFAR-10 and pre-classification encoding

For the noisy CIFAR-10 setup considered in Section 4.1, we also examine the performance of data processing based on encoding instead of denoising. Due to space limitations, the details are deferred to Appendix E, and the results are presented there in Figure 9. The trends stated above are observed there as well.

These results further demonstrate higher efficiency values compared to those obtained for the denoising procedure in Section 4.1, indicating that, for the classification task, encoding may be a more effective low-level processing method than denoising. However, we believe that this may not be the case for other high-level tasks, which may require preserving spatial information in the image (e.g., object detection).

## 5 Conclusion

In this paper, we addressed the question: How can we explain the common practice of performing a "low-level" task before a "high-level" downstream task, such as classification, despite theoretical principles like the data processing inequality and the overwhelming capabilities of modern deep neural networks? We presented a theoretical study of a binary classification setup, where we considered a "strong" classifier that is tightly connected to the optimal Bayes classifier (and converges to it), and yet, we constructed a pre-classification processing step that for any finite number of training samples provably improves the classification accuracy. We also provided both theoretical and empirical insights into various factors that affect the gains from such low-level processing. Finally, we demonstrated that the trends observed in four practical deep learning settings, where image denoising or encoding is applied before image classification, are consistent with those established by our theoretical study.

Our work motivates ongoing research on signal and image restoration and enhancement (Garber & Tirer, 2024; 2025; Zhang et al., 2025; Hen et al., 2026) and other low-level tasks. Since it shows the benefit of low-level tasks even when the classifier's training and test data share the same distribution, it naturally suggests an *even greater advantage* in out-of-distribution scenarios. As directions for future research, it would be valuable to extend the theoretical analysis to high-level tasks beyond classification or to investigate non-linear low-level processing. Another interesting direction is to study the optimal low-level processing corresponding to a given high-level task.

ACKNOWLEDGMENTS

The work was supported by the Israel Science Foundation (No. 1940/23) and MOST (No. 0007091) grants. LLMs were used only to polish the writing.

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

# A   PROOFS

## A.1   EXISTING RESULTS

Let us present a proof for Theorem 1, which is similar to a proof that can be found in an arXiv version of (Liu et al., 2019), but better clarifies how the Markovianity is used.

**Theorem 1**. *Let $y \to x \to z$ be a Markov chain where $y \in \{1, 2\}$ denotes the sample class. We have*

$$\mathbb{P}(c_{opt}(x) \neq y) \leq \mathbb{P}(\tilde{c}_{opt}(z) \neq y),$$

*where $c_{opt}$ and $\tilde{c}_{opt}$ denote optimal Bayes classifiers.*

*Proof.* Let us denote by $\mathcal{X}, \mathcal{Z}$ the supports of $x, z$, respectively, and by

$$P_1 := \mathbb{P}(y = 1), \ P_2 := \mathbb{P}(y = 2), \tag{15}$$

the prior probability for the binary label $y \in \{1, 2\}$. Let us also define:

$$p_{x_1}(\xi) := \mathbb{P}(x = \xi \mid y = 1), \ p_{x_2}(\xi) := \mathbb{P}(x = \xi \mid y = 2). \tag{16}$$

Now, from Eq. 15 and Eq. 16, the probability of error of the optimal Bayes classifier on $x$ reads:

$$
\begin{aligned}
\mathbb{P}(c_{opt}(x) \neq y) &= \sum_{\xi \in \mathcal{X}} \min (P_1 p_{x_1}(\xi), P_2 p_{x_2}(\xi)) \\
&= \frac{1}{2} - \frac{1}{2} \sum_{\xi \in \mathcal{X}} |P_1 p_{x_1}(\xi) - P_2 p_{x_2}(\xi)|.
\end{aligned}
\tag{17}
$$

Similarly to Eq. 16, we define:

$$p_{z_1}(\zeta) := \mathbb{P}(z = \zeta \mid y = 1), \ p_{z_2}(\zeta) := \mathbb{P}(z = \zeta \mid y = 2). \tag{18}$$

From Eq. 15 and Eq. 18, the probability of error of the optimal Bayes classifier on $z$ reads:

$$
\begin{aligned}
\mathbb{P}(\tilde{c}_{opt}(z) \neq y) &= \sum_{\zeta \in \mathcal{Z}} \min (P_1 p_{z_1}(\zeta), P_2 p_{z_2}(\zeta)) \\
&= \frac{1}{2} - \frac{1}{2} \sum_{\zeta \in \mathcal{Z}} |P_1 p_{z_1}(\zeta) - P_2 p_{z_2}(\zeta)|.
\end{aligned}
\tag{19}
$$

From Eq. 18 and the Markov assumption, we expand:

$$
\begin{aligned}
p_{z_i}(\zeta) = \mathbb{P}(z = \zeta \mid y = i) &= \sum_{\xi \in \mathcal{X}} \mathbb{P}(z = \zeta, x = \xi \mid y = i) \\
&= \sum_{\xi \in \mathcal{X}} \mathbb{P}(z = \zeta \mid x = \xi, \ y = i) \mathbb{P}(x = \xi \mid y = i) \\
&= \sum_{\xi \in \mathcal{X}} \mathbb{P}(z = \zeta \mid x = \xi) \mathbb{P}(x = \xi \mid y = i) \\
&= \sum_{\xi \in \mathcal{X}} p_{z|x}(\zeta \mid \xi) p_{x_i}(\xi).
\end{aligned}
\tag{20}
$$

The key step is the fourth equality, which eliminates the dependence on $y$ in the first factor of the summand. We also denote

$$p_{z|x}(\zeta \mid \xi) := \mathbb{P}(z = \zeta \mid x = \xi).$$

By substituting Eq. 20 into Eq. 19, we get:

$$
\begin{aligned}
\mathbb{P}\left(\tilde{c}_{opt}(z) \neq y\right) &= \frac{1}{2} - \frac{1}{2} \sum_{\zeta \in \mathcal{Z}} |P_1 p_{z_1}(\zeta) - P_2 p_{z_2}(\zeta)| \\
&= \frac{1}{2} - \frac{1}{2} \sum_{\zeta \in \mathcal{Z}} \left| \sum_{\xi \in \mathcal{X}} P_1 p_{z|x}(\zeta \mid \xi) p_{x_1}(\xi) - P_2 p_{z|x}(\zeta \mid \xi) p_{x_2}(\xi) \right| \\
&\geq \frac{1}{2} - \frac{1}{2} \sum_{\zeta \in \mathcal{Z}} \sum_{\xi \in \mathcal{X}} p_{z|x}(\zeta \mid \xi) \cdot |P_1 p_{x_1}(\xi) - P_2 p_{x_2}(\xi)| \\
&= \frac{1}{2} - \frac{1}{2} \sum_{\xi \in \mathcal{X}} \sum_{\zeta \in \mathcal{Z}} p_{z|x}(\zeta \mid \xi) \cdot |P_1 p_{x_1}(\xi) - P_2 p_{x_2}(\xi)| \\
&= \frac{1}{2} - \frac{1}{2} \sum_{\xi \in \mathcal{X}} \left( |P_1 p_{x_1}(\xi) - P_2 p_{x_2}(\xi)| \cdot \sum_{\zeta \in \mathcal{Z}} p_{z|x}(\zeta \mid \xi) \right) \\
&= \frac{1}{2} - \frac{1}{2} \sum_{\xi \in \mathcal{X}} |P_1 p_{x_1}(\xi) - P_2 p_{x_2}(\xi)| \\
&= \mathbb{P}\left(c_{opt}(x) \neq y\right).
\end{aligned}
\tag{21}
$$

$\square$

Let us now present a theorem that will be utilized in the proof of Theorem 2.

**Theorem 9** (Generalized Berry-Esseen Theorem, (Feller, 1991)). *Let $X_1, X_2, \ldots, X_d$ be independent random variables with:*

- *Means $\eta_i = \mathbb{E}[X_i]$.*

- *Variances $\xi_i^2 = \mathrm{Var}(X_i)$.*

- *Third absolute moments $\rho_i = \mathbb{E}\left[|X_i - \eta_i|^3\right]$.*

*Define the normalized sum:*

$$
S_d = \frac{1}{\sqrt{\sum_{i=1}^d \xi_i^2}} \sum_{i=1}^d (X_i - \eta_i).
$$

*Then, there exists an absolute constant $C_0 > 0$ independent of $d$ such that:*

$$
\sup_{x \in \mathbb{R}} |\mathbb{P}(S_d > x) - \mathcal{Q}(x)| \leq \frac{C_0 \sum_{i=1}^d \rho_i}{\left(\sum_{i=1}^d \xi_i^2\right)^{\frac{3}{2}}}.
$$

In the following subsections, we present the proofs of Theorems 2, 3, 4, 5, 6, 7, 8.

## A.2 PROOF OF THEOREM 2

*Proof.* The probability of error is

$$
\begin{aligned}
p_{\boldsymbol{x}}(\text{error}) &= \mathbb{P}\left(\widehat{c}(\boldsymbol{x}) \neq y\right) = \pi_1 \cdot \mathbb{P}\left(\widehat{c}(\boldsymbol{x}) = 2 \mid y = 1\right) + \pi_2 \cdot \mathbb{P}\left(\widehat{c}(\boldsymbol{x}) = 1 \mid y = 2\right) \\
&= \frac{1}{2} \cdot \mathbb{P}\left(\widehat{c}(\boldsymbol{x}) = 2 \mid y = 1\right) + \frac{1}{2} \cdot \mathbb{P}\left(\widehat{c}(\boldsymbol{x}) = 1 \mid y = 2\right) \\
&= \frac{1}{2} \cdot q(1, 2) + \frac{1}{2} \cdot q(2, 1)
\end{aligned}
\tag{22}
$$

where we used the assumption of a uniform prior and defined

$$q(i, j) = \mathbb{P}\left(\widehat{c}(\boldsymbol{x}) = j \mid y = i\right). \tag{23}$$

Following Eq. 6, the first conditional probability of error $q(1, 2)$ reads:

$$
\begin{aligned}
q(1, 2) &= \mathbb{P}\left(\|\boldsymbol{x} - \widehat{\boldsymbol{\mu}}_2\|^2 < \|\boldsymbol{x} - \widehat{\boldsymbol{\mu}}_1\|^2 \mid y = 1\right) \\
&= \mathbb{P}\left((\boldsymbol{x} - \widehat{\boldsymbol{\mu}}_2)^\top(\boldsymbol{x} - \widehat{\boldsymbol{\mu}}_2) < (\boldsymbol{x} - \widehat{\boldsymbol{\mu}}_1)^\top(\boldsymbol{x} - \widehat{\boldsymbol{\mu}}_1) \mid y = 1\right) \\
&= \mathbb{P}\left(\boldsymbol{x}^\top\boldsymbol{x} - \boldsymbol{x}^\top\widehat{\boldsymbol{\mu}}_2 - \widehat{\boldsymbol{\mu}}_2^\top\boldsymbol{x} + \widehat{\boldsymbol{\mu}}_2^\top\widehat{\boldsymbol{\mu}}_2 < \boldsymbol{x}^\top\boldsymbol{x} - \boldsymbol{x}^\top\widehat{\boldsymbol{\mu}}_1 - \widehat{\boldsymbol{\mu}}_1^\top\boldsymbol{x} + \widehat{\boldsymbol{\mu}}_1^\top\widehat{\boldsymbol{\mu}}_1 \mid y = 1\right) \\
&= \mathbb{P}\left(-2\widehat{\boldsymbol{\mu}}_2^\top\boldsymbol{x} + \|\widehat{\boldsymbol{\mu}}_2\|^2 < -2\widehat{\boldsymbol{\mu}}_1^\top\boldsymbol{x} + \|\widehat{\boldsymbol{\mu}}_1\|^2 \mid y = 1\right) \\
&= \mathbb{P}\left(2(\widehat{\boldsymbol{\mu}}_2 - \widehat{\boldsymbol{\mu}}_1)^\top\boldsymbol{x} > \|\widehat{\boldsymbol{\mu}}_2\|^2 - \|\widehat{\boldsymbol{\mu}}_1\|^2 \mid y = 1\right) \\
&= \mathbb{P}\left((\widehat{\boldsymbol{\mu}}_2 - \widehat{\boldsymbol{\mu}}_1)^\top\boldsymbol{x} > \frac{\|\widehat{\boldsymbol{\mu}}_2\|^2 - \|\widehat{\boldsymbol{\mu}}_1\|^2}{2} \mid y = 1\right) \\
&= \mathbb{P}\left((\widehat{\boldsymbol{\mu}}_2 - \widehat{\boldsymbol{\mu}}_1)^\top\boldsymbol{x} - \frac{\|\widehat{\boldsymbol{\mu}}_2\|^2 - \|\widehat{\boldsymbol{\mu}}_1\|^2}{2} > 0 \mid y = 1\right) \\
&= \mathbb{P}\left(w > 0 \mid y = 1\right)
\end{aligned}
\tag{24}
$$

where we defined

$$w = (\widehat{\boldsymbol{\mu}}_2 - \widehat{\boldsymbol{\mu}}_1)^\top \boldsymbol{x} - \frac{1}{2}\left(\|\widehat{\boldsymbol{\mu}}_2\|^2 - \|\widehat{\boldsymbol{\mu}}_1\|^2\right). \tag{25}$$

Let us define

$$y_i = (\widehat{\mu}_2)_i \cdot x_i - (\widehat{\mu}_1)_i \cdot x_i - \frac{1}{2} \cdot (\widehat{\mu}_2)_i^2 + \frac{1}{2} \cdot (\widehat{\mu}_1)_i^2. \tag{26}$$

Thus,

$$w = \sum_{i=1}^d y_i. \tag{27}$$

In total, from Eq. 24, Eq. 27, it follows that:

$$q(1, 2) = \mathbb{P}\left(\sum_{i=1}^d y_i > 0 \mid y = 1\right). \tag{28}$$

The setup of our theoretical investigation clearly implies that the random variables $\{y_i\}_{i=1}^d$, defined in Eq. 26, are independent, and thus we can apply Theorem 9. Let us now compute the following expressions, that will be crucial when applying Theorem 9:

1.
$$\eta_i := \mathbb{E}[y_i] \tag{29}$$

2.
$$\xi_i^2 := \mathrm{Var}(y_i) \tag{30}$$

3.
$$\rho_i := \mathbb{E}\left[|y_i - \eta_i|^3\right] \tag{31}$$

Note that:

$$\widehat{\boldsymbol{\mu}}_j \sim \mathcal{N}\left(\boldsymbol{\mu}_j, \frac{\sigma^2}{N_j}\boldsymbol{I}_d\right).$$

Thus, given equations Eq. 3, Eq. 4, for each $1 \le i \le d$, we have:

$$p_{j,i} := (\widehat{\mu}_j)_i \sim \mathcal{N}\left((\mu_j)_i, \frac{\sigma^2}{N_j}\right), \quad x_i \mid y = 1 \sim \mathcal{N}\left(-\mu_i, \sigma^2\right) \tag{32}$$

and from Eq. 26, it follows that:

$$y_i = p_{2,i}x_i - p_{1,i}x_i - \frac{1}{2}p_{2,i}^2 + \frac{1}{2}p_{1,i}^2 \tag{33}$$

where from Eq. 4, Eq. 32, we have:

$$p_{2,i} \sim \mathcal{N}\left(\mu_i, \frac{\sigma^2}{N_2}\right), \; p_{1,i} \sim \mathcal{N}\left(-\mu_i, \frac{\sigma^2}{N_1}\right). \tag{34}$$

For every $1 \leq i \leq d$, let us define the following random variables:

$$a_i := (p_{2,i} - p_{1,i}) \cdot x_i, \; b_i := p_{2,i}^2 - p_{1,i}^2. \tag{35}$$

Thus, from Eq. 33, it follows that:

$$y_i = a_i - \frac{1}{2}b_i. \tag{36}$$

We first compute $\eta_i = \mathbb{E}[y_i]$ : From Eq. 36, it follows that:

$$\eta_i = \mathbb{E}[y_i] = \mathbb{E}[a_i] - \frac{1}{2}\mathbb{E}[b_i]. \tag{37}$$

We now compute each expectation separately. From Eq. 32, Eq. 34, and the assumption of independence, it follows that:

$$\begin{aligned}
\mathbb{E}[a_i] &= \mathbb{E}[p_{2,i}x_i] - \mathbb{E}[p_{2,i}x_i] \\
&= \mathbb{E}[p_{2,i}] \cdot \mathbb{E}[x_i] - \mathbb{E}[p_{1,i}] \cdot \mathbb{E}[x_i] \\
&= -\mu_i^2 - \mu_i^2 \\
&= -2\mu_i^2
\end{aligned} \tag{38}$$

and

$$\begin{aligned}
\mathbb{E}[b_i] &= \mathbb{E}[p_{2,i}^2] - \mathbb{E}[p_{1,i}^2] \\
&= \left(\frac{\sigma^2}{N_2} + \mu_i^2\right) - \left(\frac{\sigma^2}{N_1} + \mu_i^2\right) \\
&= \sigma^2\left(\frac{1}{\gamma N} - \frac{1}{N}\right) \\
&= \frac{1-\gamma}{\gamma} \cdot \frac{\sigma^2}{N}.
\end{aligned} \tag{39}$$

Thus, from Eq. 37, Eq. 38, Eq. 39, we have:

$$\eta_i = -2\mu_i^2 - \frac{1}{2} \cdot \frac{1-\gamma}{\gamma} \cdot \frac{\sigma^2}{N} = -\left(2\mu_i^2 + \frac{1-\gamma}{\gamma} \cdot \frac{\sigma^2}{2N}\right). \tag{40}$$

We now compute $\xi_i^2 = \mathrm{Var}(y_i)$ : From Eq. 36, we have:

$$\xi_i^2 = \mathrm{Var}(y_i) = \mathrm{Var}(a_i) + \frac{1}{4} \cdot \mathrm{Var}(b_i) - \mathrm{Cov}(a_i, b_i). \tag{41}$$

We now compute each piece separately, starting from $\mathrm{Var}(a_i)$. From equations Eq. 32, Eq. 34, Eq. 35, Eq. 38, it follows that:

$$\begin{aligned}
\mathrm{Var}(a_i) &= \mathbb{E}[a_i^2] - \mathbb{E}[a_i]^2 \\
&= \mathbb{E}\left[(p_{2,i} - p_{1,i})^2\right] \cdot \mathbb{E}[x_i^2] - 4\mu_i^4 \\
&= \left(\frac{1+\gamma}{\gamma} \cdot \frac{\sigma^2}{N} + 4\mu_i^2\right) \cdot (\sigma^2 + \mu_i^2) - 4\mu_i^4 \\
&= \frac{1+\gamma}{\gamma} \cdot \frac{\sigma^4}{N} + \mu_i^2 \cdot \left(\frac{1+\gamma}{\gamma} \cdot \frac{\sigma^2}{N} + 4\sigma^2\right)
\end{aligned} \tag{42}$$

where we used the statistical independence between $p_{2,i}, p_{1,i}$ and Eq. 34, to conclude that:

$$p_{2,i} - p_{1,i} \sim \mathcal{N}\left(2\mu_i, \frac{\sigma^2}{N_1} + \frac{\sigma^2}{N_2}\right) \Rightarrow p_{2,i} - p_{1,i} \sim \mathcal{N}\left(2\mu_i, \frac{1+\gamma}{\gamma} \cdot \frac{\sigma^2}{N}\right). \qquad (43)$$

We now compute $\mathrm{Var}(b_i)$. From equations Eq. 34, Eq. 35, and the statistical independence between $p_{2,i}, p_{1,i}$, it follows that:

$$
\begin{aligned}
\mathrm{Var}(b_i) &= \mathrm{Var}\left(p_{2,i}^2 - p_{1,i}^2\right) \\
&= \mathrm{Var}\left(p_{2,i}^2\right) + \mathrm{Var}\left(p_{1,i}^2\right) \\
&= \frac{2\sigma^2}{N_2}\left(\frac{\sigma^2}{N_2} + 2\mu_i^2\right) + \frac{2\sigma^2}{N_1}\left(\frac{\sigma^2}{N_1} + 2\mu_i^2\right) \\
&= \frac{2\sigma^4}{\gamma^2 N^2} + \frac{4\sigma^2 \mu_i^2}{\gamma N} + \frac{2\sigma^4}{N^2} + \frac{4\sigma^2 \mu_i^2}{N} \\
&= \frac{1+\gamma^2}{\gamma^2} \cdot \frac{2\sigma^4}{N^2} + \frac{1+\gamma}{\gamma} \cdot \frac{4\sigma^2 \mu_i^2}{N}
\end{aligned}
\qquad (44)
$$

where we used the fact that if $x \sim \mathcal{N}\left(\mu_x, \sigma_x^2\right)$, then

$$
\begin{aligned}
\mathrm{Var}(x^2) &= \mathbb{E}[x^4] - \mathbb{E}[x^2]^2 \\
&= \left(3\sigma_x^4 + 6\sigma_x^2 \mu_x^2 + \mu_x^4\right) - \left(\sigma_x^2 + \mu_x^2\right)^2 \\
&= \left(3\sigma_x^4 + 6\sigma_x^2 \mu_x^2 + \mu_x^4\right) - \left(\sigma_x^4 + 2\sigma_x^2 \mu_x^2 + \mu_x^4\right) \\
&= 2\sigma_x^4 + 4\sigma_x^2 \mu_x^2 \\
&= 2\sigma_x^2 \cdot \left(\sigma_x^2 + 2\mu_x^2\right).
\end{aligned}
$$

Finally, we compute $\mathrm{Cov}(a_i, b_i)$. From equations Eq. 32, Eq. 34, Eq. 35, Eq. 38, Eq. 39, it follows that:

$$
\begin{aligned}
\mathrm{Cov}(a_i, b_i) &= \mathbb{E}\left[a_i b_i\right] - \mathbb{E}[a_i] \cdot \mathbb{E}[b_i] \\
&= \mathbb{E}\left[(p_{2,i} - p_{1,i})\left(p_{2,i}^2 - p_{1,i}^2\right) \cdot x_i\right] + \frac{1-\gamma}{\gamma} \cdot \frac{2\sigma^2 \mu_i^2}{N} \\
&= \left(\mathbb{E}\left[p_{2,i}^3\right] - \mathbb{E}\left[p_{2,i}\right] \cdot \mathbb{E}\left[p_{1,i}^2\right] - \mathbb{E}\left[p_{1,i}\right] \cdot \mathbb{E}\left[p_{2,i}^2\right] + \mathbb{E}\left[p_{1,i}^3\right]\right) \cdot \mathbb{E}\left[x_i\right] + \frac{1-\gamma}{\gamma} \cdot \frac{2\sigma^2 \mu_i^2}{N} \\
&= -\mu_i \cdot \left(\left(\mu_i^3 + 3\mu_i \cdot \frac{\sigma^2}{N_2}\right) - \mu_i \cdot \left(\frac{\sigma^2}{N_1} + \mu_i^2\right) + \mu_i \cdot \left(\frac{\sigma^2}{N_2} + \mu_i^2\right) - \left(\mu_i^3 + 3\mu_i \cdot \frac{\sigma^2}{N_1}\right)\right) \\
&\quad + \frac{1-\gamma}{\gamma} \cdot \frac{2\sigma^2 \mu_i^2}{N} \\
&= -\mu_i \cdot \left(4\mu_i \cdot \frac{\sigma^2}{\gamma N} - 4\mu_i \cdot \frac{\sigma^2}{N}\right) + \frac{1-\gamma}{\gamma} \cdot \frac{2\sigma^2 \mu_i^2}{N} \\
&= \frac{4\sigma^2 \mu_i^2}{N} - \frac{4\sigma^2 \mu_i^2}{\gamma N} + \frac{1-\gamma}{\gamma} \cdot \frac{2\sigma^2 \mu_i^2}{N} \\
&= -\frac{1-\gamma}{\gamma} \cdot \frac{4\sigma^2 \mu_i^2}{N} + \frac{1-\gamma}{\gamma} \cdot \frac{2\sigma^2 \mu_i^2}{N} \\
&= -\frac{1-\gamma}{\gamma} \cdot \frac{2\sigma^2 \mu_i^2}{N}
\end{aligned}
$$

$$(45)$$

where we used the fact that if $x \sim \mathcal{N}\left(\mu_x, \sigma_x^2\right)$, then

$$\mathbb{E}[x^3] = \mu_x^3 + 3\mu_x \sigma_x^2. \qquad (46)$$

Thus, from equations Eq. 41, Eq. 42, Eq. 44, Eq. 45, it follows that:

$$
\begin{aligned}
\xi_i^2 &= \frac{1+\gamma}{\gamma}\cdot\frac{\sigma^4}{N} + \mu_i^2\cdot\left(\frac{1+\gamma}{\gamma}\cdot\frac{\sigma^2}{N} + 4\sigma^2\right) \\
&\quad + \frac{1}{4}\cdot\left(\frac{1+\gamma^2}{\gamma^2}\cdot\frac{2\sigma^4}{N^2} + \frac{1+\gamma}{\gamma}\cdot\frac{4\sigma^2\mu_i^2}{N}\right) + \frac{1-\gamma}{\gamma}\cdot\frac{2\sigma^2\mu_i^2}{N} \\
&= \frac{1+\gamma}{\gamma}\cdot\frac{\sigma^4}{N} + \frac{1+\gamma^2}{\gamma^2}\cdot\frac{\sigma^4}{2N^2} + \mu_i^2\left(\frac{1+\gamma}{\gamma}\cdot\frac{2\sigma^2}{N} + \frac{1-\gamma}{\gamma}\cdot\frac{2\sigma^2}{N} + 4\sigma^2\right) \\
&= \sigma^2\left(\frac{1+\gamma}{\gamma}\cdot\frac{\sigma^2}{N} + \frac{1+\gamma^2}{2\gamma^2}\cdot\frac{\sigma^2}{N^2} + 4\mu_i^2\left(1 + \frac{1+\gamma}{2\gamma}\cdot\frac{1}{N} + \frac{1-\gamma}{2\gamma}\cdot\frac{1}{N}\right)\right) \\
&= \sigma^2\left(\frac{1+\gamma}{\gamma}\cdot\frac{\sigma^2}{N} + \frac{1+\gamma^2}{2\gamma^2}\cdot\frac{\sigma^2}{N^2} + 4\mu_i^2\left(1 + \frac{1}{\gamma N}\right)\right).
\end{aligned} \tag{47}
$$

We get the following lower bound:

$$
\xi_i^2 \geq D := \frac{\sigma^4}{N}\cdot\left(\frac{1+\gamma}{\gamma} + \frac{1+\gamma^2}{2\gamma^2}\cdot\frac{1}{N}\right). \tag{48}
$$

Finally, we compute $\rho_i = \mathbb{E}\left[|y_i - \eta_i|^3\right]$: We will show that $\rho_i$ is globally bounded. We first note the following inequality, which holds for any real-valued random variable $x$ with $\mathbb{E}[x^4] < \infty$:

$$
\mathbb{E}\left[|x|^3\right] \leq \left(\mathbb{E}\left[x^4\right]\right)^{\frac{3}{4}}.
$$

This is a consequence of Lyapunov's inequality. Setting $x = y_i - \mu_i$ yields the following upper bound:

$$
\rho_i = \mathbb{E}\left[|y_i - \eta_i|^3\right] \leq \left(\mathbb{E}\left[(y_i - \eta_i)^4\right]\right)^{\frac{3}{4}}. \tag{49}
$$

We now expand:

$$
(y_i - \eta_i)^4 = y_i^4 - 4y_i^3\eta_i + 6y_i^2\eta_i^2 - 4y_i\eta_i^3 + \eta_i^4
$$

which, from Eq. 49, implies that:

$$
\begin{aligned}
\rho_i &\leq \left(\mathbb{E}[y_i^4] - 4\eta_i\cdot\mathbb{E}[y_i^3] + 6\eta_i^2\cdot\mathbb{E}[y_i^2] - 4\eta_i^3\cdot\mathbb{E}[y_i] + \eta_i^4\right)^{\frac{3}{4}} \\
&= \left(\mathbb{E}[y_i^4] - 4\eta_i\cdot\mathbb{E}[y_i^3] + 6\eta_i^2\cdot\left(\mathrm{Var}(y_i) + \mathbb{E}[y_i]^2\right) - 3\eta_i^4\right)^{\frac{3}{4}} \\
&= \left(\mathbb{E}[y_i^4] - 4\eta_i\cdot\mathbb{E}[y_i^3] + 6\eta_i^2\cdot\left(\xi_i^2 + \eta_i^2\right) - 3\eta_i^4\right)^{\frac{3}{4}} \\
&= \left(\mathbb{E}[y_i^4] - 4\eta_i\cdot\mathbb{E}[y_i^3] + 6\eta_i^2\xi_i^2 + 3\eta_i^4\right)^{\frac{3}{4}}
\end{aligned} \tag{50}
$$

where we used the definitions $\xi_i^2 = \mathrm{Var}(y_i)$, $\eta_i = \mathbb{E}[y_i]$. It is now left to compute

$$
\chi_i := \mathbb{E}[y_i^4], \ \delta_i := \mathbb{E}[y_i^3] \tag{51}
$$

which implies, from Eq. 50, that

$$
\rho_i \leq \left(\chi_i - 4\delta_i\eta_i + 6\xi_i^2\eta_i^2 + 3\eta_i^4\right)^{\frac{3}{4}}. \tag{52}
$$

Let $f \in \{\eta, \xi^2, \delta, \chi\}$. We argue that for all $1 \leq i \leq d$:

$$
f_i = \sum_{k=0}^{q(i,f)} c_k(i,f)\cdot\mu_i^k \tag{53}
$$

where $q(i,f) \in \mathbb{N}$ and the constants $\{c_k(i,f)\}_{k=0}^{q(i,f)}$ don't depend on $d$. We already saw that $\eta_i, \xi_i$ follows that structure in equations Eq. 40, Eq. 41.

We now compute $\delta_i$. From Eq. 36, it follows that:

$$
\begin{aligned}
\delta_i &= \mathbb{E}\left[\left(a_i - \frac{1}{2}b_i\right)^3\right] \\
&= \mathbb{E}\left[a_i^3\right] - \frac{3}{2}\mathbb{E}\left[a_i^2 b_i\right] + \frac{3}{4}\mathbb{E}\left[a_i b_i^2\right] - \frac{1}{8}\mathbb{E}\left[b_i^3\right]
\end{aligned} \tag{54}
$$

we now compute each part separately, starting with $\mathbb{E}[a_i^3]$. From equations Eq. 32, Eq. 35, and the assumption of independence, it follows that:

$$
\begin{aligned}
\mathbb{E}\left[a_i^3\right] &= \mathbb{E}\left[(p_{2,i} - p_{1,i})^3 \cdot x_i^3\right] \\
&= \mathbb{E}\left[(p_{2,i} - p_{1,i})^3\right] \cdot \mathbb{E}\left[x_i^3\right] \\
&= \left(8\mu_i^3 + 6\mu_i \cdot \frac{1+\gamma}{\gamma} \cdot \frac{\sigma^2}{N}\right)\left(-\mu_i^3 - 3\mu_i\sigma^2\right) \\
&= -\mu_i^2\left(8\mu_i^2 + \frac{6(1+\gamma)}{\gamma} \cdot \frac{\sigma^2}{N}\right)\left(\mu_i^2 + 3\sigma^2\right)
\end{aligned}
\tag{55}
$$

which is a polynomial in $\mu_i$ with real coefficients. The other expressions are computed similarly, and they all have the form as in Eq. 53. We now turn to the assumption that

$$
\exists_{M \geq 0} \, \forall_{d \in \mathbb{N}} \, \forall_{1 \leq i \leq d} \; |\mu_i| \leq M \tag{56}
$$

and thus for each $f \in \mathcal{F} := \{\eta, \xi^2, \delta, \chi\}$ and for all $1 \leq i \leq d$, from the triangle inequality, it follows that:

$$
\begin{aligned}
|f_i| &\leq \sum_{k=0}^{q(i,f)} |c_k(i,f)| \cdot |\mu_i|^k \leq \sum_{k=0}^{q(i,f)} |c_k(i,f)| \cdot M^k \\
&\leq \sum_{k=0}^{\max_{f \in \mathcal{F}} q(i,f)} \max_{f \in \mathcal{F}} |c_k(i,f)| \cdot M^k \\
&\leq \max_{1 \leq i \leq d} \sum_{k=0}^{\max_{f \in \mathcal{F}} q(i,f)} \max_{f \in \mathcal{F}} |c_k(i,f)| \cdot M^k
\end{aligned}
\tag{57}
$$

where we define $c_k(i,f) = 0$ for all $k > q(i,f)$. Let us denote

$$
L := \max_{1 \leq i \leq d} \sum_{k=0}^{\max_{f \in \mathcal{F}} q(i,f)} \max_{f \in \mathcal{F}} |c_k(i,f)| \cdot M^k. \tag{58}
$$

Thus, from Eq. 57, we have:

$$
\forall_{f \in \mathcal{F}} \forall_{1 \leq i \leq d} \; |f_i| \leq L. \tag{59}
$$

Now, $L$ is independent of $i$ (because we took the maximum over all possible $1 \leq i \leq d$) and $d$ (because the degree $q$ and the coefficients $c$ will never depend directly on $d$, because $\sigma$ doesn't depend on $d$). We thus showed that the absolute value of each relevant moment is upper bounded by a global value $L \geq 0$ that is independent of $i$ and $d$. Thus, from Eq. 52, Eq. 59, it follows that:

$$
\begin{aligned}
\rho_i &\leq \left(\left|\chi_i - 4\delta_i\eta_i + 6\xi_i^2\eta_i^2 + 3\eta_i^4\right|\right)^{\frac{3}{4}} \\
&\leq \left(|\chi_i| + 4|\delta_i||\eta_i| + 6\xi_i^2\eta_i^2 + 3\eta_i^4\right)^{\frac{3}{4}} \\
&\leq \left(L + 4L^2 + 6L^4 + 3L^4\right)^{\frac{3}{4}} \\
&= \left(9L^4 + 4L^2 + L\right)^{\frac{3}{4}}.
\end{aligned}
$$

Let us now denote $C = \left(9L^4 + 4L^2 + L\right)^{\frac{3}{4}}$, where $L \geq 0$ is defined in Eq. 58. Thus, $C \geq 0$ is independent of both $i$ and $d$, and

$$
\forall_{1 \leq i \leq d} \, \rho_i \leq C.
$$

When combining this result with Eq. 48, we get that there exists some $C \geq 0$, $D > 0$ that doesn't depend on $i$ or $d$ such that

$$
\rho_i \leq C, \; \xi_i^2 \geq D.
$$

Thus,

$$
\frac{\sum_{i=1}^d \rho_i}{\left(\sum_{i=1}^d \xi_i^2\right)^{\frac{3}{2}}} \leq \frac{\sum_{i=1}^d C}{\left(\sum_{i=1}^d D\right)^{\frac{3}{2}}} \leq \frac{C}{D^{\frac{3}{2}}\sqrt{d}}. \tag{60}
$$

We have verified that the conditions of Theorem 9 are satisfied, and thus, there exists some $C_0 > 0$ independent of $d$ such that for all $x \in \mathbb{R}$

$$\left| \mathbb{P}\left( \frac{1}{\sqrt{\sum_{i=1}^d \xi_i^2}} \sum_{i=1}^d (y_i - \eta_i) > x \mid y = 1 \right) - \mathcal{Q}(x) \right| \le \frac{C_0 \sum_{i=1}^d \rho_i}{\left( \sum_{i=1}^d \xi_i^2 \right)^{\frac{3}{2}}} \le \frac{A}{\sqrt{d}}$$

where we used Eq. 60, and denoted $A = \dfrac{C_0 C}{D^{\frac{3}{2}}} \ge 0$. Now, $q(1, 2)$, which is defined in Eq. 28, reads:

$$q(1, 2) = \mathbb{P}\left( \sum_{i=1}^d y_i > 0 \mid y = 1 \right)$$

$$= \mathbb{P}\left( \sum_{i=1}^d (y_i - \eta_i) > -\sum_{i=1}^d \eta_i \mid y = 1 \right)$$

$$= \mathbb{P}\left( \frac{1}{\sqrt{\sum_{i=1}^d \xi_i^2}} \sum_{i=1}^d (y_i - \eta_i) > -\frac{\sum_{i=1}^d \eta_i}{\sqrt{\sum_{i=1}^d \xi_i^2}} \mid y = 1 \right)$$

$$= \mathcal{Q}\left( -\frac{\sum_{i=1}^d \eta_i}{\sqrt{\sum_{i=1}^d \xi_i^2}} \right) + \mathcal{O}\left( \frac{1}{\sqrt{d}} \right)$$

$$= \mathcal{Q}\left( \frac{\sum_{i=1}^d \left( 2\mu_i^2 + \frac{1-\gamma}{\gamma} \cdot \frac{\sigma^2}{2N} \right)}{\sqrt{\sum_{i=1}^d \sigma^2 \left( \frac{1+\gamma}{\gamma} \cdot \frac{\sigma^2}{N} + \frac{1+\gamma^2}{2\gamma^2} \cdot \frac{\sigma^2}{N^2} + 4\mu_i^2 \left( 1 + \frac{1}{\gamma N} \right) \right)}} \right) + \mathcal{O}\left( \frac{1}{\sqrt{d}} \right)$$

$$= \mathcal{Q}\left( \frac{2\|\boldsymbol{\mu}\|^2 + \frac{d}{2N} \cdot \frac{1-\gamma}{\gamma} \cdot \sigma^2}{\sqrt{\sigma^2 \cdot \left( \left( \frac{1+\gamma}{\gamma} \cdot \frac{\sigma^2}{N} + \frac{1+\gamma^2}{2\gamma^2} \cdot \frac{\sigma^2}{N^2} \right) d + 4 \left( 1 + \frac{1}{\gamma N} \right) \|\boldsymbol{\mu}\|^2 \right)}} \right) + \mathcal{O}\left( \frac{1}{\sqrt{d}} \right)$$

$$= \mathcal{Q}\left( \frac{\|\boldsymbol{\mu}\| + \frac{d}{4N} \cdot \frac{1-\gamma}{\gamma} \cdot \frac{\sigma^2}{\|\boldsymbol{\mu}\|}}{\sigma \cdot \sqrt{\left( \frac{1}{4N} \cdot \frac{1+\gamma}{\gamma} \cdot \left( \frac{\sigma}{\|\boldsymbol{\mu}\|} \right)^2 + \frac{1}{8N^2} \cdot \frac{1+\gamma^2}{\gamma^2} \cdot \left( \frac{\sigma}{\|\boldsymbol{\mu}\|} \right)^2 \right) \cdot d + \left( 1 + \frac{1}{\gamma N} \right)}} \right) + \mathcal{O}\left( \frac{1}{\sqrt{d}} \right)$$

$$= \mathcal{Q}\left( \frac{\frac{\|\boldsymbol{\mu}\|}{\sigma} + \frac{d}{4N} \cdot \frac{1-\gamma}{\gamma} \cdot \frac{\sigma}{\|\boldsymbol{\mu}\|}}{\sqrt{\left( \frac{1}{4N} \cdot \frac{1+\gamma}{\gamma} \cdot \left( \frac{\sigma}{\|\boldsymbol{\mu}\|} \right)^2 + \frac{1}{8N^2} \cdot \frac{1+\gamma^2}{\gamma^2} \cdot \left( \frac{\sigma}{\|\boldsymbol{\mu}\|} \right)^2 \right) \cdot d + \left( 1 + \frac{1}{\gamma N} \right)}} \right) + \mathcal{O}\left( \frac{1}{\sqrt{d}} \right).$$

$$\tag{61}$$

We now revisit Eq. 5:

$$\mathcal{S} = \left( \frac{\|\boldsymbol{\mu}\|}{\sigma} \right)^2.$$

Thus, from Eq. 61, $q(1, 2)$ reads:

$$q(1, 2) = \mathcal{Q}\left( \frac{\sqrt{\mathcal{S}} + \frac{1}{4N} \cdot \frac{1-\gamma}{\gamma} \cdot \frac{d}{\sqrt{\mathcal{S}}}}{\sqrt{\frac{1}{4N} \cdot \frac{1+\gamma}{\gamma} \cdot \frac{d}{\mathcal{S}} + \frac{1}{8N^2} \cdot \frac{1+\gamma^2}{\gamma^2} \cdot \frac{d}{\mathcal{S}} + \frac{1}{\gamma N} + 1}} \right) + \mathcal{O}\left( \frac{1}{\sqrt{d}} \right). \tag{62}$$

We now compute $q(2,1)$. Similarly to the computation of $q(1,2)$, we have:

$$
\begin{aligned}
q(2,1) &= \mathbb{P}(\widehat{c}(\boldsymbol{x}) = 1 \mid y = 2) \\
&= \mathbb{P}\left( (\widehat{\boldsymbol{\mu}}_2 - \widehat{\boldsymbol{\mu}}_1)^\top \boldsymbol{x} < \frac{\|\widehat{\boldsymbol{\mu}}_2\|^2 - \|\widehat{\boldsymbol{\mu}}_1\|^2}{2} \mid y = 2 \right) \\
&= \mathbb{P}(w < 0 \mid y = 2) \\
&= \mathbb{P}\left( \sum_{i=1}^{d} y_i < 0 \mid y = 2 \right)
\end{aligned}
\tag{63}
$$

where the random variables $\{y_i\}_{i=1}^d$ are defined in Eq. 33. The new conditional distribution of $x_i$ is:

$$
\forall_{1 \leq i \leq d} \, x_i \mid y = 2 \sim \mathcal{N}(\mu_i, \sigma^2).
\tag{64}
$$

We first compute $\eta_i = \mathbb{E}[y_i]$. It is easy to verify from Eq. 37, Eq. 38, Eq. 39, Eq. 64, that $\eta_i$ is given by:

$$
\eta_i = 2\mu_i^2 - \frac{1-\gamma}{\gamma} \cdot \frac{\sigma^2}{2N}.
\tag{65}
$$

We now compute $\xi_i^2 = \mathrm{Var}\,(y_i)$. It still has three components, as in Eq. 41. It is easy to see from Eq. 44 that $\mathrm{Var}(b_i)$ remains unchanged because it doesn't depend on the conditional distribution of $\boldsymbol{x}_i$. Thus,

$$
\mathrm{Var}(b_i) = \frac{1+\gamma^2}{\gamma^2} \cdot \frac{2\sigma^4}{N^2} + \frac{1+\gamma}{\gamma} \cdot \frac{4\sigma^2 \mu_i^2}{N}.
\tag{66}
$$

We observe from Eq. 42, Eq. 64 that $\mathrm{Var}(a_i)$ remains unchanged since it depends on $\mathbb{E}[\boldsymbol{x}_i^2] = \sigma^2 + \mu_i^2$, which is unaffected.

$$
\mathrm{Var}(a_i) = \frac{1+\gamma}{\gamma} \cdot \frac{\sigma^4}{N} + \mu_i^2 \cdot \left( \frac{1+\gamma}{\gamma} \cdot \frac{\sigma^2}{N} + 4\sigma^2 \right).
\tag{67}
$$

It remains to compute $\mathrm{Cov}(a_i, b_i) = \mathbb{E}[a_i b_i] - \mathbb{E}[a_i]\mathbb{E}[b_i]$. From Eq. 38 and Eq. 64, we have $\mathbb{E}[a_i] = 2\mu_i^2$. According to Eq. 39, $\mathbb{E}[b_i]$ is unchanged, as it does not depend on the conditional distribution of $\boldsymbol{x}_i$. Similarly, from Eq. 45, $\mathbb{E}[a_i b_i]$ picks up a minus sign, so overall, $\mathrm{Cov}(a_i, b_i)$ changes sign. Therefore, Eq. 45 implies:

$$
\mathrm{Cov}(a_i, b_i) = \frac{1-\gamma}{\gamma} \cdot \frac{2\sigma^2 \mu_i^2}{N}.
\tag{68}
$$

Thus, from equations Eq. 41, Eq. 66, Eq. 67, Eq. 68, it follows that:

$$
\begin{aligned}
\xi_i^2 &= \frac{1+\gamma}{\gamma} \cdot \frac{\sigma^4}{N} + \mu_i^2 \cdot \left( \frac{1+\gamma}{\gamma} \cdot \frac{\sigma^2}{N} + 4\sigma^2 \right) \\
&\quad + \frac{1}{4} \cdot \left( \frac{1+\gamma^2}{\gamma^2} \cdot \frac{2\sigma^4}{N^2} + \frac{1+\gamma}{\gamma} \cdot \frac{4\sigma^2 \mu_i^2}{N} \right) - \frac{1-\gamma}{\gamma} \cdot \frac{2\sigma^2 \mu_i^2}{N} \\
&= \frac{1+\gamma}{\gamma} \cdot \frac{\sigma^4}{N} + \frac{1+\gamma^2}{\gamma^2} \cdot \frac{\sigma^4}{2N^2} + \mu_i^2 \left( \frac{1+\gamma}{\gamma} \cdot \frac{2\sigma^2}{N} - \frac{1-\gamma}{\gamma} \cdot \frac{2\sigma^2}{N} + 4\sigma^2 \right) \\
&= \sigma^2 \left( \frac{1+\gamma}{\gamma} \cdot \frac{\sigma^2}{N} + \frac{1+\gamma^2}{2\gamma^2} \cdot \frac{\sigma^2}{N^2} + 4\mu_i^2 \left( 1 + \frac{1+\gamma}{2\gamma} \cdot \frac{1}{N} - \frac{1-\gamma}{2\gamma} \cdot \frac{1}{N} \right) \right) \\
&= \sigma^2 \left( \frac{1+\gamma}{\gamma} \cdot \frac{\sigma^2}{N} + \frac{1+\gamma^2}{2\gamma^2} \cdot \frac{\sigma^2}{N^2} + 4\mu_i^2 \left( 1 + \frac{1}{N} \right) \right) \\
&\geq \frac{\sigma^4}{N} \cdot \left( \frac{1+\gamma}{\gamma} + \frac{1+\gamma^2}{2\gamma^2} \cdot \frac{1}{N} \right) = D.
\end{aligned}
\tag{69}
$$

Thus, $\xi_i^2 \geq D$ where $D > 0$ is the same constant defined in Eq. 48. A similar argument for the case $y = 1$ shows that $\rho_i = \mathbb{E}[|y_i - \eta_i|^3] \leq C$, where $C \geq 0$ and $D > 0$ are constants independent of $i$

and $d$. Since the variables $\{y_i\}_{i=1}^d$ are independent, we may apply Theorem 9, which guarantees the existence of a constant $C_0 > 0$ independent of $d$ such that for all $x \in \mathbb{R}$:

$$\left| \mathbb{P} \left( \frac{1}{\sqrt{\sum_{i=1}^d \xi_i^2}} \sum_{i=1}^d (y_i - \eta_i) > x \mid y = 2 \right) - \mathcal{Q}(x) \right| \leq \frac{C_0 \sum_{i=1}^d \rho_i}{\left( \sum_{i=1}^d \xi_i^2 \right)^{\frac{3}{2}}} \leq \frac{A}{\sqrt{d}}.$$

Where we denoted $A = \dfrac{C_0 C}{D^{\frac{3}{2}}} \geq 0$. Now, $q(2,1)$, which is defined in Eq. 63, reads:

$$q(2,1) = \mathbb{P} \left( \sum_{i=1}^d y_i < 0 \mid y = 2 \right)$$

$$= \mathbb{P} \left( \sum_{i=1}^d (y_i - \eta_i) < - \sum_{i=1}^d \eta_i \mid y = 2 \right)$$

$$= \mathbb{P} \left( \frac{1}{\sqrt{\sum_{i=1}^d \xi_i^2}} \sum_{i=1}^d (y_i - \eta_i) < - \frac{\sum_{i=1}^d \eta_i}{\sqrt{\sum_{i=1}^d \xi_i^2}} \mid y = 2 \right)$$

$$= 1 - \mathbb{P} \left( \frac{1}{\sqrt{\sum_{i=1}^d \xi_i^2}} \sum_{i=1}^d (y_i - \eta_i) \geq - \frac{\sum_{i=1}^d \eta_i}{\sqrt{\sum_{i=1}^d \xi_i^2}} \mid y = 2 \right)$$

$$= 1 - \left( \mathcal{Q} \left( - \frac{\sum_{i=1}^d \eta_i}{\sqrt{\sum_{i=1}^d \xi_i^2}} \right) + \mathcal{O} \left( \frac{1}{\sqrt{d}} \right) \right)$$

$$= \mathcal{Q} \left( \frac{\sum_{i=1}^d \eta_i}{\sqrt{\sum_{i=1}^d \xi_i^2}} \right) + \mathcal{O} \left( \frac{1}{\sqrt{d}} \right)$$

$$= \mathcal{Q} \left( \frac{\sum_{i=1}^d \left( 2\mu_i^2 + \frac{1-\gamma}{\gamma} \cdot \frac{\sigma^2}{2N} \right)}{\sqrt{\sum_{i=1}^d \sigma^2 \left( \frac{1+\gamma}{\gamma} \cdot \frac{\sigma^2}{N} + \frac{1+\gamma^2}{2\gamma^2} \cdot \frac{\sigma^2}{N^2} + 4\mu_i^2 \left( 1 + \frac{1}{N} \right) \right)}} \right) + \mathcal{O} \left( \frac{1}{\sqrt{d}} \right)$$

$$= \mathcal{Q} \left( \frac{2 \|\boldsymbol{\mu}\|^2 + \frac{d}{2N} \cdot \frac{1-\gamma}{\gamma} \cdot \sigma^2}{\sqrt{\sigma^2 \cdot \left( \left( \frac{1+\gamma}{\gamma} \cdot \frac{\sigma^2}{N} + \frac{1+\gamma^2}{2\gamma^2} \cdot \frac{\sigma^2}{N^2} \right) d + 4 \left( 1 + \frac{1}{N} \right) \|\boldsymbol{\mu}\|^2 \right)}} \right) + \mathcal{O} \left( \frac{1}{\sqrt{d}} \right)$$

$$= \mathcal{Q} \left( \frac{\|\boldsymbol{\mu}\| + \frac{d}{4N} \cdot \frac{1-\gamma}{\gamma} \cdot \frac{\sigma^2}{\|\boldsymbol{\mu}\|}}{\sigma \cdot \sqrt{\left( \frac{1}{4N} \cdot \frac{1+\gamma}{\gamma} \cdot \left( \frac{\sigma}{\|\boldsymbol{\mu}\|} \right)^2 + \frac{1}{8N^2} \cdot \frac{1+\gamma^2}{\gamma^2} \cdot \left( \frac{\sigma}{\|\boldsymbol{\mu}\|} \right)^2 \right) \cdot d + \left( 1 + \frac{1}{N} \right)}} \right) + \mathcal{O} \left( \frac{1}{\sqrt{d}} \right)$$

$$= \mathcal{Q} \left( \frac{\frac{\|\boldsymbol{\mu}\|}{\sigma} + \frac{d}{4N} \cdot \frac{1-\gamma}{\gamma} \cdot \frac{\sigma}{\|\boldsymbol{\mu}\|}}{\sqrt{\left( \frac{1}{4N} \cdot \frac{1+\gamma}{\gamma} \cdot \left( \frac{\sigma}{\|\boldsymbol{\mu}\|} \right)^2 + \frac{1}{8N^2} \cdot \frac{1+\gamma^2}{\gamma^2} \cdot \left( \frac{\sigma}{\|\boldsymbol{\mu}\|} \right)^2 \right) \cdot d + \left( 1 + \frac{1}{N} \right)}} \right) + \mathcal{O} \left( \frac{1}{\sqrt{d}} \right)$$

$$(70)$$

where we used the identity $\mathcal{Q}(-x) = 1 - \mathcal{Q}(x)$. We now revisit Eq. 5:

$$\mathcal{S} = \left( \frac{\|\boldsymbol{\mu}\|}{\sigma} \right)^2.$$

Thus, from Eq. 70, $q(2, 1)$ reads:

$$q(2,1) = \mathcal{Q}\left(\frac{\sqrt{\mathcal{S}} + \frac{1}{4N} \cdot \frac{1-\gamma}{\gamma} \cdot \frac{d}{\sqrt{\mathcal{S}}}}{\sqrt{\frac{1}{4N} \cdot \frac{1+\gamma}{\gamma} \cdot \frac{d}{\mathcal{S}} + \frac{1}{8N^2} \cdot \frac{1+\gamma^2}{\gamma^2} \cdot \frac{d}{\mathcal{S}} + \frac{1}{N} + 1}}\right) + \mathcal{O}\left(\frac{1}{\sqrt{d}}\right). \tag{71}$$

To finish the proof, from Eq. 22, Eq. 62, Eq. 71, the probability of error reads:

$$p_{\boldsymbol{x}}(\text{error}) = \frac{1}{2} \cdot q(1,2) + \frac{1}{2} \cdot q(2,1)$$

$$= \frac{1}{2} \cdot \left(\mathcal{Q}\left(\frac{\sqrt{\mathcal{S}} + \frac{1}{4N} \cdot \frac{1-\gamma}{\gamma} \cdot \frac{d}{\sqrt{\mathcal{S}}}}{\sqrt{\frac{1}{4N} \cdot \frac{1+\gamma}{\gamma} \cdot \frac{d}{\mathcal{S}} + \frac{1}{8N^2} \cdot \frac{1+\gamma^2}{\gamma^2} \cdot \frac{d}{\mathcal{S}} + \frac{1}{\gamma N} + 1}}\right) + \mathcal{O}\left(\frac{1}{\sqrt{d}}\right)\right)$$

$$+ \frac{1}{2} \cdot \left(\mathcal{Q}\left(\frac{\sqrt{\mathcal{S}} - \frac{1}{4N} \cdot \frac{1-\gamma}{\gamma} \cdot \frac{d}{\sqrt{\mathcal{S}}}}{\sqrt{\frac{1}{4N} \cdot \frac{1+\gamma}{\gamma} \cdot \frac{d}{\mathcal{S}} + \frac{1}{8N^2} \cdot \frac{1+\gamma^2}{\gamma^2} \cdot \frac{d}{\mathcal{S}} + \frac{1}{N} + 1}}\right) + \mathcal{O}\left(\frac{1}{\sqrt{d}}\right)\right)$$

$$= \frac{1}{2} \cdot \mathcal{Q}\left(\frac{\sqrt{\mathcal{S}} + \frac{1}{4N} \cdot \frac{1-\gamma}{\gamma} \cdot \frac{d}{\sqrt{\mathcal{S}}}}{\sqrt{\frac{1}{4N} \cdot \frac{1+\gamma}{\gamma} \cdot \frac{d}{\mathcal{S}} + \frac{1}{8N^2} \cdot \frac{1+\gamma^2}{\gamma^2} \cdot \frac{d}{\mathcal{S}} + \frac{1}{\gamma N} + 1}}\right)$$

$$+ \frac{1}{2} \cdot \mathcal{Q}\left(\frac{\sqrt{\mathcal{S}} - \frac{1}{4N} \cdot \frac{1-\gamma}{\gamma} \cdot \frac{d}{\sqrt{\mathcal{S}}}}{\sqrt{\frac{1}{4N} \cdot \frac{1+\gamma}{\gamma} \cdot \frac{d}{\mathcal{S}} + \frac{1}{8N^2} \cdot \frac{1+\gamma^2}{\gamma^2} \cdot \frac{d}{\mathcal{S}} + \frac{1}{N} + 1}}\right) + \mathcal{O}\left(\frac{1}{\sqrt{d}}\right)$$

$$= \hat{p}(\mathcal{S}, N, \gamma, d) + \mathcal{O}\left(\frac{1}{\sqrt{d}}\right)$$

where $\hat{p}$ is given in Eq. 9. $\qquad\square$

### A.3 PROOF OF THEOREM 3

*Proof.* We provide an algorithm to construct $\boldsymbol{A} \in \mathbb{R}^{k \times d}$ given $\frac{\boldsymbol{\mu}}{\|\boldsymbol{\mu}\|}$ and prove that it satisfies Eq. 7. Later, we will show how to estimate it from unlabeled data.

1. Define $\boldsymbol{u} := \boldsymbol{a}_1 = \frac{\boldsymbol{\mu}}{\|\boldsymbol{\mu}\|}$. If $k = 1$, define $\boldsymbol{A} = \boldsymbol{u}^\top$. Else, continue.

2. Find $\boldsymbol{a}_2, \ldots, \boldsymbol{a}_k \in \mathbb{R}^d$ such that $\langle \boldsymbol{a}_i, \boldsymbol{u} \rangle = 0$ and $\langle \boldsymbol{a}_i, \boldsymbol{a}_j \rangle = \delta_{ij}$.

3. Define the matrix $\boldsymbol{A} \in \mathbb{R}^{k \times d}$ where the $i$-th row is given by $\boldsymbol{a}_i^\top$.

The proof that the algorithm works is given below.

- Step 1: If $k = 1$, we define $\boldsymbol{A} = \frac{\boldsymbol{\mu}^\top}{\|\boldsymbol{\mu}\|}$. It is easy to ensure that it satisfies Eq. 7.

- Step 2: If $\boldsymbol{\mu} = \boldsymbol{0}$, then the result is trivial, because we can construct on orthonormal set

$$\{\boldsymbol{a}_2, \ldots, \boldsymbol{a_k}\} \subset \mathbb{R}^d.$$

Otherwise, $\boldsymbol{\mu} \neq \boldsymbol{0}$ and let us define the following subset of $\mathbb{R}^d$:

$$V := \{\boldsymbol{x} \in \mathbb{R}^d : \langle \boldsymbol{x}, \boldsymbol{\mu} \rangle = 0\} \subset \mathbb{R}^d.$$

We see that $V = (\text{span}\{\boldsymbol{\mu}\})^\perp$ is a linear subspace of $\mathbb{R}^d$ of dimension $d - 1$. Thus, there exists a basis $\{\boldsymbol{v}_1, \ldots, \boldsymbol{v}_{d-1}\} \subseteq V$. We know that $k - 1 \leq d - 1$ and thus $\{\boldsymbol{v}_1, \ldots, \boldsymbol{v}_{k-1}\} \subseteq V$ is a linearly independent set. That is, we can apply the Gram-Schmidt procedure on this set, to get an orthonormal set $\{\boldsymbol{a}_2, \ldots, \boldsymbol{a}_k\} \subseteq V$. This is a subset of $V$ because Gram–Schmidt outputs vectors that are linear combinations of the input, which lie in $V$.

- Step 3: The rows of $\boldsymbol{A}$ are orthonormal, so $\boldsymbol{A}\boldsymbol{A}^\top = \boldsymbol{I}_k$. From step 2, it follows easily that

$$
\boldsymbol{A}\boldsymbol{\mu} = \begin{bmatrix} \|\boldsymbol{\mu}\| \\ 0 \\ \vdots \\ 0 \end{bmatrix} \Rightarrow \|\boldsymbol{A}\boldsymbol{\mu}\| = \|\boldsymbol{\mu}\|
$$

Thus, $\boldsymbol{A}$ meets the needed requirements, and thus we have proved the existence of such a matrix $\boldsymbol{A}$. Assuming $\boldsymbol{\mu} \neq \boldsymbol{0}$, it is now left to prove that one can learn $\boldsymbol{A}$ from infinite unlabeled data $\{\boldsymbol{x}_i\}_{i=1}^\infty$. This data is taken from the distribution

$$
\boldsymbol{x} \sim \frac{1}{2}\mathcal{N}\left(-\boldsymbol{\mu}, \sigma^2 \boldsymbol{I}_d\right) + \frac{1}{2}\mathcal{N}\left(\boldsymbol{\mu}, \sigma^2 \boldsymbol{I}_d\right) \tag{72}
$$

where the label is called $y \in \{1, 2\}$. Let us assume that there is $m$ unlabeled data. We first compute

$$
\boldsymbol{\Sigma}_m = \frac{1}{m}\sum_{i=1}^m \boldsymbol{x}_i \boldsymbol{x}_i^\top.
$$

As $m \to \infty$, we have $\boldsymbol{\Sigma}_m \xrightarrow{\text{a.s.}} \boldsymbol{\Sigma}$, where

$$
\begin{aligned}
\boldsymbol{\Sigma} = \mathbb{E}\left[\boldsymbol{x}\boldsymbol{x}^\top\right] &= \mathbb{E}\left[\boldsymbol{x}\boldsymbol{x}^\top \mid y = 1\right] \cdot \mathbb{P}\left(y = 1\right) + \mathbb{E}\left[\boldsymbol{x}\boldsymbol{x}^\top \mid y = 2\right] \cdot \mathbb{P}\left(y = 2\right) \\
&= \frac{1}{2}\left(\sigma^2 \boldsymbol{I}_d + \boldsymbol{\mu}\boldsymbol{\mu}^\top\right) + \frac{1}{2}\left(\sigma^2 \boldsymbol{I}_d + \boldsymbol{\mu}\boldsymbol{\mu}^\top\right) \\
&= \sigma^2 \boldsymbol{I}_d + \boldsymbol{\mu}\boldsymbol{\mu}^\top
\end{aligned}
$$

where we used Eq. 72. That is, we can learn the matrix

$$
\boldsymbol{\Sigma} = \sigma^2 \boldsymbol{I}_d + \boldsymbol{\mu}\boldsymbol{\mu}^\top. \tag{73}
$$

We now argue that the maximal eigenvalue of $\boldsymbol{\Sigma}$ is $\lambda_{\max} = \sigma^2 + \|\boldsymbol{\mu}\|^2$, with eigen-space $V_{\lambda_{\max}} = \text{span}\{\boldsymbol{\mu}\}$. Indeed, from Eq. 73, it follows that:

$$
\boldsymbol{\Sigma}\boldsymbol{\mu} = \left(\sigma^2 + \|\boldsymbol{\mu}\|^2\right)\boldsymbol{\mu}
$$

and for all $\boldsymbol{v} \perp \boldsymbol{\mu}$ we have

$$
\boldsymbol{\Sigma}\boldsymbol{v} = \sigma^2 \boldsymbol{v}.
$$

Thus, the eigenvalues of $\boldsymbol{\Sigma}$ are

$$
\sigma^2 = \lambda_{\min} < \lambda_{\max} = \sigma^2 + \|\boldsymbol{\mu}\|^2.
$$

The eigen-space of $\lambda_{\min}$ satisfies:

$$
V_{\lambda_{\min}} = \left(\text{span}\{\boldsymbol{\mu}\}\right)^\perp \Rightarrow \dim\left(V_{\lambda_{\min}}\right) = d - 1
$$

Thus, $\dim\left(V_{\lambda_{\max}}\right) = 1$, which implies that

$$
V_{\lambda_{\max}} = \text{span}\{\boldsymbol{\mu}\}. \tag{74}
$$

We now apply the power iteration method on the matrix $\boldsymbol{\Sigma}_m$. For large enough number of iterations and sufficiently large $m \gg 1$, it returns a vector that is *arbitrarily close* to the eigenvector of $\boldsymbol{\Sigma}$ that corresponds to the maximal eigenvalue $\lambda_{\max}$ (ensured by the spectral gap of $\|\boldsymbol{\mu}\|^2 > 0$ between the two largest eigenvalues of $\boldsymbol{\Sigma}$), which from Eq. 74, is characterized as $\alpha\boldsymbol{\mu}$ where $\alpha \neq 0$ is a constant. Normalizing this vector leads to $\pm\dfrac{\boldsymbol{\mu}}{\|\boldsymbol{\mu}\|}$. Now, we apply the algorithm we presented above to compute $\boldsymbol{A}$. As a side note, using the vector $\boldsymbol{a}_1 = -\dfrac{\boldsymbol{\mu}^\top}{\|\boldsymbol{\mu}\|}$ as the first row of $\boldsymbol{A}$ has no effect on the resulting properties of $\boldsymbol{A}$. $\qquad\square$

### A.4 PROOF OF THEOREM 4

*Proof.* We know that

$$z = Ax,$$

where $A \in \mathbb{R}^{k \times d}$ is a deterministic matrix satisfying:

- $AA^\top = I_k$.

- $\|A\mu\| = \|\mu\|$.

It is a standard result that a linear transformation of a Gaussian vector is also a Gaussian vector, thus:

$$\forall_{j \in \{1,2\}} \ z \mid y = j \sim \mathcal{N}\left(A\mu_j, A\sigma^2 I_d A^\top\right)$$

that is, for all $j \in \{1, 2\}$ we have:

$$z \mid y = j \sim \mathcal{N}\left(\eta_j, \sigma^2 I_k\right)$$

where

$$\eta_j = A\mu_j.$$

We know that $\mu_2 = -\mu_1 = \mu$, and thus $\eta_2 = -\eta_1 = \eta = A\mu$. That is, our model assumptions still hold, with the following modifications:

- $d \mapsto k$.

- $\mu \mapsto \eta = A\mu$.

The new separation quality factor $\mathcal{S}_z$ of the new GMM (computed similarly to Eq. 5) is given by:

$$\mathcal{S}_z = \left(\frac{\|\eta_2 - \eta_1\|}{2\sigma}\right)^2 = \left(\frac{\|\eta\|}{\sigma}\right)^2 = \left(\frac{\|A\mu\|}{\sigma}\right)^2 = \left(\frac{\|\mu\|}{\sigma}\right)^2 = \mathcal{S}.$$

That is, the separation quality factor remains the same after the processing. The result is now immediate from Theorem 2 and changing $d \mapsto k$. □

### A.5 PROOF OF THEOREM 5

*Proof.* Let us fix $\gamma = 1$ and take some

$$\mathcal{S} > 0, \ 1 \le k < d, \ N \in \mathbb{N}.$$

From Theorems 2 and 4, it follows that we need to show the following:

$$\hat{p}(\mathcal{S}, N, 1, k) < \hat{p}(\mathcal{S}, N, 1, d) \tag{75}$$

where $\hat{p}$ is given in Eq. 9. It is easy to prove that:

$$\forall_{q \in \mathbb{N}} \ \hat{p}\left(\mathcal{S}, N, 1, q\right) = \mathcal{Q}\left(\frac{\sqrt{\mathcal{S}}}{\sqrt{\left(\frac{q}{2\mathcal{S}} + 1\right) \cdot \frac{1}{N} + \frac{q}{4\mathcal{S}} \cdot \frac{1}{N^2} + 1}}\right).$$

Following Eq. 75, we need to show that:

$$\mathcal{Q}\left(\frac{\sqrt{\mathcal{S}}}{\sqrt{\left(\frac{k}{2\mathcal{S}} + 1\right) \cdot \frac{1}{N} + \frac{k}{4\mathcal{S}} \cdot \frac{1}{N^2} + 1}}\right) < \mathcal{Q}\left(\frac{\sqrt{\mathcal{S}}}{\sqrt{\left(\frac{d}{2\mathcal{S}} + 1\right) \cdot \frac{1}{N} + \frac{d}{4\mathcal{S}} \cdot \frac{1}{N^2} + 1}}\right)$$

which is immediate because the argument in the $\mathcal{Q}$ is strictly higher in the LHS, and the $\mathcal{Q}$ function is strictly decreasing. □

### A.6  PROOF OF THEOREM 6

*Proof.* Let us take some

$$0 < \gamma < 1,\ 0 < \mathcal{S} \le 1, 1 \le k < d, N \ge \frac{\gamma^2 - 4\gamma + 1}{2\gamma \cdot (1+\gamma)}$$

we need to show that

$$\hat{p}(\mathcal{S}, N, \gamma, k) < \hat{p}(\mathcal{S}, N, \gamma, d).$$

That is, it is sufficient to show that the function

$$f(x) = 2\hat{p}(\mathcal{S}, N, \gamma, x)$$

is strictly increasing for all $x \ge 1$, where $\hat{p}$ is defined in Eq. 9. It is easy to verify that:

$$f(x) = \mathcal{Q}\left( \frac{\sqrt{\mathcal{S}} + \frac{1-\gamma}{4\gamma N \sqrt{\mathcal{S}}} \cdot x}{\sqrt{\left( \frac{1+\gamma}{4\gamma N \mathcal{S}} + \frac{1+\gamma^2}{8\gamma^2 N^2 \mathcal{S}} \right) \cdot x + \frac{1}{\gamma N} + 1}} \right) + \mathcal{Q}\left( \frac{\sqrt{\mathcal{S}} - \frac{1-\gamma}{4\gamma N \sqrt{\mathcal{S}}} \cdot x}{\sqrt{\left( \frac{1+\gamma}{4\gamma N \mathcal{S}} + \frac{1+\gamma^2}{8\gamma^2 N^2 \mathcal{S}} \right) \cdot x + \frac{1}{N} + 1}} \right). \tag{76}$$

Let us define the following functions:

$$g_1(x) = \frac{\sqrt{\mathcal{S}} + \frac{1-\gamma}{4\gamma N \sqrt{\mathcal{S}}} \cdot x}{\sqrt{\left( \frac{1+\gamma}{4\gamma N \mathcal{S}} + \frac{1+\gamma^2}{8\gamma^2 N^2 \mathcal{S}} \right) \cdot x + \frac{1}{\gamma N} + 1}} \tag{77}$$

and

$$g_2(x) = \frac{\sqrt{\mathcal{S}} - \frac{1-\gamma}{4\gamma N \sqrt{\mathcal{S}}} \cdot x}{\sqrt{\left( \frac{1+\gamma}{4\gamma N \mathcal{S}} + \frac{1+\gamma^2}{8\gamma^2 N^2 \mathcal{S}} \right) \cdot x + \frac{1}{N} + 1}}. \tag{78}$$

Thus, Eq. 76 reads:

$$f(x) = \mathcal{Q}\left( g_1(x) \right) + \mathcal{Q}\left( g_2(x) \right). \tag{79}$$

From the chain rule, the derivative reads:

$$\begin{aligned} f'(x) &= \mathcal{Q}'\left( g_1(x) \right) \cdot g_1'(x) + \mathcal{Q}'\left( g_2(x) \right) \cdot g_2'(x) \\ &= -\frac{1}{\sqrt{2\pi}} \cdot \left( \exp\left( -\frac{1}{2} \cdot g_1^2(x) \right) \cdot g_1'(x) + \exp\left( -\frac{1}{2} \cdot g_2^2(x) \right) \cdot g_2'(x) \right) \\ &= -\frac{1}{\sqrt{2\pi}} \cdot \left( w_1(x) \cdot g_1'(x) + w_2(x) \cdot g_2'(x) \right). \end{aligned} \tag{80}$$

We used the following property of the $\mathcal{Q}$ function:

$$\frac{d}{dx}\mathcal{Q}(x) = -\frac{1}{\sqrt{2\pi}} \cdot \exp\left( -\frac{x^2}{2} \right)$$

and the following notation:

$$w_i(x) = \exp\left( -\frac{1}{2} \cdot g_i^2(x) \right). \tag{81}$$

Thus, showing that $f$ is strictly increasing for all $x \ge 1$ is equivalent to proving that for all $x \ge 1$

$$f'(x) > 0 \Leftrightarrow w_1(x) \cdot g_1'(x) + w_2(x) \cdot g_2'(x) < 0. \tag{82}$$

We argue now that for all $x \ge 1$:

1.

$$w_1(x) < w_2(x) \tag{83}$$

2.

$$g_2'(x) < 0 \tag{84}$$

3.

$$g_1'(x) + g_2'(x) \le 0 \tag{85}$$

Let us first prove Eq. 83: From Eq. 81 it follows that it is sufficient to prove:

$$\forall_{x \ge 1} \ |g_1(x)| > |g_2(x)| . \tag{86}$$

Let us take some $x \ge 1$. It is easy to see from Eq. 77 that $g_1(x) \ge 0$ and thus $|g_1(x)| = g_1(x)$. That is, it is sufficient to prove that

$$g_1(x) > g_2(x) \tag{87}$$

and

$$g_2(x) > -g_1(x) \Leftrightarrow g_1(x) + g_2(x) > 0. \tag{88}$$

Let us now define the following parameters:

$$\begin{cases} B = \dfrac{1-\gamma}{4\gamma N\sqrt{\mathcal{S}}} \\ C = \dfrac{1+\gamma}{4\gamma N\mathcal{S}} + \dfrac{1+\gamma^2}{8\gamma^2 N^2 \mathcal{S}} \\ c_1 = \dfrac{1}{\gamma N} + 1 \\ c_2 = \dfrac{1}{N} + 1 < c_1 \end{cases} \tag{89}$$

We also define the following functions:

$$\begin{cases} D_1(x) = \sqrt{Cx + c_1} \\ D_2(x) = \sqrt{Cx + c_2} < D_1(x) \end{cases} \tag{90}$$

From Eq. 77, Eq. 78, Eq. 89, Eq. 90, it follows that:

$$\begin{cases} g_1(x) = \dfrac{\sqrt{\mathcal{S}} + Bx}{D_1(x)} \\ g_2(x) = \dfrac{\sqrt{\mathcal{S}} - Bx}{D_2(x)} \end{cases} \tag{91}$$

We first prove Eq. 87. Their difference $g_1(x) - g_2(x)$ reads:

$$\begin{aligned} g_1(x) - g_2(x) &= \frac{\sqrt{\mathcal{S}} + Bx}{D_1(x)} - \frac{\sqrt{\mathcal{S}} - Bx}{D_2(x)} = \frac{(\sqrt{\mathcal{S}} + Bx) \cdot D_2(x) - (\sqrt{\mathcal{S}} - Bx) \cdot D_1(x)}{D_1(x) \cdot D_2(x)} \\ &= \frac{\sqrt{\mathcal{S}} \cdot (D_2(x) - D_1(x)) + Bx \cdot (D_2(x) + D_1(x))}{D_1(x) \cdot D_2(x)} . \end{aligned} \tag{92}$$

Now, from Eq. 89, Eq. 90, we have:

$$\begin{aligned} D_2(x) - D_1(x) &= \frac{D_2^2(x) - D_1^2(x)}{D_2(x) + D_1(x)} = \frac{c_2 - c_1}{D_2(x) + D_1(x)} \\ &= \frac{\frac{1}{N} - \frac{1}{\gamma \cdot N}}{D_1(x) + D_2(x)} \\ &= -\frac{1}{N} \cdot \frac{1-\gamma}{\gamma} \cdot \frac{1}{D_2(x) + D_1(x)} . \end{aligned} \tag{93}$$

In order to show Eq. 87, it is sufficient to show that the expression in Eq. 92 is strictly positive. Substituting Eq. 93, we get:

$$\underbrace{-\frac{\sqrt{\mathcal{S}}}{N} \cdot \frac{1-\gamma}{\gamma} \cdot \frac{1}{D_2(x) + D_1(x)}}_{\sqrt{\mathcal{S}} \cdot (D_2(x) - D_1(x))} + \frac{1-\gamma}{4\gamma \cdot N \cdot \sqrt{\mathcal{S}}} \cdot x \cdot (D_2(x) + D_1(x)) > 0$$

$$\frac{1}{4\sqrt{\mathcal{S}}} \cdot x \cdot (D_2(x) + D_1(x)) > \frac{\sqrt{\mathcal{S}}}{D_2(x) + D_1(x)}$$

$$x \cdot (D_2(x) + D_1(x))^2 > 4\mathcal{S}$$

That is, in order to show Eq. 83, it is sufficient to show that:

$$h(x) := x \cdot (D_2(x) + D_1(x))^2 > 4\mathcal{S}. \tag{94}$$

We now show that $h(x)$ is strictly increasing:

$$h'(x) = (D_2(x) + D_1(x))^2 + 2x \cdot (D_2(x) + D_1(x)) > 0.$$

Thus, it follows that:

$$x \geq 1 \Rightarrow h(x) > h(1). \tag{95}$$

Now, from Eq. 94, it follows that:

$$h(1) = (D_2(1) + D_1(1))^2 > 4\mathcal{S} \Leftrightarrow D_2(1) + D_1(1) > 2\sqrt{\mathcal{S}}.$$

Indeed, from Eq. 89, Eq. 90, we have:

$$D_2(1) + D_1(1) > 2 \cdot D_2(1) = 2 \cdot \sqrt{C + c_2}$$
$$= 2 \cdot \sqrt{C + \frac{1}{N} + 1}$$
$$> 2$$
$$\geq 2\sqrt{\mathcal{S}}$$

where we used the assumption that $\mathcal{S} \leq 1$, and $C > 0$. That is, we proved Eq. 87. We will now prove Eq. 88. From Eq. 91, the sum $g_1(x) + g_2(x)$ reads:

$$g_1(x) + g_2(x) = \frac{\sqrt{\mathcal{S}} + B \cdot x}{D_1(x)} + \frac{\sqrt{\mathcal{S}} - Bx}{D_2(x)} = \frac{\left(\sqrt{\mathcal{S}} + Bx\right) \cdot D_2(x) + \left(\sqrt{\mathcal{S}} - Bx\right) \cdot D_1(x)}{D_1(x) \cdot D_2(x)}$$
$$= \frac{\sqrt{\mathcal{S}} \cdot (D_2(x) + D_1(x)) + Bx \cdot (D_2(x) - D_1(x))}{D_1(x) \cdot D_2(x)}. \tag{96}$$

In order to show Eq. 88, it is sufficient to show that the expression in Eq. 96 is strictly positive. Substituting Eq. 93, we get:

$$\sqrt{\mathcal{S}} \cdot (D_2(x) + D_1(x)) + \underbrace{\left(-\frac{1 - \gamma}{4\gamma \cdot N \cdot \sqrt{\mathcal{S}}} \cdot \frac{1}{N} \cdot \frac{1 - \gamma}{\gamma} \cdot \frac{x}{D_2(x) + D_1(x)}\right)}_{Bx \cdot (D_2(x) - D_1(x))} > 0$$

$$\sqrt{\mathcal{S}} \cdot (D_2(x) + D_1(x)) > \frac{(1 - \gamma)^2}{4\gamma^2 N^2 \sqrt{\mathcal{S}}} \cdot \frac{x}{D_2(x) + D_1(x)}$$

$$\frac{(D_2(x) + D_1(x))^2}{x} > \frac{(1 - \gamma)^2}{4\gamma^2 N^2 \mathcal{S}}$$

That is, in order to show Eq. 88, it is sufficient to show that:

$$p(x) := \frac{(D_2(x) + D_1(x))^2}{x} > \frac{(1 - \gamma)^2}{4\gamma^2 N^2 \mathcal{S}}. \tag{97}$$

Indeed,

$$p(x) = \frac{D_2^2(x) + 2D_2(x)D_1(x) + D_1^2(x)}{x} \geq \frac{D_2^2(x) + D_1^2(x)}{x}$$
$$= \frac{Cx + c_1 + Cx + c_2}{x}$$
$$= 2C + \frac{c_1 + c_2}{x}$$
$$> 2C \tag{98}$$
$$= \frac{1 + \gamma}{2\gamma N \mathcal{S}} + \frac{1 + \gamma^2}{4\gamma^2 N^2 \mathcal{S}}$$
$$> \frac{1 + \gamma^2}{4\gamma^2 N^2 \mathcal{S}}$$
$$> \frac{(1 - \gamma)^2}{4\gamma^2 N^2 \mathcal{S}}$$

where we used $c_1, c_2 > 0$ and $(1 - \gamma)^2 < 1 + \gamma^2$ for all $\gamma > 0$. That is, we proved Eq. 88 and thus we showed that Eq. 83 is satisfied. We will now prove Eq. 84: Let us first define the following parametric function

$$T_{B,C,D}(x) = \frac{\sqrt{\mathcal{S}} + Bx}{\sqrt{Cx + D}}. \tag{99}$$

Its derivative reads:

$$
\begin{aligned}
T'_{B,C,D}(x) &= \frac{B \cdot \sqrt{Cx + D} - \frac{C}{2 \cdot \sqrt{Cx+D}} \cdot \left(\sqrt{\mathcal{S}} + Bx\right)}{Cx + D} \\
&= \frac{2B \cdot (Cx + D) - C \cdot (\sqrt{\mathcal{S}} + Bx)}{2 \cdot (Cx + D)^{1.5}} \\
&= \frac{BC \cdot x + 2 \cdot BD - \sqrt{\mathcal{S}}C}{2 \cdot (Cx + D)^{1.5}}.
\end{aligned} \tag{100}
$$

Now, from Eq. 78, Eq. 99, it follows that:

$$g_2(x) = T_{-B,C,c_2}(x).$$

That is, from Eq. 100, we have:

$$g'_2(x) = \frac{-BC \cdot x - 2B \cdot c_2 - \sqrt{\mathcal{S}} \cdot C}{2 \cdot (Cx + c_2)^{1.5}} = -\frac{BC \cdot x + 2B \cdot c_2 + \sqrt{\mathcal{S}} \cdot C}{2 \cdot (Cx + c_2)^{1.5}} < 0 \tag{101}$$

where we used $B, C, c_2 > 0$, which follows from Eq. 89, and $\mathcal{S} > 0$. Finally, we will prove Eq. 85: From Eq. 77, Eq. 99, Eq. 100, it follows that:

$$g_1(x) = T_{B,C,c_1}(x) \Rightarrow g'_1(x) = \frac{BC \cdot x + 2B \cdot c_1 - \sqrt{\mathcal{S}} \cdot C}{2 \cdot (Cx + c_1)^{1.5}}. \tag{102}$$

Thus, from Eq. 101, Eq. 102, proving that $g'_1(x) + g'_2(x) \leq 0$ is equivalent to proving that:

$$\frac{BC \cdot x + 2B \cdot c_1 - \sqrt{\mathcal{S}} \cdot C}{2 \cdot (Cx + c_1)^{1.5}} \leq \frac{BC \cdot x + 2B \cdot c_2 + \sqrt{\mathcal{S}} \cdot C}{2 \cdot (Cx + c_2)^{1.5}}.$$

From Eq. 89, and the assumption of $0 < \gamma < 1$, we know that $c_1 > c_2$. Thus, if the numerator in the LHS is negative, then the inequality holds trivially. Otherwise, it is sufficient to prove that:

$$2B \cdot c_1 - \sqrt{\mathcal{S}} \cdot C \leq 2B \cdot c_2 + \sqrt{\mathcal{S}} \cdot C$$

$$2B \cdot (c_1 - c_2) \leq 2\sqrt{\mathcal{S}} \cdot C$$

$$B \cdot \left(\frac{1}{\gamma \cdot N} - \frac{1}{N}\right) \leq \sqrt{\mathcal{S}} \cdot C$$

Thus, we need to prove that:

$$\frac{1}{N} \cdot \frac{1 - \gamma}{\gamma} \leq \frac{\sqrt{\mathcal{S}} \cdot C}{B}. \tag{103}$$

From Eq. 89, the RHS in Eq. 103 reads:

$$
\begin{aligned}
\frac{\sqrt{\mathcal{S}} \cdot C}{B} &= \frac{\left(\frac{1+\gamma}{4 \cdot \gamma \cdot N \cdot \sqrt{\mathcal{S}}} + \frac{1+\gamma^2}{8 \cdot \gamma^2 \cdot N^2 \cdot \sqrt{\mathcal{S}}}\right)}{\left(\frac{1-\gamma}{4 \cdot \gamma \cdot N \cdot \sqrt{\mathcal{S}}}\right)} = \frac{\left(\frac{1+\gamma}{4} + \frac{1+\gamma^2}{8 \cdot \gamma \cdot N}\right)}{\left(\frac{1-\gamma}{4}\right)} \\
&= \left(\frac{1+\gamma}{4} + \frac{1+\gamma^2}{8 \cdot \gamma \cdot N}\right) \cdot \frac{4}{1-\gamma} \\
&= \frac{1+\gamma}{1-\gamma} + \frac{1+\gamma^2}{\gamma \cdot (1-\gamma)} \cdot \frac{1}{2 \cdot N} \\
&= \frac{2N \cdot \gamma(1+\gamma) + 1 + \gamma^2}{2N \cdot \gamma(1-\gamma)} \\
&= \frac{(2N+1) \cdot \gamma^2 + 2N \cdot \gamma + 1}{2N \cdot \gamma(1-\gamma)}.
\end{aligned}
$$

Thus, Eq. 103 reads:

$$\frac{1-\gamma}{\gamma \cdot N} \leq \frac{(2N+1) \cdot \gamma^2 + 2N \cdot \gamma + 1}{2N \cdot \gamma(1-\gamma)}$$
$$2 \cdot (1-\gamma)^2 \leq (2N+1) \cdot \gamma^2 + 2N \cdot \gamma + 1$$
$$2 \cdot (\gamma^2 - 2\gamma + 1) \leq (2N+1) \cdot \gamma^2 + 2N \cdot \gamma + 1$$
$$(2N-1) \cdot \gamma^2 + 2 \cdot (N+2) \cdot \gamma - 1 \geq 0$$
$$2 \cdot \gamma \cdot (\gamma + 1) \cdot N - (\gamma^2 - 4\gamma + 1) \geq 0$$
$$N \geq \frac{\gamma^2 - 4\gamma + 1}{2\gamma(1+\gamma)}$$

Which holds from the theorem assumptions. Finally, let us take some $x \geq 1$. We need to prove that:

$$w_1(x) \cdot g_1'(x) + w_2(x) \cdot g_2'(x) < 0 \Leftrightarrow w_1(x) \cdot g_1'(x) < -w_2(x) \cdot g_2'(x)$$
$$\Leftrightarrow g_1'(x) < -\frac{w_2(x)}{w_1(x)} \cdot g_2'(x).$$

Indeed, from Eq. 83, Eq. 101, Eq. 85, it follows that:

$$g_1'(x) \leq -g_2'(x) < \frac{w_2(x)}{w_1(x)} \cdot (-g_2'(x)) = -\frac{w_2(x)}{w_1(x)} \cdot g_2'(x).$$

Which finishes the proof. Note that for $\gamma \geq 0.162$, the requirement $N \geq \frac{\gamma^2 - 4\gamma + 1}{2\gamma(1+\gamma)}$ is vacuous (since $N \geq 1$), so it only matters under severe imbalance ($\gamma < 0.162$). $\qquad \square$

### A.7 PROOF OF THEOREM 7

In order to have a fair comparison between cases with different values of $\gamma$, we fix the total number of samples to be $N_T$. Thus, we take $N_1 = xN_T$ samples from the first class and $N_2 = \gamma \cdot xN_T$ from the second class such that:

$$N_1 + N_2 = xN_T + \gamma \cdot xN_T = N_T \Rightarrow x = \frac{1}{1+\gamma}$$

Meaning, the number of samples in the first class is

$$N = \frac{N_T}{1+\gamma}. \tag{104}$$

*Proof.* Let us take some $N_T \in \mathbb{N}$ and

$$\mathcal{S} > 0, 1 \leq k < d, 0 < \gamma \leq 1.$$

Let us define the following parametric function:

$$f_{s,a,q}(x) := \frac{\sqrt{\mathcal{S}} + s \cdot \frac{(1-\gamma) \cdot q}{4\gamma \cdot \sqrt{\mathcal{S}}} \cdot \frac{1}{x}}{\sqrt{\left(\frac{(1+\gamma) \cdot q}{4\gamma \cdot \mathcal{S}} + \frac{1}{a}\right) \cdot \frac{1}{x} + \frac{(1+\gamma^2) \cdot q}{8\gamma^2 \cdot \mathcal{S}} \cdot \frac{1}{x^2} + 1}} = \frac{\sqrt{\mathcal{S}} + \frac{B}{x}}{\sqrt{\frac{C}{x} + \frac{D}{x^2} + 1}} \tag{105}$$

where the parameters $B, C, D$ are:

$$\begin{cases} B = s \cdot \dfrac{(1-\gamma) \cdot q}{4\gamma \cdot \sqrt{\mathcal{S}}} \\ C = \dfrac{(1+\gamma) \cdot q}{4\gamma \cdot \mathcal{S}} + \dfrac{1}{a} \\ D = \dfrac{(1+\gamma^2) \cdot q}{8\gamma^2 \cdot \mathcal{S}} \end{cases} \tag{106}$$

From Definition 13 and Eq. 9, it follows that:

$$
\begin{aligned}
\eta &= 100 \cdot \left(1 - \frac{\hat{p}_{\boldsymbol{z}}(\text{error})}{\hat{p}_{\boldsymbol{x}}(\text{error})}\right) = 100 \cdot \left(1 - \frac{\hat{p}(\mathcal{S}, N, \gamma, k)}{\hat{p}(\mathcal{S}, N, \gamma, d)}\right) \\
&= 100 \cdot \left(1 - \frac{\mathcal{Q}\left(f_{1,\gamma,k}(N)\right) + \mathcal{Q}\left(f_{-1,1,k}(N)\right)}{\mathcal{Q}\left(f_{1,\gamma,d}(N)\right) + \mathcal{Q}\left(f_{-1,1,d}(N)\right)}\right) \\
&= 100 \cdot h(N)
\end{aligned}
\tag{107}
$$

where we defined the following function:

$$
h(x) := 1 - \frac{\mathcal{Q}\left(f_{1,\gamma,k}(x)\right) + \mathcal{Q}\left(f_{-1,1,k}(x)\right)}{\mathcal{Q}\left(f_{1,\gamma,d}(x)\right) + \mathcal{Q}\left(f_{-1,1,d}(x)\right)}.
\tag{108}
$$

Now, let us define the following parametric function:

$$
g_{s,a,q}(x) := f_{s,a,q}\left(\frac{1}{x}\right) = \frac{\sqrt{\mathcal{S}} + B \cdot x}{\sqrt{D \cdot x^2 + C \cdot x + 1}}
\tag{109}
$$

where we used Eq. 105, Eq. 106. Let us also define:

$$
\ell(x) := h\left(\frac{1}{x}\right) = 1 - \frac{\mathcal{Q}\left(g_{1,\gamma,k}(x)\right) + \mathcal{Q}\left(g_{-1,1,k}(x)\right)}{\mathcal{Q}\left(g_{1,\gamma,d}(x)\right) + \mathcal{Q}\left(g_{-1,1,d}(x)\right)}
\tag{110}
$$

where we used Eq. 109, Eq. 108. Thus, Taylor expansion to first order of $\ell$ yields:

$$
\ell(x) = \ell(0) + \ell'(0) \cdot x + \mathcal{O}\left(x^2\right)
$$

Where the approximation is exact for $x \ll 1$. Thus, the following is exact for $x \gg 1$:

$$
x \gg 1 \Rightarrow h(x) = \ell\left(\frac{1}{x}\right) = \ell(0) + \frac{\ell'(0)}{x} + \mathcal{O}\left(\frac{1}{x^2}\right).
\tag{111}
$$

Assuming:

$$
N = \frac{N_T}{1 + \gamma} \gg 1 \Leftrightarrow N_T \gg 1 + \gamma
\tag{112}
$$

means that following first-order approximation is exact:

$$
h(N) = \ell(0) + \frac{\ell'(0)}{N} + \mathcal{O}\left(\frac{1}{N^2}\right).
\tag{113}
$$

Let us first compute $\ell(0)$:

$$
\ell(0) = 1 - \frac{\mathcal{Q}\left(g_{1,\gamma,k}(0)\right) + \mathcal{Q}\left(g_{-1,1,k}(0)\right)}{\mathcal{Q}\left(g_{1,\gamma,d}(0)\right) + \mathcal{Q}\left(g_{-1,1,d}(0)\right)} = 1 - \frac{2 \cdot \mathcal{Q}(\sqrt{\mathcal{S}})}{2 \cdot \mathcal{Q}(\sqrt{\mathcal{S}})} = 0.
\tag{114}
$$

Finally, we will compute $\ell'(0)$: We first compute $g'_{s,a,q}(0)$. From Eq. 109 it follows that:

$$
\begin{aligned}
g'_{s,a,q}(x) &= \frac{B \cdot \sqrt{D \cdot x^2 + C \cdot x + 1} - \frac{2D \cdot x + C}{2 \cdot \sqrt{D \cdot x^2 + C \cdot x + 1}} \cdot \left(\sqrt{\mathcal{S}} + B \cdot x\right)}{D \cdot x^2 + C \cdot x + 1} \\
&= \frac{2B \cdot \left(D \cdot x^2 + C \cdot x + 1\right) - (2D \cdot x + C) \cdot \left(\sqrt{\mathcal{S}} + B \cdot x\right)}{2 \cdot \left(D \cdot x^2 + C \cdot x + 1\right)^{1.5}} \\
&= \frac{(BC - 2SD) \cdot x + \left(2B - C\sqrt{\mathcal{S}}\right)}{2 \cdot \left(D \cdot x^2 + C \cdot x + 1\right)^{1.5}}.
\end{aligned}
$$

Thus, the derivative at 0 is:

$$
\begin{aligned}
g'_{s,a,q}(0) &= \frac{2B - C\sqrt{\mathcal{S}}}{2} = B - \frac{1}{2} \cdot C\sqrt{\mathcal{S}} \\
&= \frac{s \cdot (1 - \gamma) \cdot q}{4\gamma \cdot S} - \frac{1}{2} \cdot \left(\frac{(1 + \gamma) \cdot q}{4\gamma \cdot S} + \frac{1}{a}\right) \cdot \sqrt{\mathcal{S}} \\
&= \frac{s \cdot (1 - \gamma) \cdot q}{4\gamma \cdot S} - \frac{(1 + \gamma) \cdot q}{8\gamma \cdot \sqrt{\mathcal{S}}} - \frac{\sqrt{\mathcal{S}}}{2a} \\
&= \frac{2s \cdot (1 - \gamma) \cdot q - (1 + \gamma) \cdot q}{8\gamma \cdot S} - \frac{\sqrt{\mathcal{S}}}{2a} \\
&= \frac{((2s - 1) - (2s + 1) \cdot \gamma) \cdot q}{8\gamma \cdot \sqrt{\mathcal{S}}} - \frac{\sqrt{\mathcal{S}}}{2a}.
\end{aligned}
\tag{115}
$$

Now, from Eq. 110, the derivative $\ell'(x)$ reads

$$
\begin{aligned}
\ell'(x) &= -\frac{d}{dx}\left(\frac{\mathcal{Q}\left(g_{1,\gamma,k}(x)\right) + \mathcal{Q}\left(g_{-1,1,k}(x)\right)}{\mathcal{Q}\left(g_{1,\gamma,d}(x)\right) + \mathcal{Q}\left(g_{-1,1,d}(x)\right)}\right) \\
&= -\frac{d}{dx}\left(\frac{u(x)}{v(x)}\right) \\
&= -\frac{u'(x) \cdot v(x) - v'(x) \cdot u(x)}{v^2(x)} \\
&= \frac{v'(x) \cdot u(x) - u'(x) \cdot v(x)}{v^2(x)}
\end{aligned}
\tag{116}
$$

where we defined the following auxiliary functions:

$$
\begin{cases}
u(x) = \mathcal{Q}\left(g_{1,\gamma,k}(x)\right) + \mathcal{Q}\left(g_{-1,1,k}(x)\right) \\
v(x) = \mathcal{Q}\left(g_{1,\gamma,d}(x)\right) + \mathcal{Q}\left(g_{-1,1,d}(x)\right)
\end{cases}
\tag{117}
$$

From the chain rule, their derivatives are:

$$
\begin{cases}
u'(x) = g'_{1,\gamma,k}(x) \cdot \mathcal{Q}'\left(g_{1,\gamma,k}(x)\right) + g'_{-1,1,k}(x) \cdot \mathcal{Q}'\left(g_{-1,1,k}(x)\right) \\
v'(x) = g'_{1,\gamma,d}(x) \cdot \mathcal{Q}'\left(g_{1,\gamma,d}(x)\right) + g'_{-1,1,d}(x) \cdot \mathcal{Q}'\left(g_{-1,1,d}(x)\right)
\end{cases}
\tag{118}
$$

Now, from Eq. 116, Eq. 118 it follows that:

$$
\ell'(0) = \frac{v'(0) \cdot u(0) - u'(0) \cdot v(0)}{v(0)^2}.
\tag{119}
$$

It is easy to verify from Eq. 109 that $g_{s,a,q}(0) = \sqrt{\mathcal{S}}$. Thus, from Eq. 117, Eq. 118, Eq. 115, we have the following formulas:

$$
\begin{cases}
u(0) = 2 \cdot \mathcal{Q}(\sqrt{\mathcal{S}}) \\
v(0) = 2 \cdot \mathcal{Q}(\sqrt{\mathcal{S}}) \\
u'(0) = \left(\frac{(1-3\gamma)\cdot k}{8\gamma \cdot \sqrt{\mathcal{S}}} - \frac{\sqrt{\mathcal{S}}}{2\gamma}\right) \cdot \mathcal{Q}'(\sqrt{\mathcal{S}}) + \left(\frac{-(3+\gamma)\cdot k}{8\gamma \cdot \sqrt{\mathcal{S}}} - \frac{\sqrt{\mathcal{S}}}{2}\right) \cdot \mathcal{Q}'(\sqrt{\mathcal{S}}) \\
v'(0) = \left(\frac{(1-3\gamma)\cdot d}{8\gamma \cdot \sqrt{\mathcal{S}}} - \frac{\sqrt{\mathcal{S}}}{2\gamma}\right) \cdot \mathcal{Q}'(\sqrt{\mathcal{S}}) + \left(\frac{-(3+\gamma)\cdot d}{8\gamma \cdot \sqrt{\mathcal{S}}} - \frac{\sqrt{\mathcal{S}}}{2}\right) \cdot \mathcal{Q}'(\sqrt{\mathcal{S}})
\end{cases}
\tag{120}
$$

Now, we substitute Eq. 120 in Eq. 119, to get the following formula for $\ell'(0)$:

$$
\begin{aligned}
\ell'(0) &= \frac{2 \cdot \mathcal{Q}(\sqrt{\mathcal{S}}) \cdot \left(\left(\frac{(1-3\gamma)\cdot d}{8\gamma \cdot \sqrt{\mathcal{S}}} - \frac{\sqrt{\mathcal{S}}}{2\gamma}\right) \cdot \mathcal{Q}'(\sqrt{\mathcal{S}}) + \left(\frac{-(3+\gamma)\cdot d}{8\gamma \cdot \sqrt{\mathcal{S}}} - \frac{\sqrt{\mathcal{S}}}{2}\right) \cdot \mathcal{Q}'(\sqrt{\mathcal{S}})\right)}{4 \cdot \mathcal{Q}^2(\sqrt{\mathcal{S}})} \\
&\quad - \frac{2\mathcal{Q}(\sqrt{\mathcal{S}}) \cdot \left(\left(\frac{(1-3\gamma)\cdot k}{8\gamma \cdot \sqrt{\mathcal{S}}} - \frac{\sqrt{\mathcal{S}}}{2\gamma}\right) \cdot \mathcal{Q}'(\sqrt{\mathcal{S}}) + \left(\frac{-(3+\gamma)\cdot k}{8\gamma \cdot \sqrt{\mathcal{S}}} - \frac{\sqrt{\mathcal{S}}}{2}\right) \cdot \mathcal{Q}'(\sqrt{\mathcal{S}})\right)}{4 \cdot \mathcal{Q}^2(\sqrt{\mathcal{S}})} \\
&= \frac{\mathcal{Q}'(\sqrt{\mathcal{S}})}{2 \cdot \mathcal{Q}(\sqrt{\mathcal{S}})} \cdot \left(\frac{(1-3\gamma)\cdot(d-k)}{8\gamma \cdot \sqrt{\mathcal{S}}} - \frac{(3+\gamma)\cdot(d-k)}{8\gamma \cdot \sqrt{\mathcal{S}}}\right) \\
&= \frac{\mathcal{Q}'(\sqrt{\mathcal{S}})}{2 \cdot \mathcal{Q}(\sqrt{\mathcal{S}})} \cdot (d-k) \cdot \left(\frac{-2-4\gamma}{8\gamma \cdot \sqrt{\mathcal{S}}}\right) \\
&= -\frac{\mathcal{Q}'(\sqrt{\mathcal{S}})}{\mathcal{Q}(\mathcal{S})} \cdot (d-k) \cdot \left(\frac{1+2\gamma}{8\gamma \cdot \sqrt{\mathcal{S}}}\right) \\
&= -\frac{\mathcal{Q}'(\sqrt{\mathcal{S}})}{\sqrt{\mathcal{S}} \cdot \mathcal{Q}(\sqrt{\mathcal{S}})} \cdot (d-k) \cdot \left(\frac{1}{8\gamma} + \frac{1}{4}\right).
\end{aligned}
\tag{121}
$$

Finally, from Eq. 107, Eq. 112, Eq. 113, Eq. 114, Eq. 121, it follows that:

$$
\begin{aligned}
\eta &= 100 \cdot h(N) \\
&= 100 \cdot \left( \ell(0) + \frac{\ell'(0)}{N} + \mathcal{O}\left(\frac{1}{N^2}\right) \right) \\
&= 100 \cdot \frac{\ell'(0)}{N} + \mathcal{O}\left(\frac{1}{N^2}\right) \\
&= -\frac{100 \cdot \mathcal{Q}'(\sqrt{\mathcal{S}})}{\sqrt{\mathcal{S}} \cdot \mathcal{Q}(\sqrt{\mathcal{S}})} \cdot (d-k) \cdot \left(\frac{1}{8\gamma} + \frac{1}{4}\right) \cdot \frac{1}{N} + \mathcal{O}\left(\frac{1}{N^2}\right)
\end{aligned}
\tag{122}
$$

We proceed with Eq. 122 and substitute Eq. 112:

$$
N = \frac{N_T}{1+\gamma}
$$

to get the following approximation for $\eta$:

$$
\begin{aligned}
\eta &= -100 \cdot \frac{\mathcal{Q}'(\sqrt{\mathcal{S}})}{\sqrt{\mathcal{S}} \cdot \mathcal{Q}(\sqrt{\mathcal{S}})} \cdot (d-k) \cdot \left(\frac{1}{8\gamma} + \frac{1}{4}\right) \cdot (1+\gamma) \cdot \frac{1}{N_T} + \mathcal{O}\left(\frac{1}{N_T^2}\right) \\
&= -50 \cdot \frac{\mathcal{Q}'(\sqrt{\mathcal{S}})}{\sqrt{\mathcal{S}} \cdot \mathcal{Q}(\sqrt{\mathcal{S}})} \cdot (d-k) \cdot \left(\frac{1}{4\gamma} + \frac{1}{2}\right) \cdot (1+\gamma) \cdot \frac{1}{N_T} + \mathcal{O}\left(\frac{1}{N_T^2}\right) \\
&= -50 \cdot \frac{\mathcal{Q}'(\sqrt{\mathcal{S}})}{\sqrt{\mathcal{S}} \cdot \mathcal{Q}(\sqrt{\mathcal{S}})} \cdot (d-k) \cdot \left(\frac{1+2\gamma}{4\gamma}\right) \cdot (1+\gamma) \cdot \frac{1}{N_T} + \mathcal{O}\left(\frac{1}{N_T^2}\right) \\
&= -\frac{50 \cdot \mathcal{Q}'(\sqrt{\mathcal{S}})}{4 \cdot \sqrt{\mathcal{S}}\mathcal{Q}(\sqrt{\mathcal{S}})} \cdot (d-k) \cdot (1+2\gamma) \cdot \left(1 + \frac{1}{\gamma}\right) \cdot \frac{1}{N_T} + \mathcal{O}\left(\frac{1}{N_T^2}\right) \\
&= -\frac{25 \cdot \mathcal{Q}'(\sqrt{\mathcal{S}})}{2 \cdot \sqrt{\mathcal{S}}\mathcal{Q}(\sqrt{\mathcal{S}})} \cdot (d-k) \cdot \left(3 + 2\gamma + \frac{1}{\gamma}\right) \cdot \frac{1}{N_T} + \mathcal{O}\left(\frac{1}{N_T^2}\right) \\
&= \frac{25}{2\sqrt{2\pi}} \cdot \frac{\exp\left(-\frac{\mathcal{S}}{2}\right)}{\sqrt{\mathcal{S}} \cdot \mathcal{Q}\left(\sqrt{\mathcal{S}}\right)} \cdot \left(3 + 2\gamma + \frac{1}{\gamma}\right) \cdot (d-k) \cdot \frac{1}{N_T} + \mathcal{O}\left(\frac{1}{N_T^2}\right)
\end{aligned}
\tag{123}
$$

where we used the following property of the $\mathcal{Q}$ function:

$$
\mathcal{Q}'(x) = -\frac{1}{\sqrt{2\pi}} \cdot \exp\left(-\frac{x^2}{2}\right).
$$

It is now left to show the conclusions. For $N_T \gg 1$, we have from Eq. 123 that

$$
\eta = C \cdot f(\mathcal{S}) \cdot g(\gamma) \cdot (d-k) \cdot \frac{1}{N_T}
\tag{124}
$$

where $C = \dfrac{25}{2\sqrt{2\pi}} > 0$, and

$$
\begin{cases}
f(\mathcal{S}) = \dfrac{\exp\left(-\frac{\mathcal{S}}{2}\right)}{\sqrt{\mathcal{S}} \cdot \mathcal{Q}(\sqrt{\mathcal{S}})} \\
g(\gamma) = 3 + 2\gamma + \dfrac{1}{\gamma}
\end{cases}
\tag{125}
$$

It is now clear from Eq. 124 that as $d - k$ increases, the efficiency increases (linearly), and as $N_T$ increases, the efficiency decreases. It is easy to see that in $(0, 1]$, the function $g$ defined in Eq. 125 achieves a minimum at $\gamma = \dfrac{1}{\sqrt{2}}$:

$$
g'(\gamma) = 2 - \frac{1}{\gamma^2} = 0 \Rightarrow \gamma = \pm\frac{1}{\sqrt{2}}.
$$

It is easy to check that

$$
g\left(\frac{1}{\sqrt{2}}\right) < \lim_{x \to 0^+} g(x) = \infty, \quad g\left(\frac{1}{\sqrt{2}}\right) < g(1).
$$

Thus, from Eq. 124, in the range $\left(0, \frac{1}{\sqrt{2}}\right]$, as $\gamma$ decreases, the efficiency increases. Finally, we will prove that the function $f$, defined in Eq. 125, decreases as $\mathcal{S}$ increases, and thus from Eq. 124, the efficiency decreases as $\mathcal{S}$ increases: It is now sufficient to prove the following:

$$f'(\mathcal{S}) = \frac{-\frac{1}{2}\exp\left(-\frac{\mathcal{S}}{2}\right)\cdot\sqrt{\mathcal{S}}\mathcal{Q}\left(\sqrt{\mathcal{S}}\right) - \frac{d}{d\mathcal{S}}\left(\sqrt{\mathcal{S}}\mathcal{Q}\left(\sqrt{\mathcal{S}}\right)\right)\cdot\exp\left(-\frac{\mathcal{S}}{2}\right)}{\mathcal{S}\mathcal{Q}^2\left(\sqrt{\mathcal{S}}\right)} < 0$$

$$-\frac{1}{2}\sqrt{\mathcal{S}}\cdot\mathcal{Q}\left(\sqrt{\mathcal{S}}\right) - \frac{d}{d\mathcal{S}}\left(\sqrt{\mathcal{S}}\mathcal{Q}\left(\sqrt{\mathcal{S}}\right)\right) < 0$$

$$-\frac{1}{2}\sqrt{\mathcal{S}}\mathcal{Q}\left(\sqrt{\mathcal{S}}\right) < \frac{1}{2\sqrt{\mathcal{S}}}\mathcal{Q}\left(\sqrt{\mathcal{S}}\right) + \sqrt{\mathcal{S}}\cdot\mathcal{Q}'\left(\sqrt{\mathcal{S}}\right)\frac{1}{2\sqrt{\mathcal{S}}}$$

$$-\sqrt{\mathcal{S}}\mathcal{Q}\left(\sqrt{\mathcal{S}}\right) < \frac{1}{\sqrt{\mathcal{S}}}\mathcal{Q}\left(\sqrt{\mathcal{S}}\right) - \frac{1}{\sqrt{2\pi}}\exp\left(-\frac{\mathcal{S}}{2}\right)$$

$$\left(\sqrt{\mathcal{S}} + \frac{1}{\sqrt{\mathcal{S}}}\right)\mathcal{Q}\left(\sqrt{\mathcal{S}}\right) > \frac{1}{\sqrt{2\pi}}\exp\left(-\frac{\mathcal{S}}{2}\right)$$

$$\mathcal{Q}\left(\sqrt{\mathcal{S}}\right) > \frac{1}{\sqrt{2\pi}}\cdot\frac{\sqrt{\mathcal{S}}}{\mathcal{S}+1}\cdot\exp\left(-\frac{\mathcal{S}}{2}\right)$$

Where we used the identity $\mathcal{Q}'(\mathcal{S}) = -\frac{1}{\sqrt{2\pi}}\cdot\exp\left(-\frac{\mathcal{S}^2}{2}\right)$. Let us now prove the final inequality. It is equivalent to the following inequality $\left(x = \sqrt{\mathcal{S}}\right)$:

$$\forall_{x\geq 0} \; \mathcal{Q}(x) > \frac{1}{\sqrt{2\pi}}\cdot\frac{x}{x^2+1}\cdot\exp\left(-\frac{x^2}{2}\right) = \frac{x}{x^2+1}\cdot\phi(x) \tag{126}$$

where we defined the following function:

$$\phi(x) = \frac{1}{\sqrt{2\pi}}\cdot\exp\left(-\frac{x^2}{2}\right). \tag{127}$$

Indeed, for all $x \geq 0$:

$$\begin{aligned}\left(1 + \frac{1}{x^2}\right)\cdot\mathcal{Q}(x) &= \int_x^\infty \left(1 + \frac{1}{x^2}\right)\cdot\phi(u)\,du \\ &> \int_x^\infty \left(1 + \frac{1}{u^2}\right)\cdot\phi(u)\,du \\ &= -\left[\frac{\phi(u)}{u}\right]_x^\infty \\ &= \frac{\phi(x)}{x}\end{aligned} \tag{128}$$

where we used the following identity:

$$\frac{d}{du}\left(-\frac{\phi(u)}{u}\right) = -\frac{\phi'(u)\cdot u - \phi(u)}{u^2} = \frac{\phi(u) + u^2\cdot\phi(u)}{u^2} = \left(1 + \frac{1}{u^2}\right)\cdot\phi(u). \tag{129}$$

And Eq. 129 follows from the identity $\phi'(u) = -u\cdot\phi(u)$ which is straightforward from the definition of $\phi$ in Eq. 127. Now, from Eq. 128, it follows that:

$$\mathcal{Q}(x) > \frac{\phi(x)}{x}\cdot\frac{x^2}{x^2+1} = \frac{x}{x^2+1}\cdot\phi(x)$$

which proves exactly Eq. 126. It is left to show that:

$$\exists_{N_0\in\mathbb{N}}\forall_{N\geq N_0} \; p_{\boldsymbol{z}}(\text{error}) < p_{\boldsymbol{x}}(\text{error}). \tag{130}$$

That is, $\eta > 0$. We proved in Eq. 119 that:

$$\ell'(0) = -\frac{\mathcal{Q}'(\sqrt{\mathcal{S}})}{\sqrt{\mathcal{S}}\cdot\mathcal{Q}(\sqrt{\mathcal{S}})}\cdot(d - k)\cdot\left(\frac{1}{8\gamma} + \frac{1}{4}\right) > 0. \tag{131}$$

We defined $\ell(x)$ in Eq. 111 as:

$$\ell(x) = h\left(\frac{1}{x}\right) \Rightarrow \ell'(x) = -\frac{1}{x^2} \cdot h'\left(\frac{1}{x}\right) \tag{132}$$

where we used the chain rule. Finally, from Eq. 131, Eq. 132, it follows that:

$$\ell'(0) = -\lim_{x \to 0^+} \frac{1}{x^2} \cdot h'\left(\frac{1}{x}\right) = -\lim_{t \to \infty} t^2 \cdot h'(t) > 0$$

where we used the fact that if the two-sided limit exists, then each one-sided limit exists and they are equal to the limit. In total,

$$\lim_{t \to \infty} t^2 \cdot h'(t) < 0.$$

This means that for large enough $t$, we have

$$h'(t) < 0$$

which implies that $h$ is strictly decreasing. Thus, from Eq. 107, we have that for $N \gg 1$, $\eta$ is decreasing. In addition, from Eq. 111, Eq. 114, it follows that:

$$\lim_{N \to \infty} \eta = 100 \cdot \lim_{N \to \infty} h(N) = 100 \cdot \lim_{N \to \infty} \ell\left(\frac{1}{N}\right) = 100 \cdot \ell(0) = 0$$

Finally, $\eta$ is decreasing for large enough $N$ and approaches 0. It is now easy to see that:

$$\exists_{N_0 \in \mathbb{N}} \forall_{N \geq N_0} \, \eta > 0$$

which exactly proves Eq. 130.

$\square$

### A.8 Proof of Theorem 8

Let us state an extended and more detailed version of the Theorem 8.

**Theorem** (Analysis of the maximal efficiency). *Fix $\gamma = 1$, and let $\mathcal{S} > 0, 1 \leq k < d$. Consider the efficiency $\eta = \eta(N)$ as a function of continuous $N \in \mathbb{R}_+$. The following hold.*

- *The maximal efficiency $\eta_{\max} = \max_{N \geq 0} \eta(N)$ increases as a function of $\mathcal{S} > 0$.*

- *For fixed $r := \dfrac{d}{k}$ and $k \gg \max\{1, \mathcal{S}\}$, the maximizer $N_{\max} = \arg\max_{N \geq 0} \eta(N)$ decreases with $\mathcal{S} > 0$ in both regimes $\mathcal{S} \ll 1, \mathcal{S} \gg 1$. In addition, in the regime $\mathcal{S} \ll 1$ the following approximation holds:*

$$N_{\max} \approx \frac{k}{2\mathcal{S}} \cdot \frac{r^{\frac{2}{3}}\left(r^{\frac{1}{3}} - 1\right)}{r^{\frac{2}{3}} - 1}. \tag{133}$$

*Finally, in the regime $\mathcal{S} \gg 1$, the following approximation holds:*

$$N_{\max} \approx \frac{k}{2\mathcal{S}} \cdot \sqrt{r}. \tag{134}$$

*Proof.* Fix $\gamma = 1$, and take

$$\mathcal{S} > 0, 1 \leq k < d.$$

Let us now define the following parametric function:

$$
\begin{aligned}
f_q(x, \mathcal{S}) &:= \frac{\sqrt{\mathcal{S}}}{\sqrt{\left(\frac{q}{2\mathcal{S}} + 1\right) \cdot \frac{1}{x} + \frac{q}{4\mathcal{S}} \cdot \frac{1}{x^2} + 1}} \\
&= \frac{\sqrt{\mathcal{S}}}{\sqrt{\frac{q+2\mathcal{S}}{2\mathcal{S}x} + \frac{q}{4\mathcal{S}x^2} + 1}} \\
&= \frac{\sqrt{\mathcal{S}}}{\sqrt{\frac{2x(q+2\mathcal{S})+q+4\mathcal{S}^2 x}{4\mathcal{S}x^2}}} \\
&= \frac{2\mathcal{S}x}{\sqrt{2qx + q + 4\mathcal{S}x + 4\mathcal{S}x^2}} \\
&= \frac{2\mathcal{S}x}{\sqrt{(2x+1)q + 4x(x+1)\mathcal{S}}} \\
&= \frac{2\mathcal{S}x}{\sqrt{D_q(x, \mathcal{S})}}
\end{aligned}
\tag{135}
$$

where we denoted

$$
\begin{aligned}
D_q(x, \mathcal{S}) &= (2x+1)q + 4x(x+1)\mathcal{S} \\
&= 4\mathcal{S}x^2 + 2(2\mathcal{S}+q)x + q.
\end{aligned}
\tag{136}
$$

From Definition 13 and Eq. 9, it follows that:

$$
\begin{aligned}
\eta &= 100 \cdot \left(1 - \frac{\hat{p}_{\boldsymbol{z}}(\text{error})}{\hat{p}_{\boldsymbol{x}}(\text{error})}\right) = 100 \cdot \left(1 - \frac{\hat{p}(\mathcal{S}, N, 1, k)}{\hat{p}(\mathcal{S}, N, 1, d)}\right) \\
&= 100 \cdot \left(1 - \frac{2 \cdot \mathcal{Q}\left(f_k(N)\right)}{2 \cdot \mathcal{Q}\left(f_d(N)\right)}\right) \\
&= 100 \cdot \left(1 - \frac{\mathcal{Q}\left(f_k(N)\right)}{\mathcal{Q}\left(f_d(N)\right)}\right) \\
&= 100 \cdot h(N, \mathcal{S})
\end{aligned}
\tag{137}
$$

where we defined the following function:

$$
h(x, \mathcal{S}) := 1 - \frac{\mathcal{Q}\left(f_k(x, \mathcal{S})\right)}{\mathcal{Q}\left(f_d(x, \mathcal{S})\right)}.
\tag{138}
$$

Hence, our task reduces to proving that the following function is increasing:

$$
V(\mathcal{S}) := \max_{x > 0} h(x, \mathcal{S}) = h\left(x^*(\mathcal{S}), \mathcal{S}\right)
\tag{139}
$$

where we denoted

$$
x^*(\mathcal{S}) = \arg\max_{x > 0} h(x, \mathcal{S}).
\tag{140}
$$

We note that the proof holds for each stationary point, and in particular for a maximizer that achieves the maximum value of the function $h(x, \mathcal{S})$. In addition, using the first-order condition:

$$
\frac{\partial h}{\partial x}\left(x^*(\mathcal{S}), \mathcal{S}\right) = 0.
\tag{141}
$$

We now aim to prove that $V'(\mathcal{S}) > 0$, where $V(\mathcal{S})$ is defined in Eq. 139. From the chain rule, we have:

$$
V'(\mathcal{S}) = \frac{\partial h}{\partial \mathcal{S}}\left(x^*(\mathcal{S}), \mathcal{S}\right) + \frac{\partial h}{\partial x}\left(x^*(\mathcal{S}), \mathcal{S}\right) \cdot \frac{\partial x^*}{\partial \mathcal{S}} = \frac{\partial h}{\partial \mathcal{S}}\left(x^*(\mathcal{S}), \mathcal{S}\right)
\tag{142}
$$

where we used Eq. 141. Let us compute the partial derivative of $h$ with respect to $\mathcal{S}$:

$$
\begin{aligned}
\frac{\partial h}{\partial \mathcal{S}} &= -\frac{\partial}{\partial \mathcal{S}}\left(\frac{\mathcal{Q}\left(f_k(x,\mathcal{S})\right)}{\mathcal{Q}\left(f_d(x,\mathcal{S})\right)}\right) \\
&= -\frac{\frac{\partial}{\partial \mathcal{S}}\mathcal{Q}\left(f_k(x,\mathcal{S})\right)\cdot \mathcal{Q}\left(f_d(x,\mathcal{S})\right) - \frac{\partial}{\partial \mathcal{S}}\mathcal{Q}\left(f_d(x,\mathcal{S})\right)\cdot \mathcal{Q}\left(f_k(x,\mathcal{S})\right)}{\mathcal{Q}^2\left(f_d(x,\mathcal{S})\right)} \\
&= \frac{\frac{\partial}{\partial \mathcal{S}}\mathcal{Q}\left(f_d(x,\mathcal{S})\right)\cdot \mathcal{Q}\left(f_k(x,\mathcal{S})\right) - \frac{\partial}{\partial \mathcal{S}}\mathcal{Q}\left(f_k(x,\mathcal{S})\right)\cdot \mathcal{Q}\left(f_d(x,\mathcal{S})\right)}{\mathcal{Q}^2\left(f_d(x,\mathcal{S})\right)} \\
&= \frac{\mathcal{Q}'\left(f_d(x,\mathcal{S})\right)\cdot \frac{\partial f_d}{\partial \mathcal{S}}\cdot \mathcal{Q}\left(f_k(x,\mathcal{S})\right) - \mathcal{Q}'\left(f_k(x,\mathcal{S})\right)\cdot \frac{\partial f_k}{\partial \mathcal{S}}\cdot \mathcal{Q}\left(f_d(x,\mathcal{S})\right)}{\mathcal{Q}^2\left(f_d(x,\mathcal{S})\right)} \\
&= \frac{1}{\sqrt{2\pi}}\cdot\frac{\exp\left(-\frac{1}{2}f_k^2(x,\mathcal{S})\right)\cdot\frac{\partial f_k}{\partial \mathcal{S}}\cdot\mathcal{Q}\left(f_d(x,\mathcal{S})\right) - \exp\left(-\frac{1}{2}f_d^2(x,\mathcal{S})\right)\cdot\frac{\partial f_d}{\partial \mathcal{S}}\cdot\mathcal{Q}\left(f_k(x,\mathcal{S})\right)}{\mathcal{Q}^2\left(f_d(x,\mathcal{S})\right)}
\end{aligned}
\tag{143}
$$

where we used the identity $\mathcal{Q}'(x) = -\dfrac{1}{\sqrt{2\pi}}\cdot\exp\left(-\dfrac{1}{2}x^2\right)$. It now follows immediately from Eq. 142, Eq. 143, that proving $V'(\mathcal{S}) > 0$ is equivalent to proving the following inequality:

$$
\exp\left(-\frac{1}{2}f_k^2(x^*,\mathcal{S})\right)\frac{\partial f_k}{\partial \mathcal{S}}(x^*,\mathcal{S})\mathcal{Q}\left(f_d(x^*,\mathcal{S})\right) > \exp\left(-\frac{1}{2}f_d^2(x^*,\mathcal{S})\right)\frac{\partial f_d}{\partial \mathcal{S}}(x^*,\mathcal{S})\mathcal{Q}\left(f_k(x^*,\mathcal{S})\right).
\tag{144}
$$

We now turn to the first order condition for $x^*(\mathcal{S})$ in Eq. 141, and thus equate the partial derivative of $h$ with respect to $x$ to zero. Similarly to Eq. 143, one can prove that for all $x > 0$ we have:

$$
\begin{aligned}
\frac{\partial h}{\partial x} &= -\frac{\partial}{\partial x}\left(\frac{\mathcal{Q}\left(f_k(x)\right)}{\mathcal{Q}\left(f_d(x)\right)}\right) \\
&= \frac{1}{\sqrt{2\pi}}\cdot\frac{\exp\left(-\frac{1}{2}f_k^2(x,\mathcal{S})\right)\cdot\frac{\partial f_k}{\partial x}\cdot\mathcal{Q}\left(f_d(x,\mathcal{S})\right) - \exp\left(-\frac{1}{2}f_d^2(x,\mathcal{S})\right)\cdot\frac{\partial f_d}{\partial x}\cdot\mathcal{Q}\left(f_k(x,\mathcal{S})\right)}{\mathcal{Q}^2\left(f_d(x,\mathcal{S})\right)}.
\end{aligned}
\tag{145}
$$

That is, equating $\dfrac{\partial h}{\partial x} = 0$ yields the following equation for $x^*(\mathcal{S})$:

$$
\exp\left(-\frac{1}{2}f_k^2(x^*,\mathcal{S})\right)\frac{\partial f_k}{\partial x}(x^*,\mathcal{S})\mathcal{Q}\left(f_d(x^*,\mathcal{S})\right) = \exp\left(-\frac{1}{2}f_d^2(x^*,\mathcal{S})\right)\frac{\partial f_d}{\partial x}(x^*,\mathcal{S})\mathcal{Q}\left(f_k(x^*,\mathcal{S})\right).
\tag{146}
$$

We now divide both sides of the inequality in Eq. 144 by the (positive) value we have in the latter equality, in order to get the following simplified inequality:

$$
\frac{\frac{\partial f_k}{\partial \mathcal{S}}(x^*,\mathcal{S})}{\frac{\partial f_k}{\partial x}(x^*,\mathcal{S})} > \frac{\frac{\partial f_d}{\partial \mathcal{S}}(x^*,\mathcal{S})}{\frac{\partial f_d}{\partial x}(x^*,\mathcal{S})}.
\tag{147}
$$

We indeed divided by a positive amount, because $\exp(\cdot) > 0$, $\mathcal{Q}(\cdot) > 0$, and $\dfrac{\partial f_q}{\partial x} > 0$: for all $x > 0$, from Eq. 135, Eq. 136, we have:

$$
\begin{aligned}
\frac{\partial f_q}{\partial x} &= \frac{\partial}{\partial x}\left(\frac{2\mathcal{S}x}{\sqrt{D_q(x,\mathcal{S})}}\right) \\
&= 2\mathcal{S}\cdot\frac{\partial}{\partial x}\left(\frac{x}{\sqrt{D_q(x,\mathcal{S})}}\right) \\
&= 2\mathcal{S}\cdot\left(\frac{\sqrt{D_q(x,\mathcal{S})} - x\cdot\frac{\partial}{\partial x}\left(\sqrt{D_q(x,\mathcal{S})}\right)}{D_q(x,\mathcal{S})}\right) \\
&= 2\mathcal{S}\cdot\left(\frac{\sqrt{D_q(x,\mathcal{S})} - x\cdot\frac{\partial D_q(x,\mathcal{S})/\partial x}{2\sqrt{D_q(x,\mathcal{S})}}}{D_q(x,\mathcal{S})}\right) \\
&= 2\mathcal{S}\cdot\left(\frac{2\cdot D_q(x,\mathcal{S}) - x\cdot\frac{\partial D_q(x,\mathcal{S})}{\partial x}}{2\cdot(D_q(x,\mathcal{S}))^{3/2}}\right) \\
&= \frac{\mathcal{S}}{(D_q(x,\mathcal{S}))^{3/2}}\cdot\left(2\left(4\mathcal{S}x^2 + 2\left(2\mathcal{S}+q\right)x + q\right) - x\left(8\mathcal{S}x + 2\left(2\mathcal{S}+q\right)\right)\right) \\
&= \frac{\mathcal{S}}{(D_q(x,\mathcal{S}))^{3/2}}\cdot\left(2\left(2\mathcal{S}+q\right)x + 2q\right) \\
&= \frac{2\mathcal{S}}{(D_q(x,\mathcal{S}))^{3/2}}\cdot\left(\left(2\mathcal{S}+q\right)x + q\right) > 0
\end{aligned}
\tag{148}
$$

where we used the definition of $D_q(x,\mathcal{S})$ in Eq. 136. We now compute the partial derivative of $f_q$ with respect to $\mathcal{S}$:

$$
\begin{aligned}
\frac{\partial f_q}{\partial \mathcal{S}} &= \frac{\partial}{\partial \mathcal{S}}\left(\frac{2\mathcal{S}x}{\sqrt{D_q(x,\mathcal{S})}}\right) \\
&= 2x\cdot\frac{\partial}{\partial \mathcal{S}}\left(\frac{\mathcal{S}}{\sqrt{D_q(x,\mathcal{S})}}\right) \\
&= 2x\cdot\left(\frac{\sqrt{D_q(x,\mathcal{S})} - \mathcal{S}\cdot\frac{\partial}{\partial \mathcal{S}}\left(\sqrt{D_q(x,\mathcal{S})}\right)}{D_q(x,\mathcal{S})}\right) \\
&= 2x\cdot\left(\frac{\sqrt{D_q(x,\mathcal{S})} - \mathcal{S}\cdot\frac{\partial D_q(x,\mathcal{S})/\partial \mathcal{S}}{2\sqrt{D_q(x,\mathcal{S})}}}{D_q(x,\mathcal{S})}\right) \\
&= 2x\cdot\left(\frac{2\cdot D_q(x,\mathcal{S}) - \mathcal{S}\cdot\frac{\partial D_q(x,\mathcal{S})}{\partial \mathcal{S}}}{2\cdot(D_q(x,\mathcal{S}))^{3/2}}\right).
\end{aligned}
\tag{149}
$$

We now use the definition of $D_q(x,\mathcal{S})$ from Eq. 136, and get

$$
\begin{aligned}
\frac{\partial f_q}{\partial \mathcal{S}} &= \frac{x}{(D_q(x,\mathcal{S}))^{3/2}}\cdot\left(2\left(4\mathcal{S}x^2 + 2\left(2\mathcal{S}+q\right)x + q\right) - \mathcal{S}\cdot 4x(x+1)\right) \\
&= \frac{x}{(D_q(x,\mathcal{S}))^{3/2}}\cdot\left(4\mathcal{S}x^2 + 4\left(2\mathcal{S}+q\right)x - 4\mathcal{S}x + 2q\right) \\
&= \frac{x}{(D_q(x,\mathcal{S}))^{3/2}}\cdot\left(4\mathcal{S}x^2 + 4\left(\mathcal{S}+q\right)x + 2q\right) \\
&= \frac{2x}{(D_q(x,\mathcal{S}))^{3/2}}\cdot\left(2\mathcal{S}x^2 + 2\left(\mathcal{S}+q\right)x + q\right).
\end{aligned}
\tag{150}
$$

Let us now define

$$
\begin{aligned}
R_q(x, \mathcal{S}) &:= \frac{\frac{\partial f_q}{\partial \mathcal{S}}(x, \mathcal{S})}{\frac{\partial f_q}{\partial x}(x, \mathcal{S})} \\
&= \frac{\frac{2x}{(D_q(x,\mathcal{S}))^{3/2}} \cdot \left(2\mathcal{S}x^2 + 2(\mathcal{S} + q)x + q\right)}{\frac{2\mathcal{S}}{(D_q(x,\mathcal{S}))^{3/2}} \cdot ((2\mathcal{S} + q)x + q)} \\
&= \frac{x}{\mathcal{S}} \cdot \left(\frac{2\mathcal{S}x^2 + 2\mathcal{S}x + 2x \cdot q + q}{2\mathcal{S}x + x \cdot q + q}\right) \\
&= \frac{x}{\mathcal{S}} \cdot \left(\frac{2\mathcal{S}x(x+1) + (2x+1) \cdot q}{2\mathcal{S}x + (x+1) \cdot q}\right)
\end{aligned}
\tag{151}
$$

where we used the partial derivatives of $f_q$, computed in Eq. 148, Eq. 150. We remember that we need to prove Eq. 147, which is equivalent to $R_q(x, \mathcal{S})$ being a decreasing function in the argument $q$ (this is because $1 \leq k < d$). Indeed, let us compute

$$
\begin{aligned}
\frac{\partial R_q(x, \mathcal{S})}{\partial q} &= \frac{x}{\mathcal{S}} \cdot \frac{\partial}{\partial q}\left(\frac{2\mathcal{S}x(x+1) + (2x+1) \cdot q}{2\mathcal{S}x + (x+1) \cdot q}\right) \\
&= \frac{x}{\mathcal{S}} \cdot \frac{(2x+1) \cdot (2\mathcal{S}x + (x+1) \cdot q) - (x+1) \cdot (2\mathcal{S}x(x+1) + (2x+1) \cdot q)}{(2\mathcal{S}x + (x+1) \cdot q)^2} \\
&= \frac{x}{\mathcal{S}} \cdot \frac{2x(2x+1) \cdot \mathcal{S} - 2x(x+1)^2 \cdot \mathcal{S}}{(2\mathcal{S}x + (x+1) \cdot q)^2} \\
&= \frac{2x^2 \cdot \left(2x + 1 - (x+1)^2\right)}{(2\mathcal{S}x + (x+1) \cdot q)^2} \\
&= -\frac{2x^4}{(2\mathcal{S}x + (x+1) \cdot q)^2}.
\end{aligned}
\tag{152}
$$

That is,

$$
\frac{\partial R_q(x, \mathcal{S})}{\partial q} = -\frac{2x^4}{(2\mathcal{S}x + (x+1) \cdot q)^2} < 0
$$

which proves Eq. 147. We argued that this is equivalent to Eq. 144. As we proved in Eq. 142, this inequality is equivalent to proving $V'(\mathcal{S}) > 0$. Finally, the result is straightforward because $\eta = 100 \cdot h(N, \mathcal{S})$. This proves the first part of the theorem.

We will now prove the second part of the theorem. Let us fix $d > k \gg \max\{1, \mathcal{S}\}$, and $r := \frac{d}{k} > 1$. We prove that the maximizer $x^*(\mathcal{S})$, defined in Eq. 140, decreases as a function of $\mathcal{S}$. Let us first define the following rescaled $x$ value:

$$
x(t) := \frac{k}{\mathcal{S}} \cdot t.
\tag{153}
$$

Thus,

$$
x^*(\mathcal{S}) = \frac{k}{\mathcal{S}} \cdot \arg\max_{t > 0} h(x(t), \mathcal{S}).
\tag{154}
$$

From Eq. 135, it follows that:

$$
\begin{aligned}
f_q\left(x(t), \mathcal{S}\right) &= \frac{2\mathcal{S} \cdot x(t)}{\sqrt{(2x(t)+1)\,q + 4x(t)\,(x(t)+1)\,\mathcal{S}}} \\
&\sim \frac{2\mathcal{S} \cdot x(t)}{\sqrt{2x(t) \cdot q + 4x^2(t) \cdot \mathcal{S}}} \\
&= \frac{2\mathcal{S}}{\sqrt{\frac{2q}{x(t)} + 4\mathcal{S}}} \\
&= \frac{2\mathcal{S}}{\sqrt{2q \cdot \frac{\mathcal{S}}{k \cdot t} + 4\mathcal{S}}} \\
&= \frac{\sqrt{2}\sqrt{\mathcal{S}} \cdot \sqrt{kt}}{\sqrt{2q + 4kt}} \\
&= \sqrt{\frac{2\mathcal{S}k \cdot t}{q + 2k \cdot t}} \\
&= \sqrt{\frac{2\mathcal{S}t}{\frac{q}{k} + 2t}}
\end{aligned}
\tag{155}
$$

where we assumed $x(t) \gg 1$ which follows from $q \gg 1, \mathcal{S} \ll q, t = O(1)$. We explain the assumption $t = O(1)$ in a moment. Now, from Eq. 138, we have:

$$
h\left(x(t), \mathcal{S}\right) = 1 - \frac{\mathcal{Q}\left(f_k\left(x(t), \mathcal{S}\right)\right)}{\mathcal{Q}\left(f_d\left(x(t), \mathcal{S}\right)\right)} \sim 1 - \frac{\mathcal{Q}\left(\sqrt{\frac{2\mathcal{S}t}{1+2t}}\right)}{\mathcal{Q}\left(\sqrt{\frac{2\mathcal{S}t}{r+2t}}\right)}
\tag{156}
$$

this motivates the assumption $t = O(1)$ we used in Eq. 155: the maximizer

$$
\begin{aligned}
t^*(r, \mathcal{S}) = \arg\max_{t>0} h\left(x(t), \mathcal{S}\right) &\sim \arg\min_{t>0} \left(\frac{\mathcal{Q}\left(\sqrt{\frac{2\mathcal{S}t}{1+2t}}\right)}{\mathcal{Q}\left(\sqrt{\frac{2\mathcal{S}t}{r+2t}}\right)}\right) \\
&= \arg\min_{t>0} g_{\mathcal{S},r}(t)
\end{aligned}
\tag{157}
$$

is a function of $r = O(1)$ which is a constant and $\mathcal{S} \ll k$, and thus in the region of interest (close to the maximizer), $t = O(1)$ does not scale with $k$. The key insight here is that $t^*(r, \mathcal{S})$ doesn't depend on $k, d$. We also defined the following function:

$$
g_{\mathcal{S},r}(t) := \frac{\mathcal{Q}\left(\sqrt{\frac{2\mathcal{S}t}{1+2t}}\right)}{\mathcal{Q}\left(\sqrt{\frac{2\mathcal{S}t}{r+2t}}\right)}.
\tag{158}
$$

Finally, we would like to analyze the dependency of $t^*(r, \mathcal{S})$ on $\mathcal{S}$. We will prove that in both regimes $\mathcal{S} \ll 1, \mathcal{S} \gg 1$, we have that $t^*(r, \mathcal{S})$ does not depend on $\mathcal{S}$, and thus in both regimes, from Eq. 154, Eq. 157, the maximizer

$$
x^*(\mathcal{S}) = \frac{k}{\mathcal{S}} \cdot t^*(r)
\tag{159}
$$

is decreasing as a function of $\mathcal{S} > 0$. For the regime $\mathcal{S} \ll 1$: We use the approximation $\mathcal{Q}(x) \sim \frac{1}{2} - \frac{1}{\sqrt{2\pi}} \cdot x$ for $x \ll 1$ and Eq. 158 to get

$$
\begin{aligned}
g_{\mathcal{S},r}(t) = \frac{\mathcal{Q}\left(\sqrt{\frac{2\mathcal{S}t}{1+2t}}\right)}{\mathcal{Q}\left(\sqrt{\frac{2\mathcal{S}t}{r+2t}}\right)} &\sim \frac{\frac{1}{2} - \frac{1}{\sqrt{2\pi}} \cdot \sqrt{\frac{2\mathcal{S}t}{1+2t}}}{\frac{1}{2} - \frac{1}{\sqrt{2\pi}} \cdot \sqrt{\frac{2\mathcal{S}t}{r+2t}}} \\
&= \frac{1 - \frac{2}{\sqrt{2\pi}} \cdot \sqrt{\frac{2\mathcal{S}t}{1+2t}}}{1 - \frac{2}{\sqrt{2\pi}} \cdot \sqrt{\frac{2\mathcal{S}t}{r+2t}}} \\
&\sim \left(1 - \frac{2}{\sqrt{2\pi}}\sqrt{\frac{2\mathcal{S}t}{1+2t}}\right)\left(1 + \frac{2}{\sqrt{2\pi}}\sqrt{\frac{2\mathcal{S}t}{r+2t}}\right) \\
&\sim \left(1 - \frac{2}{\sqrt{2\pi}}\sqrt{\frac{2\mathcal{S}t}{1+2t}} + \frac{2}{\sqrt{2\pi}}\sqrt{\frac{2\mathcal{S}t}{r+2t}}\right) \\
&= 1 - \sqrt{\frac{2}{\pi}} \cdot \left(\sqrt{\frac{2\mathcal{S}t}{1+2t}} - \sqrt{\frac{2\mathcal{S}t}{r+2t}}\right) \\
&= 1 - 2\sqrt{\frac{\mathcal{S}}{\pi}} \cdot \left(\sqrt{\frac{t}{1+2t}} - \sqrt{\frac{t}{r+2t}}\right).
\end{aligned}
\tag{160}
$$

This approximation is motivated from the fact that the argument of the $\mathcal{Q}$ function in both the numerator and the denominator is at most $\sqrt{\mathcal{S}} \ll 1$. Thus, from Eq. 157, we have:

$$
t^*(r, \mathcal{S}) = \arg\min_{t>0} \left(\sqrt{\frac{t}{1+2t}} - \sqrt{\frac{t}{r+2t}}\right)
\tag{161}
$$

is independent of $\mathcal{S}$. We will also calculate the minimizer. Let us define

$$
\Psi_1(t) = \sqrt{\frac{t}{1+2t}} - \sqrt{\frac{t}{r+2t}}.
\tag{162}
$$

We now equate the derivative of $\Psi_1(t)$ to 0:

$$
\begin{aligned}
\frac{d\Psi_1}{dt} &= \frac{d}{dt}\left(\sqrt{\frac{t}{1+2t}}\right) - \frac{d}{dt}\left(\sqrt{\frac{t}{r+2t}}\right) \\
&= \frac{1}{2}\left(\frac{1}{\sqrt{t} \cdot (1+2t)^{\frac{3}{2}}} - \frac{r}{\sqrt{t} \cdot (r+2t)^{\frac{3}{2}}}\right) \\
&= \frac{1}{2\sqrt{t}} \cdot \left(\frac{1}{(1+2t)^{\frac{3}{2}}} - \frac{r}{(r+2t)^{\frac{3}{2}}}\right) \\
&= \frac{1}{2\sqrt{t}} \cdot \frac{(r+2t)^{\frac{3}{2}} - r(1+2t)^{\frac{3}{2}}}{(1+2t)^{\frac{3}{2}} \cdot (r+2t)^{\frac{3}{2}}}
\end{aligned}
\tag{163}
$$

where we used the following formula:

$$
\begin{aligned}
\frac{d}{dt}\left(\sqrt{\frac{t}{a+2t}}\right) &= \frac{\frac{d}{dt}\left(\frac{t}{a+2t}\right)}{2\cdot\sqrt{\frac{t}{a+2t}}} \\
&= \frac{\frac{a}{(a+2t)^2}}{2\cdot\sqrt{\frac{t}{a+2t}}} \\
&= \frac{a}{(a+2t)^2}\cdot\frac{\sqrt{a+2t}}{2\cdot\sqrt{t}} \\
&= \frac{a}{2}\cdot\frac{1}{\sqrt{t}\cdot(a+2t)^{\frac{3}{2}}}
\end{aligned}
$$

Finally, from Eq. 163, we have:

$$
\begin{aligned}
(r+2t^*)^{\frac{3}{2}} &= r\left(1+2t^*\right)^{\frac{3}{2}} \\
r+2t^* &= r^{\frac{2}{3}}\cdot\left(1+2t^*\right) \\
2t^*\cdot\left(1-r^{\frac{2}{3}}\right) &= r^{\frac{2}{3}}\left(1-r^{\frac{1}{3}}\right) \\
t^* &= \frac{r^{\frac{2}{3}}\left(1-r^{\frac{1}{3}}\right)}{2\left(1-r^{\frac{2}{3}}\right)}
\end{aligned}
$$

That is, the maximizer is unique, and from Eq. 159, in the regime $\mathcal{S}\ll 1$ we have:

$$
x^*(\mathcal{S}) \sim \frac{1}{2}k\cdot\frac{r^{\frac{2}{3}}\left(r^{\frac{1}{3}}-1\right)}{r^{\frac{2}{3}}-1}\cdot\frac{1}{\mathcal{S}} \tag{164}
$$

which is strictly decreasing as a function of $\mathcal{S}$.

For the regime $\underline{\mathcal{S}\gg 1}$: We use the approximation $\mathcal{Q}(x)\sim\frac{1}{\sqrt{2\pi}x}\cdot\exp\left(-\frac{x^2}{2}\right)$ and Eq. 158 to get

$$
\begin{aligned}
g_{\mathcal{S},r}(t) = \frac{\mathcal{Q}\left(\sqrt{\frac{2\mathcal{S}t}{1+2t}}\right)}{\mathcal{Q}\left(\sqrt{\frac{2\mathcal{S}t}{r+2t}}\right)} &\sim \frac{\frac{1}{\sqrt{2\pi}}\cdot\sqrt{\frac{1+2t}{2\mathcal{S}t}}\cdot\exp\left(-\frac{1}{2}\cdot\frac{2\mathcal{S}t}{1+2t}\right)}{\frac{1}{\sqrt{2\pi}}\cdot\sqrt{\frac{r+2t}{2\mathcal{S}t}}\cdot\exp\left(-\frac{1}{2}\cdot\frac{2\mathcal{S}t}{r+2t}\right)} \\
&= \sqrt{\frac{1+2t}{r+2t}}\cdot\frac{\exp\left(-\frac{\mathcal{S}t}{1+2t}\right)}{\exp\left(-\frac{\mathcal{S}t}{r+2t}\right)} \\
&= \sqrt{\frac{1+2t}{r+2t}}\cdot\exp\left(-\mathcal{S}t\cdot\left(\frac{1}{1+2t}-\frac{1}{r+2t}\right)\right)
\end{aligned} \tag{165}
$$

minimizing $g_{\mathcal{S},r}(t)$ is equivalent to minimizing $\ln\left(g_{\mathcal{S},r}(t)\right)$:

$$
M_{\mathcal{S},r}(t) := \ln\left(g_{\mathcal{S},r}(t)\right) = \frac{1}{2}\ln\left(1+2t\right) - \frac{1}{2}\ln\left(r+2t\right) - \mathcal{S}t\cdot\left(\frac{1}{1+2t}-\frac{1}{r+2t}\right) \tag{166}
$$

Let us equate the derivative of $M_{\mathcal{S},r}(t)$ to 0:

$$\frac{1}{1+2t} - \frac{1}{r+2t} - \mathcal{S} \cdot \left( \frac{d}{dt}\left(\frac{t}{1+2t}\right) - \frac{d}{dt}\left(\frac{t}{r+2t}\right)\right) = 0$$

$$\frac{1}{1+2t} - \frac{1}{r+2t} - \mathcal{S} \cdot \left( \frac{1}{(1+2t)^2} - \frac{r}{(r+2t)^2}\right) = 0$$

$$\frac{1}{1+2t} - \frac{1}{r+2t} = \frac{\mathcal{S}}{(1+2t)^2} - \frac{\mathcal{S}r}{(r+2t)^2}$$

$$(1+2t)(r+2t)^2 - (1+2t)^2(r+2t) = \mathcal{S}(r+2t)^2 - \mathcal{S}r(1+2t)^2$$

$$(1+2t)(r+2t)\cdot(r-1) = \mathcal{S}\cdot\left(r^2+4rt+4t^2\right) - \mathcal{S}r\cdot\left(1+4t+4t^2\right)$$

$$(r-1)\cdot\left(r+2\left(r+1\right)t+4t^2\right) = \mathcal{S}\cdot\left(r^2+4rt+4t^2\right) - \mathcal{S}r\cdot\left(1+4t+4t^2\right)$$

$$4(r-1)\cdot t^2 + 2(r^2-1)t + r(r-1) = (4\mathcal{S}-4\mathcal{S}r)\,t^2 + \mathcal{S}r^2 - \mathcal{S}r$$

$$4(r-1)\cdot t^2 + 2(r-1)(r+1)t + r(r-1) = 4\mathcal{S}\left(1-r\right)t^2 + \mathcal{S}r\left(r-1\right)$$

$$4t^2 + 2(r+1)t + r = -4\mathcal{S}t^2 + \mathcal{S}r$$

$$4\left(1+\mathcal{S}\right)t^2 + 2(r+1)\cdot t + r\left(1-\mathcal{S}\right) = 0$$

Finally, we take the positive root (because $t^*(r,\mathcal{S}) > 0$) and get:

$$
\begin{aligned}
t^*(r,\mathcal{S}) &= \frac{-2(r+1) + \sqrt{4(r+1)^2 - 16r(1+\mathcal{S})(1-\mathcal{S})}}{8(1+\mathcal{S})} \\
&= \frac{-(r+1) + \sqrt{(r+1)^2 + 4r\left(\mathcal{S}^2-1\right)}}{4(\mathcal{S}+1)}
\end{aligned}
\tag{167}
$$

we note that we got a single solution and that the maximizer is unique, and $\mathcal{S} \gg 1$ and thus the formula is well-defined. We note that

$$\lim_{\mathcal{S}\to\infty} t^*(r,\mathcal{S}) = \frac{\sqrt{r}}{2}$$

and thus, from Eq. 159, in the $\mathcal{S} \gg 1$ regime, we have

$$x^*(\mathcal{S}) \sim \frac{k}{\mathcal{S}} \cdot \frac{-(r+1) + \sqrt{(r+1)^2 + 4r\left(\mathcal{S}^2-1\right)}}{4(\mathcal{S}+1)} \tag{168}$$

and as $\mathcal{S} \to \infty$ we have $x^*(\mathcal{S}) \sim \frac{k}{\mathcal{S}} \cdot \frac{\sqrt{r}}{2}$, and thus in the $\mathcal{S} \gg 1$ regime we have $x^*(\mathcal{S}) \sim \frac{1}{\mathcal{S}}$ is a decreasing function of $\mathcal{S}$.

$\square$

# B  ADDITIONAL THEORETICAL RESULTS

We have established an approximation of the efficiency of the processing $\eta$ for $N \gg 1$. We now do the same for the difference

$$\Delta := \hat{p}_{\boldsymbol{x}}(\text{error}) - \hat{p}_{\boldsymbol{z}}(\text{error}).$$

This allows us to gain insight into the different factors that affect the difference $\Delta$ between the probability of error that is caused by the processing.

**Theorem 10** (Analysis of the asymptotic difference). *Let $\mathcal{S} > 0$, $1 \le k < d$, $0 < \gamma \le 1$. Denote by $N_T = (1 + \gamma) N$ the total number of training samples. With approximation accuracy $\mathcal{O}(1/N_T^2)$, we have*

$$\Delta \approx \frac{1}{4\sqrt{2\pi}} \cdot \frac{\exp\left(-\frac{\mathcal{S}}{2}\right)}{\sqrt{\mathcal{S}}} \cdot \left(3 + 2\gamma + \frac{1}{\gamma}\right) \cdot (d - k) \cdot \frac{1}{N_T}. \tag{169}$$

*In particular, for $N_T \gg 1$: The difference increases when $d - k$ increases or $\gamma$ decreases within $0 < \gamma \le 1/\sqrt{2}$; The difference decreases when $\mathcal{S}$ increases or $N_T$ increases.*

*Proof.* Let us take some $N_T \in \mathbb{N}$ and

$$\mathcal{S} > 0, 1 \le k < d, 0 < \gamma \le 1.$$

We have $N_T = N + \gamma N = (1 + \gamma)N$ the total number of training samples, and thus $N = \dfrac{N_T}{1 + \gamma}$.

Let us define the following parametric function:

$$f_{s,a,q}(x) := \frac{\sqrt{\mathcal{S}} + s \cdot \frac{(1-\gamma)\cdot q}{4\gamma \cdot \sqrt{\mathcal{S}}} \cdot \frac{1}{x}}{\sqrt{\left(\frac{(1+\gamma)\cdot q}{4\gamma \cdot \mathcal{S}} + \frac{1}{a}\right) \cdot \frac{1}{x} + \frac{(1+\gamma^2)\cdot q}{8\gamma^2 \cdot \mathcal{S}} \cdot \frac{1}{x^2} + 1}} = \frac{\sqrt{\mathcal{S}} + \frac{B}{x}}{\sqrt{\frac{C}{x} + \frac{D}{x^2} + 1}} \tag{170}$$

where the parameters $B, C, D$ are:

$$\begin{cases} B = s \cdot \dfrac{(1 - \gamma) \cdot q}{4\gamma \cdot \sqrt{\mathcal{S}}} \\ C = \dfrac{(1 + \gamma) \cdot q}{4\gamma \cdot \mathcal{S}} + \dfrac{1}{a} \\ D = \dfrac{(1 + \gamma^2) \cdot q}{8\gamma^2 \cdot \mathcal{S}} \end{cases} \tag{171}$$

From the Definition $\Delta := \hat{p}_{\boldsymbol{x}}(\text{error}) - \hat{p}_{\boldsymbol{z}}(\text{error})$ and Eq. 9, it follows that:

$$\begin{aligned} \Delta = \hat{p}_{\boldsymbol{x}}(\text{error}) - \hat{p}_{\boldsymbol{z}}(\text{error}) &= \hat{p}(\mathcal{S}, N, \gamma, d) - \hat{p}(\mathcal{S}, N, \gamma, k) \\ &= [\mathcal{Q}(f_{1,\gamma,d}(N)) + \mathcal{Q}(f_{-1,1,d}(N))] - [\mathcal{Q}(f_{1,\gamma,k}(N)) + \mathcal{Q}(f_{-1,1,k}(N))] \\ &= h(N) \end{aligned} \tag{172}$$

where we defined the following function:

$$h(x) := [\mathcal{Q}(f_{1,\gamma,d}(x)) + \mathcal{Q}(f_{-1,1,d}(x))] - [\mathcal{Q}(f_{1,\gamma,k}(x)) + \mathcal{Q}(f_{-1,1,k}(x))]. \tag{173}$$

Now, let us define the following parametric function:

$$g_{s,a,q}(x) := f_{s,a,q}\left(\frac{1}{x}\right) = \frac{\sqrt{\mathcal{S}} + B \cdot x}{\sqrt{D \cdot x^2 + C \cdot x + 1}} \tag{174}$$

where we used Eq. 170, Eq. 171. Let us also define:

$$\ell(x) := h\left(\frac{1}{x}\right) = [\mathcal{Q}(g_{1,\gamma,d}(x)) + \mathcal{Q}(g_{-1,1,d}(x))] - [\mathcal{Q}(g_{1,\gamma,k}(x)) + \mathcal{Q}(g_{-1,1,k}(x))] \tag{175}$$

where we used Eq. 174, Eq. 173. Thus, the first-order Taylor expansion of $\ell$:

$$\ell(x) = \ell(0) + \ell'(0) \cdot x + \mathcal{O}\left(x^2\right)$$

Where the approximation is exact for $x \ll 1$. Thus, the following is exact for $x \gg 1$:

$$x \gg 1 \Rightarrow h(x) = \ell\left(\frac{1}{x}\right) = \ell(0) + \frac{\ell'(0)}{x} + \mathcal{O}\left(\frac{1}{x^2}\right). \tag{176}$$

We have

$$N = \frac{N_T}{1+\gamma} \gg 1 \Leftrightarrow N_T \gg 1 + \gamma \tag{177}$$

Thus,

$$h(N) = \ell(0) + \frac{\ell'(0)}{N} + \mathcal{O}\left(\frac{1}{N^2}\right). \tag{178}$$

Let us first compute $\ell(0)$:

$$\ell(0) = [\mathcal{Q}(g_{1,\gamma,d}(0)) + \mathcal{Q}(g_{-1,1,d}(0))] - [\mathcal{Q}(g_{1,\gamma,k}(0)) + \mathcal{Q}(g_{-1,1,k}(0))] = 2 \cdot \mathcal{Q}(\sqrt{\mathcal{S}}) - 2 \cdot \mathcal{Q}(\sqrt{\mathcal{S}}) = 0 \tag{179}$$

Finally, we will compute $\ell'(0)$: We first compute $g'_{s,a,q}(0)$. From Eq. 174 it follows that:

$$g'_{s,a,q}(x) = \frac{B \cdot \sqrt{D \cdot x^2 + C \cdot x + 1} - \frac{2D \cdot x + C}{2 \cdot \sqrt{D \cdot x^2 + C \cdot x + 1}} \cdot \left(\sqrt{\mathcal{S}} + B \cdot x\right)}{D \cdot x^2 + C \cdot x + 1}$$

$$= \frac{2B \cdot (D \cdot x^2 + C \cdot x + 1) - (2D \cdot x + C) \cdot \left(\sqrt{\mathcal{S}} + B \cdot x\right)}{2 \cdot (D \cdot x^2 + C \cdot x + 1)^{1.5}}$$

$$= \frac{(BC - 2SD) \cdot x + (2B - C\sqrt{\mathcal{S}})}{2 \cdot (D \cdot x^2 + C \cdot x + 1)^{1.5}}.$$

Thus, the derivative at 0 is:

$$\begin{aligned} g'_{s,a,q}(0) &= \frac{2B - C\sqrt{\mathcal{S}}}{2} = B - \frac{1}{2} \cdot C\sqrt{\mathcal{S}} \\ &= \frac{s \cdot (1-\gamma) \cdot q}{4\gamma \cdot S} - \frac{1}{2} \cdot \left(\frac{(1+\gamma) \cdot q}{4\gamma \cdot \mathcal{S}} + \frac{1}{a}\right) \cdot \sqrt{\mathcal{S}} \\ &= \frac{s \cdot (1-\gamma) \cdot q}{4\gamma \cdot S} - \frac{(1+\gamma) \cdot q}{8\gamma \cdot \sqrt{\mathcal{S}}} - \frac{\sqrt{\mathcal{S}}}{2a} \\ &= \frac{2s \cdot (1-\gamma) \cdot q - (1+\gamma) \cdot q}{8\gamma \cdot S} - \frac{\sqrt{\mathcal{S}}}{2a} \\ &= \frac{((2s-1) - (2s+1) \cdot \gamma) \cdot q}{8\gamma \cdot \sqrt{\mathcal{S}}} - \frac{\sqrt{\mathcal{S}}}{2a}. \end{aligned} \tag{180}$$

Now, from Eq. 175, the derivative $\ell'(x)$ reads

$$\begin{aligned} \ell'(x) &= \frac{d}{dx} (v(x) - u(x)) \\ &= v'(x) - u'(x) \end{aligned} \tag{181}$$

where we defined the following auxiliary functions:

$$\begin{cases} u(x) = \mathcal{Q}(g_{1,\gamma,k}(x)) + \mathcal{Q}(g_{-1,1,k}(x)) \\ v(x) = \mathcal{Q}(g_{1,\gamma,d}(x)) + \mathcal{Q}(g_{-1,1,d}(x)) \end{cases} \tag{182}$$

From the chain rule, their derivatives are:

$$\begin{cases} u'(x) = g'_{1,\gamma,k}(x) \cdot \mathcal{Q}'(g_{1,\gamma,k}(x)) + g'_{-1,1,k}(x) \cdot \mathcal{Q}'(g_{-1,1,k}(x)) \\ v'(x) = g'_{1,\gamma,d}(x) \cdot \mathcal{Q}'(g_{1,\gamma,d}(x)) + g'_{-1,1,d}(x) \cdot \mathcal{Q}'(g_{-1,1,d}(x)) \end{cases} \tag{183}$$

Now, from Eq. 181, Eq. 183 it follows that:

$$\ell'(0) = v'(0) - u'(0). \tag{184}$$

It is easy to verify from Eq. 174 that $g_{s,a,q}(0) = \sqrt{\mathcal{S}}$. Thus, from Eq. 182, Eq. 183 and Eq. 180, we have the following formulas:

$$\begin{cases} u'(0) = \left( \dfrac{(1-3\gamma) \cdot k}{8\gamma \cdot \sqrt{\mathcal{S}}} - \dfrac{\sqrt{\mathcal{S}}}{2\gamma} \right) \cdot \mathcal{Q}'(\sqrt{\mathcal{S}}) + \left( \dfrac{-(3+\gamma) \cdot k}{8\gamma \cdot \sqrt{\mathcal{S}}} - \dfrac{\sqrt{\mathcal{S}}}{2} \right) \cdot \mathcal{Q}'(\sqrt{\mathcal{S}}) \\ v'(0) = \left( \dfrac{(1-3\gamma) \cdot d}{8\gamma \cdot \sqrt{\mathcal{S}}} - \dfrac{\sqrt{\mathcal{S}}}{2\gamma} \right) \cdot \mathcal{Q}'(\sqrt{\mathcal{S}}) + \left( \dfrac{-(3+\gamma) \cdot d}{8\gamma \cdot \sqrt{\mathcal{S}}} - \dfrac{\sqrt{\mathcal{S}}}{2} \right) \cdot \mathcal{Q}'(\sqrt{\mathcal{S}}) \end{cases} \tag{185}$$

Now, we substitute Eq. 185 in Eq. 184 to get the following formula for $\ell'(0)$:

$$\begin{aligned} \ell'(0) &= \frac{1-3\gamma}{8\gamma\sqrt{\mathcal{S}}} \mathcal{Q}'\left(\sqrt{\mathcal{S}}\right) \cdot (d-k) - \frac{3+\gamma}{8\gamma\sqrt{\mathcal{S}}} \mathcal{Q}'\left(\sqrt{\mathcal{S}}\right) \cdot (d-k) \\ &= \frac{-2-4\gamma}{8\gamma\sqrt{\mathcal{S}}} \mathcal{Q}'\left(\sqrt{\mathcal{S}}\right) \cdot (d-k) \\ &= \frac{-2(1+2\gamma)}{8\gamma\sqrt{\mathcal{S}}} \cdot \left( -\frac{1}{\sqrt{2\pi}} \exp\left( -\frac{\mathcal{S}}{2} \right) \right) \cdot (d-k) \\ &= \frac{1}{4\sqrt{2\pi}} \cdot \frac{\exp\left(-\frac{\mathcal{S}}{2}\right)}{\sqrt{\mathcal{S}}} \cdot \frac{1+2\gamma}{\gamma} \cdot (d-k) \\ &= \frac{1}{4\sqrt{2\pi}} \cdot \frac{\exp\left(-\frac{\mathcal{S}}{2}\right)}{\sqrt{\mathcal{S}}} \cdot \left( 2 + \frac{1}{\gamma} \right) \cdot (d-k) \end{aligned} \tag{186}$$

where we used the following property of the $\mathcal{Q}$ function:

$$\mathcal{Q}'(x) = -\frac{1}{\sqrt{2\pi}} \cdot \exp\left( -\frac{x^2}{2} \right).$$

Finally, from Eq. 172, Eq. 177, Eq. 178, Eq. 179, Eq. 186, it follows that:

$$\begin{aligned} \Delta &= h(N) \\ &= \ell(0) + \frac{\ell'(0)}{N} + \mathcal{O}\left( \frac{1}{N^2} \right) \\ &= \frac{\ell'(0)}{N} + \mathcal{O}\left( \frac{1}{N^2} \right) \\ &= \frac{1}{4\sqrt{2\pi}} \cdot \frac{\exp\left(-\frac{\mathcal{S}}{2}\right)}{\sqrt{\mathcal{S}}} \cdot \left( 2 + \frac{1}{\gamma} \right) \cdot (d-k) \cdot \frac{1}{N} + \mathcal{O}\left( \frac{1}{N^2} \right). \end{aligned} \tag{187}$$

We proceed with Eq. 187 and substitute Eq. 177:

$$N = \frac{N_T}{1+\gamma}$$

which leads to the following approximation of $\Delta$:

$$\begin{aligned} \Delta &= \frac{1}{4\sqrt{2\pi}} \cdot \frac{\exp\left(-\frac{\mathcal{S}}{2}\right)}{\sqrt{\mathcal{S}}} \cdot \left( 2 + \frac{1}{\gamma} \right) \cdot (d-k) \cdot \frac{1+\gamma}{N_T} + \mathcal{O}\left( \frac{1}{N_T^2} \right) \\ &= \frac{1}{4\sqrt{2\pi}} \cdot \frac{\exp\left(-\frac{\mathcal{S}}{2}\right)}{\sqrt{\mathcal{S}}} \cdot \left( 3 + 2\gamma + \frac{1}{\gamma} \right) \cdot (d-k) \cdot \frac{1}{N_T} + \mathcal{O}\left( \frac{1}{N_T^2} \right). \end{aligned} \tag{188}$$

We now analyze the dependence of $N_T, d-k, \gamma, \mathcal{S}$ on $\Delta$ for $N_T \gg 1$. The results will be identical to those derived in 7. From Eq. 188, the following hold for $N_T \gg 1$:

- $\Delta$ decreases with $N_T$.

- $\Delta$ increases with $d-k$.

- $\Delta$ increases when $\gamma$ decreases within $\gamma \in \left( 0, \dfrac{1}{\sqrt{2}} \right]$: this is because the function

$$f(\gamma) = 3 + 2\gamma + \frac{1}{\gamma}$$

has a minimum at $\gamma = \dfrac{1}{\sqrt{2}}$ within $(0, 1)$.

- $\Delta$ decreases with $\mathcal{S}$: this is because

$$g(\mathcal{S}) := \frac{\exp\left(-\frac{\mathcal{S}}{2}\right)}{\sqrt{\mathcal{S}}}$$

decreases with $\mathcal{S}$: indeed,

$$g'(\mathcal{S}) = \frac{-\frac{1}{2}\exp\left(-\frac{\mathcal{S}}{2}\right)\sqrt{\mathcal{S}} - \frac{1}{2\sqrt{\mathcal{S}}}\exp\left(-\frac{\mathcal{S}}{2}\right)}{\mathcal{S}}$$

$$= -\frac{\exp\left(-\frac{\mathcal{S}}{2}\right)}{2\mathcal{S}} \cdot \left(\sqrt{\mathcal{S}} + \frac{1}{\sqrt{\mathcal{S}}}\right) < 0.$$

$\square$

## C  ADDITIONAL EMPIRICAL DETAILS AND RESULTS (CIFAR-10, DENOISING)

### C.1  EXPERIMENTS COMPUTE RESOURCES

We conducted our experiments using a few NVIDIA RTX 6000 Ada Generation GPUs with 48GB memory. The training time for each data point in Figures 2 , 5 and 6 ranged from one hour to twelve hours, depending on the number of training samples.

### C.2  TRAINING THE CLASSIFIER

We consider the CIFAR-10 dataset (Krizhevsky et al., 2009) and the ResNet18 model (He et al., 2016). To train the model, we use: batch size 128 and 350 epochs; cross-entropy loss; SGD optimizer; learning rate: 0.0679; learning rate decay: 0.1 at epochs 116 and 233; momentum: 0.9; weight decay: 0.0005. This setting yields 90% accuracy for clean data.

Per noise level $\sigma \in \{0.25, 0.4\}$ of the additive Gaussian noise that has been added to the data, we use this setting to train two classifiers: one that operates directly on the noisy data and one that operates on the denoised data.

### C.3  TRAINING THE DENOISER

For the denoiser, we use the DnCNN model (Zhang et al., 2017) and 15,000 training images while ignoring their labels. Per image, the clean version, $\mathbf{x}_{gt}$, is the target and its noisy version, $\mathbf{x}$, is the input to the model. To train the model, we use: batch size 64 and 1000 epochs; MSE loss; Adam optimizer; learning rate: 0.0001; learning rate decay: 0.5 at iterations 20k, 40k, 60k, 80k, 100k, and 200k. The results with the MSE-based denoiser with $\gamma < 1$ are presented in Figure 6. Note that, in order for the division of samples among the first five classes to be valid, we require $\dfrac{N_{\text{train}}}{1 + \gamma} \leq 17500 \Rightarrow N_{\text{train}} \leq 17500\,(1 + \gamma)$. This shows that the point $N_{\text{train}} = 35000$ is invalid for all $\gamma < 1$. Thus, we add a sufficient amount of samples from the synthetic set CIFAR-5m to the classifier train set, both in the noisy and denoised case (where the noisy CIFAR-5m passes through the denoiser).

### C.4  TRAINING THE DENOISER WITHOUT CLEAN IMAGES

Replacing the MSE loss with Stein's Unbiased Risk Estimate (SURE) (Stein, 1981; Soltanayev & Chun, 2018) allows to train the denoiser using only noisy images. Specifically, instead of $\text{MSE}(\mathbf{z}_\theta(\mathbf{x}), \mathbf{x}_{gt}) = \|\mathbf{z}_\theta(\mathbf{x}) - \mathbf{x}_{gt}\|^2$, we use:

$$\text{SURE}(\mathbf{z}_\theta(\mathbf{x})) = \|\mathbf{z}_\theta(\mathbf{x}) - \mathbf{x}\|^2 - d\sigma^2 + 2\sigma^2 \sum_{i=1}^{d} \frac{\partial}{\partial x_i}\mathbf{z}_\theta(\mathbf{x}),$$

which obeys $\mathbb{E}[\mathrm{SURE}(\mathbf{z}_\theta(\mathbf{x}))] = \mathbb{E}[\mathrm{MSE}(\mathbf{z}_\theta(\mathbf{x}), \mathbf{x}_{gt})]$ for $\mathbf{x}|\mathbf{x}_{gt} \sim \mathcal{N}(\mathbf{x}_{gt}, \sigma^2 \mathbf{I})$. We use the common practice of approximating the divergence term with $\mathbf{g}^\top (\mathbf{z}_\theta(\mathbf{x} + \epsilon\mathbf{g}) - \mathbf{z}_\theta(\mathbf{x}))/\epsilon$, where $\epsilon$ is small and $\mathbf{g} \sim \mathcal{N}(\mathbf{0}, \mathbf{I})$ is drawn per optimizer iteration. Additionally, we use: batch size 64 and 1000 epochs; Adam optimizer; learning rate: 0.0001; learning rate decay: 0.5 at iterations 20k, 40k, 60k, 80k, 100k, and 200k.

The results for the setup with the SURE-based denoiser are presented in Figure 5. It can be seen that they resemble the results for the MSE-based denoiser, which are presented in Section 4.

## C.5 NUMERICAL ACCURACY RESULTS

In the following Tables 1 and 2 we report accuracy results related to Figure 2.

| $N_{\textbf{train}}$ | Error without denoising (%) | Error with denoising (%) |
|---|---|---|
| 1000 | 71.12 ±1.39 | 63.32 ±0.96 |
| 2000 | 64.91 ±1.44 | 58.14 ±1.41 |
| 3000 | 63.19 ±1.21 | 54.62 ±0.37 |
| 5000 | 60.43 ±1.42 | 49.79 ±1.80 |
| 10000 | 46.15 ±0.81 | 42.43 ±0.25 |
| 15000 | 42.14 ±0.8 | 39.74 ±0.40 |
| 25000 | 38.48 ±0.29 | 36.02 ±0.38 |
| 35000 | 35.77 ±0.30 | 33.83 ±0.25 |

Table 1: Classification error rates (%) on noisy and denoised CIFAR-10 images for varying training set sizes $N_{\text{train}}$. The noise level is $\sigma = 0.25$, $\gamma = 1$, and the denoiser is trained with MSE loss.

| $N_{\textbf{train}}$ | Error without denoising (%) | Error with denoising (%) |
|---|---|---|
| 1000 | 74.45 ± 0.81 | 64.82 ± 1.14 |
| 2000 | 71.01 ± 2.00 | 61.25 ± 0.90 |
| 3000 | 67.86 ± 1.46 | 58.30 ± 0.66 |
| 5000 | 64.98 ± 1.21 | 56.24 ± 2.03 |
| 10000 | 54.70 ± 1.14 | 49.64 ± 0.50 |
| 15000 | 50.30 ± 0.47 | 47.34 ± 0.41 |
| 25000 | 47.65 ± 0.37 | 45.14 ± 0.43 |
| 35000 | 45.40 ± 0.33 | 43.21 ± 0.20 |

Table 2: Classification error rates (%) on noisy and denoised CIFAR-10 images for varying training set sizes $N_{\text{train}}$. The noise level is $\sigma = 0.4$, $\gamma = 1$, and the denoiser is trained using MSE loss.

Figure 4 shows the classification error vs. the training epoch in a single trial for noise level 0.25, $\gamma = 1$ and 35,000 training images. It demonstrates that the classifier does not suffer from overfitting.

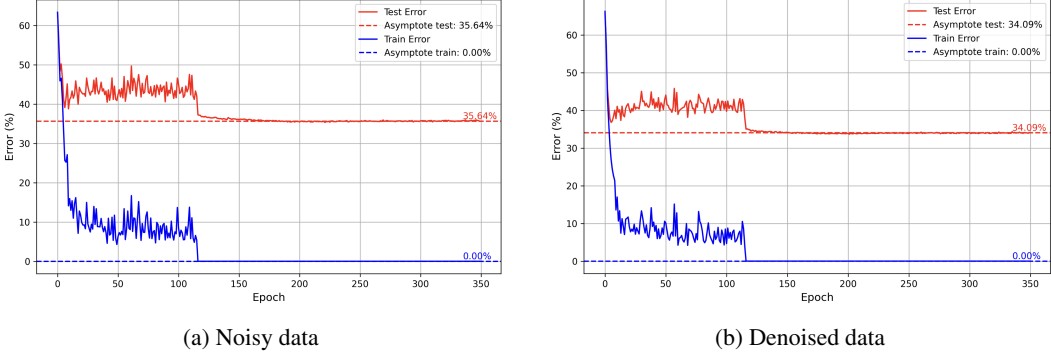

(a) Noisy data

(b) Denoised data

Figure 4: Training and testing error as a function of epochs for (a) noisy data and (b) denoised data. The noise level is $\sigma = 0.25$, $\gamma = 1$, and $N_{\text{train}} = 35,000$. The denoiser is trained using MSE loss.

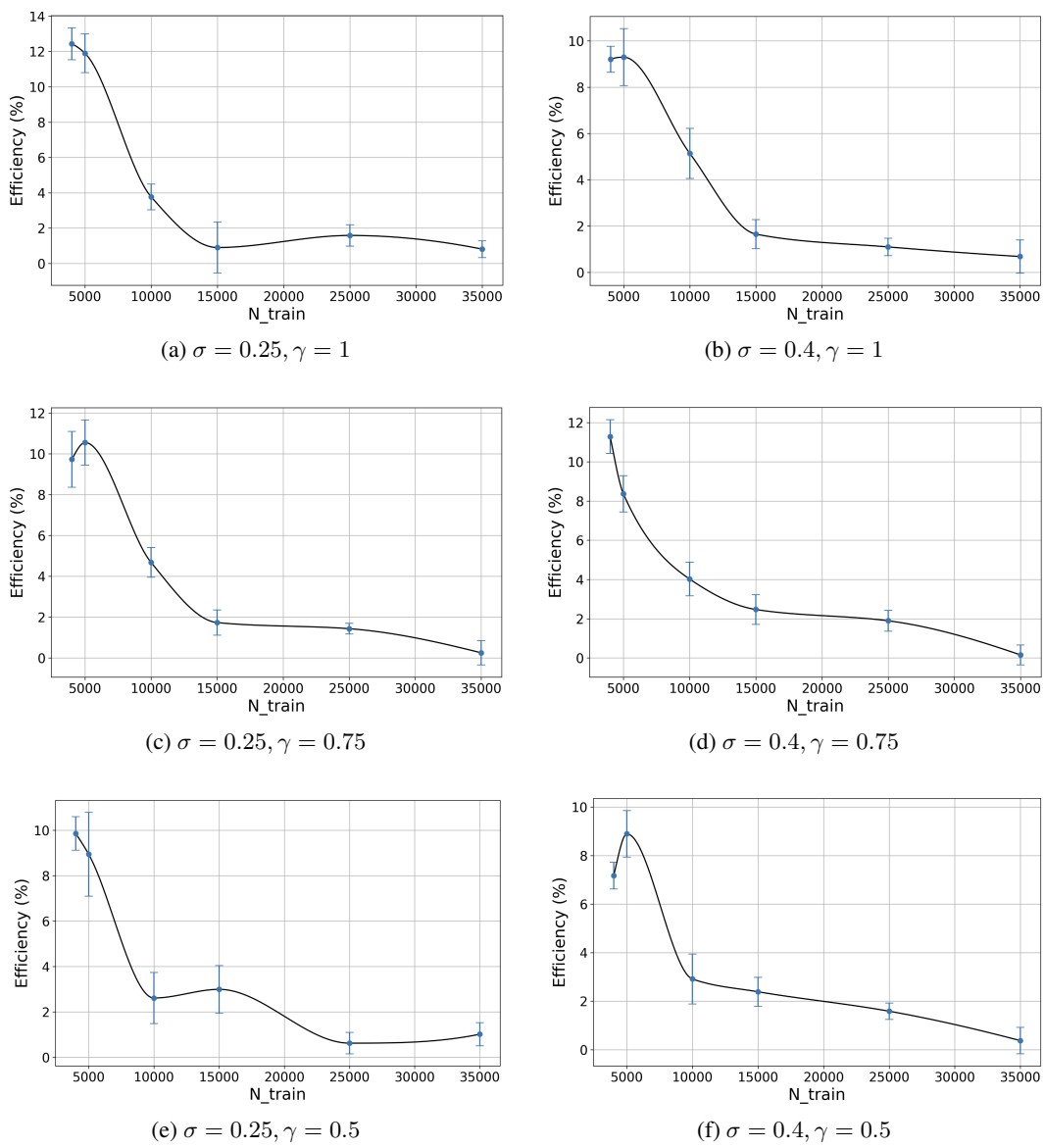

Figure 5: Practical deep learning setup with noisy CIFAR-10 and SURE-based denoiser. Efficiency of the data processing procedure versus the number of training samples for various values of the training imbalance factor, $\gamma$, and the standard deviation of the noise, $\sigma$.

# D  ADDITIONAL EMPIRICAL DETAILS AND RESULTS (MINI-IMAGENET, ENCODING)

## D.1  EXPERIMENTS COMPUTE RESOURCES

We conducted our experiments using 16 NVIDIA Tesla V100-SXM2 GPUs with 32GB memory, 12 NVIDIA RTX 6000 Ada Generation GPUs with 48GB memory, and 2 NVIDIA A100 PCIe GPUs with 80GB memory. The training time for each data point in Figures 3 and 7 ranged from 10 hours to 30 hours, depending on the number of training samples.

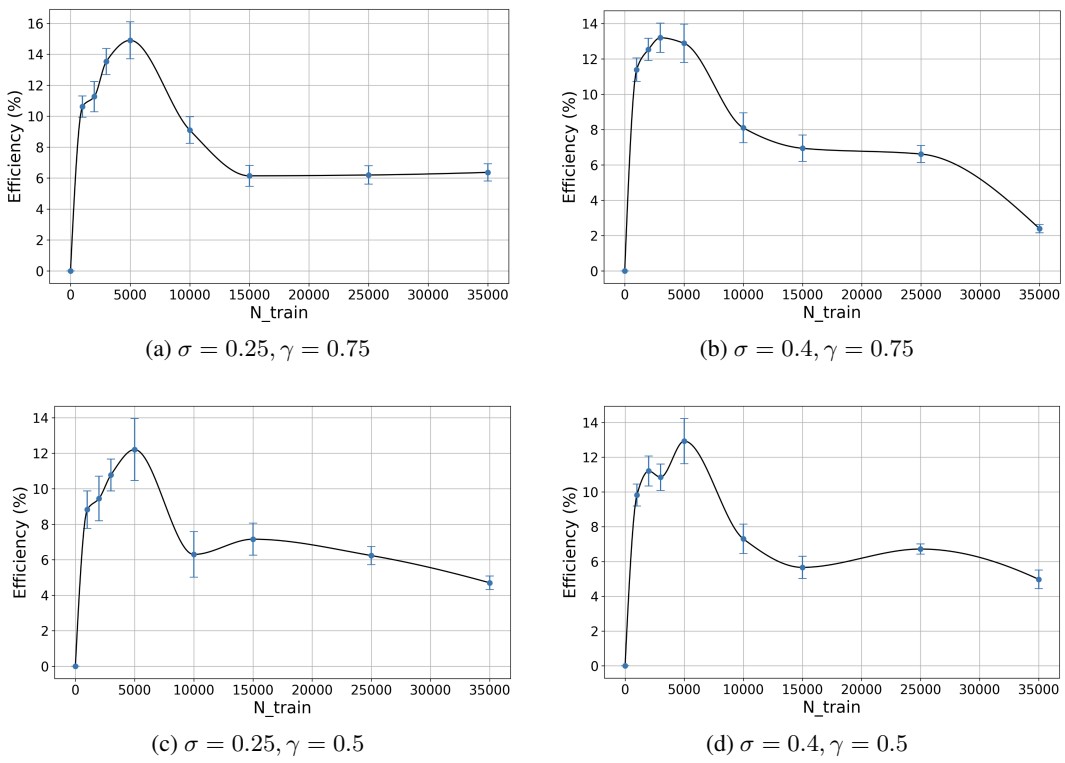

Figure 6: Practical deep learning setup with noisy CIFAR-10 and MSE-based denoiser. Efficiency of the data processing procedure versus the number of training samples for various values of the training imbalance factor, $\gamma \in \{0.5, 0.75\}$, and the standard deviation of the noise, $\sigma$.

## D.2 TRAINING THE CLASSIFIER

We consider the Mini-ImageNet dataset and the ResNet50 model. To train the model, we use: batch size 128 and 225 epochs; cross-entropy loss; SGD optimizer; learning rate: 0.0679; learning rate decay: 0.1 at epochs 75 and 150; momentum: 0.9; weight decay: 0.0005. This setting yields 73% accuracy for clean data.

Per noise level $\sigma \in \{\frac{50}{255}, \frac{100}{255}\}$ of the additive Gaussian noise that has been added to the data, we use this setting to train one classifier that operates directly on the noisy data.

## D.3 TRAINING THE ENCODER

For self-supervised learning, we adopt the DINOv2 framework (Lu et al., 2025). The student encoder is a Vision Transformer (ViT-S/16), which splits each input image of size 224×224 into 16×16 patches and produces a 384-dimensional [CLS] token representation. This is passed through a 3-layer MLP projection head to produce the final 256-dimensional embedding ($\mathbf{z} \in \mathbb{R}^{256}$), which is used for self-supervised training. The teacher network has the same architecture and is updated as an exponential moving average of the student, providing stable target embeddings. Training is performed on the Mini-ImageNet dataset for 200 epochs with a per-GPU batch size of 40. We apply the AdamW optimizer with a base learning rate of 0.004 (scaled with the square root of the effective batch size), $\beta = (0.9, 0.999)$, weight decay scheduled from 0.04 to 0.4, and gradient clipping at 3.0. The teacher momentum is linearly increased from 0.992 to 1.0 over training. Multi-crop augmentation is employed with 2 global crops of size 224×224 and 8 local crops of size 96×96. Model evaluation is conducted every 6,250 iterations.

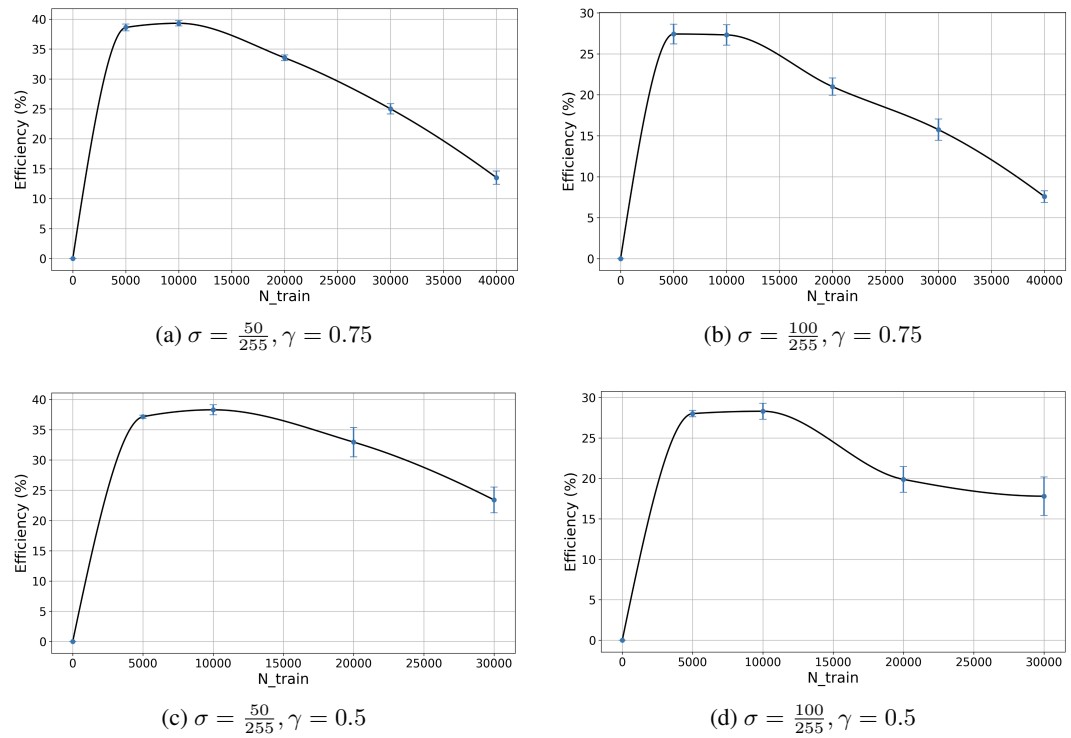

(a) $\sigma = \frac{50}{255}, \gamma = 0.75$        (b) $\sigma = \frac{100}{255}, \gamma = 0.75$

(c) $\sigma = \frac{50}{255}, \gamma = 0.5$        (d) $\sigma = \frac{100}{255}, \gamma = 0.5$

Figure 7: Practical deep learning setup with noisy Mini-ImageNet and pre-classification encoding. Efficiency of the data processing procedure versus the number of training samples for various values of the training imbalance factor, $\gamma \in \{0.5, 0.75\}$, and the standard deviation of the noise, $\sigma$.

### D.4 TRAINING AN MLP ON TOP OF THE EMBEDDINGS

Per noise level $\sigma \in \{\frac{50}{255}, \frac{100}{255}\}$, after training the DINOv2 encoder, we pass the noisy Mini-ImageNet images through the encoder to obtain 256-dimensional embeddings. On top of these embeddings, we train a multi-layer perceptron (MLP) classifier to perform image classification. The MLP consists of three hidden layers with dimensions 4096, 2048, and 1024, each followed by LayerNorm and GELU activation, and a final linear layer mapping to the number of classes (i.e. 100). Hidden layers are initialized with Xavier uniform, and the final layer with a small normal distribution.

To train the model, we use: per-GPU batch size 128 and 20 epochs, with 1250 iterations per epoch; cross-entropy loss; SGD optimizer with a cosine annealing learning rate schedule; momentum: 0.9; no weight decay. Linear evaluation is performed with periodic check-pointing and evaluation on the validation set. After training, the classifier is evaluated on the test set to report final accuracy.

The results for the setup with $\gamma < 1$ are presented in Figure 7. It can be seen that they resemble the results in 3: 1) similar non-monotonicity of the curve while remaining positive, and 2) the maximal efficiency increases with the SNR, for fixed $\gamma$.

### D.5 NUMERICAL ACCURACY RESULTS

In the following Tables 3 and 4 we report accuracy results related to Figure 3. Lastly, in Figure 8 we present an image of noisy data for $\sigma \in \{50/255, 100/255\}$, to show that the noise is not too-severe.

## E ADDITIONAL EMPIRICAL DETAILS AND RESULTS (CIFAR-10, ENCODING)

We investigate the CIFAR-10 dataset and the ResNet18 model. Both the training and test sets are subjected to additive Gaussian noise with standard deviations $\sigma \in \{0.25, 0.4\}$. This time, as the

| $N_{\text{train}}$ | Error without encoding (%) | Error with encoding (%) |
|---|---|---|
| 5000 | 80.03 ±1.25 | 48.67 ±0.48 |
| 10000 | 74.56 ±1.12 | 45.03 ±0.27 |
| 20000 | 58.81 ±0.45 | 41.45 ±0.29 |
| 30000 | 49.92 ±0.8 | 39.23 ±0.39 |
| 40000 | 44.95 ±1.37 | 37.99 ±0.31 |
| 50000 | 40.07 ±0.58 | 36.62 ±0.25 |

Table 3: Classification error rates (%) on noisy and encoded Mini-ImageNet images for varying training set sizes $N_{\text{train}}$. The noise level is $\sigma = \frac{50}{255}, \gamma = 1$.

| $N_{\text{train}}$ | Error without encoding (%) | Error with encoding (%) |
|---|---|---|
| 5000 | 85.5 ±0.95 | 60.94 ±0.25 |
| 10000 | 79.71 ±1.73 | 57.28 ±0.23 |
| 20000 | 68.47 ±2.15 | 53 ±0.27 |
| 30000 | 58.39 ±1.77 | 50.92 ±0.31 |
| 40000 | 54.62 ±0.08 | 49.35 ±0.34 |
| 50000 | 50.01 ±0.56 | 48.23 ±0.27 |

Table 4: Classification error rates (%) on noisy and encoded Mini-ImageNet images for varying training set sizes $N_{\text{train}}$. The noise level is $\sigma = \frac{100}{255}, \gamma = 1$.

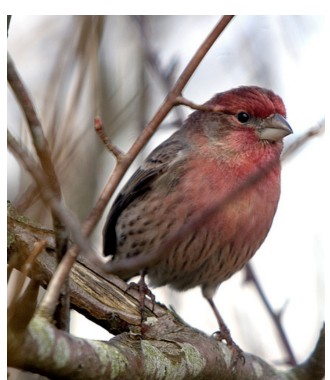

(a) clean

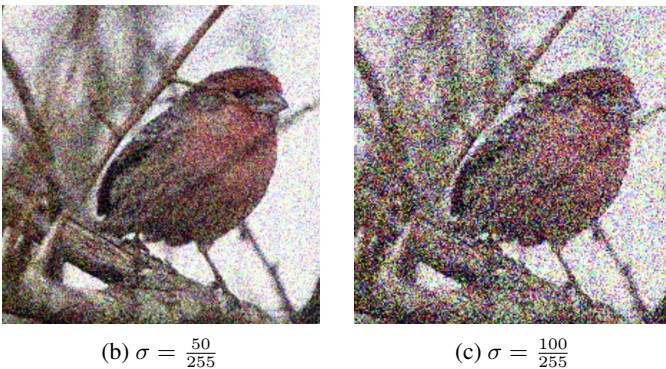

(b) $\sigma = \frac{50}{255}$        (c) $\sigma = \frac{100}{255}$

Figure 8: Clean and noisy Mini-ImageNet images. (a) Clean image. (b) Image with Gaussian noise $\sigma = \frac{50}{255}$. (c) Image with Gaussian noise $\sigma = \frac{100}{255}$.

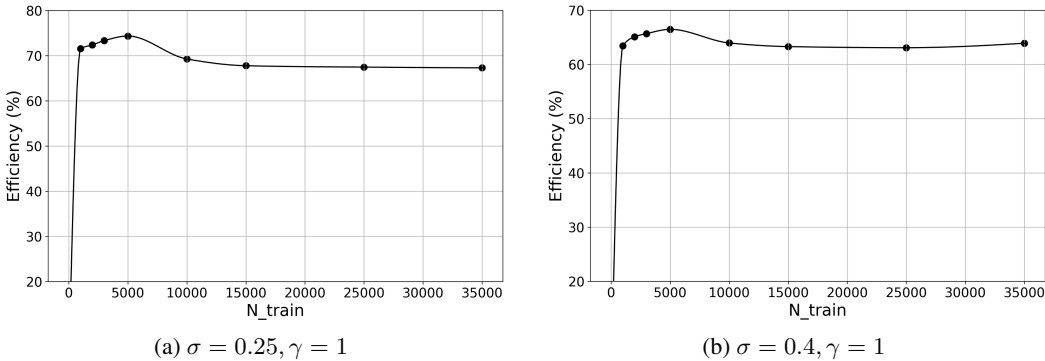

(a) $\sigma = 0.25, \gamma = 1$            (b) $\sigma = 0.4, \gamma = 1$

Figure 9: Noisy CIFAR-10 and pre-classification encoding. Efficiency versus $N_{\text{train}}$.

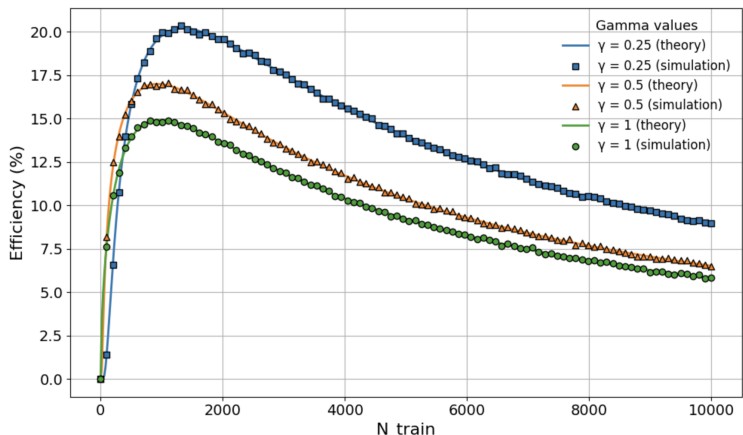

Figure 10: The theoretical setup. Efficiency of the data processing procedure versus the number of training samples $N_{\text{train}}$, for various values of the training imbalance factor $\gamma$, and SNR of $\mathcal{S} = 1$.

data processing procedure we use an encoding step that maps each image (rescaled from its original CIFAR-10 resolution to $224 \times 224$) into a 256-dimensional embedding. This encoder model follows (Lu et al., 2025) and is trained from scratch with self-supervision on $45000$ noisy unlabeled images for each noise level. Then, for each combination of $(\sigma, N_{\text{train}})$, considering the balanced case of $\gamma = 1$, we divide $N_{\text{train}}$ equally among all 10 classes. Then, we train a ResNet18 model on the noisy images across 6 seeds and, in parallel, a small MLP on the corresponding embeddings across 3 seeds. After we have the mean of the probability of error before and after the data processing, we compute the empirical efficiency, i.e., the relative percentage change in the probability of error induced by the encoding step. Details of the training procedures for the ResNet18 and the MLP are provided in Appendix C and D, respectively.

Figure 9 presents the efficiency versus $N_{\text{train}}$, for $\gamma = 1$. We see the same trends that are aligned with our theory as before: 1) similar non-monotonicity of the curve (increase to a maximal efficiency value and then decrease) while remaining positive, and 2) the maximal efficiency increases with the SNR.

## F    EXTENDED EMPIRICAL VERIFICATION

In this section, we extend our empirical verification. In Figure 10, we simulate the theoretical setup, as in 3.3 (we use $d = 2000, k = 1000$ and $\sigma = 1$), but with $\mathcal{S} = 1$. We see that the empirical efficiency coincides with the theoretical efficiency.

We now examine the effect of $\mathcal{S}$ on efficiency. We fix $\gamma = 1, d = 2000, k = 1000$ and vary $\mathcal{S} \in \{0.5^2, 1, 1.5^2\}$. The results are presented in Figure 11. Let us discuss the results. We see that for $N_{\text{train}} \gg 1$ (in the right Figure), for larger SNR (lower noise level), the efficiency decreases. However,

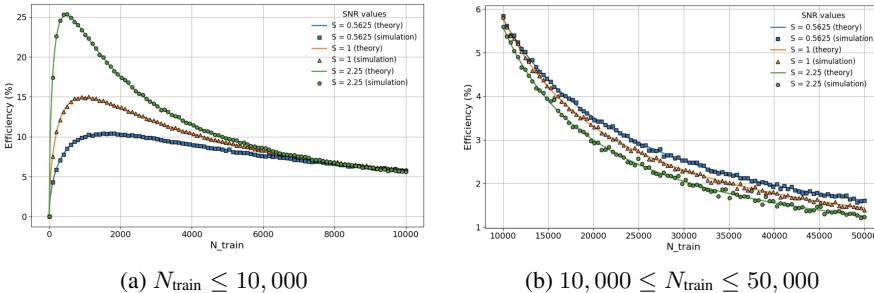

(a) $N_{\text{train}} \le 10,000$           (b) $10,000 \le N_{\text{train}} \le 50,000$

Figure 11: Extended simulation of the theoretical setup. Efficiency of the data processing versus the number of training samples $N_{\text{train}}$, for $\gamma = 1$, and various values of the SNR, $\mathcal{S}$, for (a) low samples regime, and (b) high samples regime.

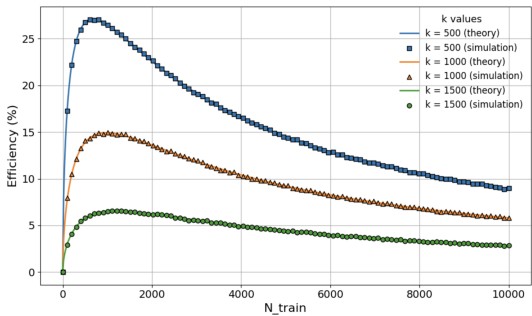

Figure 12: The theoretical setup. Efficiency of the data processing procedure versus the number of training samples $N_{\text{train}}$, for various values of the training imbalance factor, $\gamma$, and SNR of $\mathcal{S} = 1$.

as 8 suggests, when the number of samples is limited, and for larger SNR (lower noise level), the efficiency increases. We notice this phenomenon in the left Figure, presenting low $N_{\text{train}}$, compared to the right Figure. First, the efficiency increases with the SNRs, and then as $N_{\text{train}}$ gets larger, the dependency flips. We also see a different behavior that sheds more light on this conclusion: the difference between different SNRs is larger in the low-samples region than the inverse relation in the high-samples region. This concludes the non-monotonic and non-intuitive dependency of the efficiency on the SNR.

In addition, we examine the effect of $d - k$ on efficiency. We fix $\gamma = 1, d = 2000, \mathcal{S} = 1$ and vary $k \in \{500, 1000, 1500\}$. The results are presented in Figure 12. We see that larger $d - k$ corresponds to greater accuracy. Theorem 7 proves this for $N_{\text{train}} \gg 1$, but we see that this is true even for small $N_{\text{train}}$. Indeed, intuitively, reducing more dimensions is advantageous in terms of efficiency. This suggests that there is a direct monotonic relationship between the efficiency and $d - k$.

We now consider the same setting as in the empirical verification ($d = 2000$, $k = 1000$, $\sigma = 1$, $\gamma \in \{0.25, 0.5, 1\}$, $\mathcal{S} \in \{0.75^2, 1, 1.5^2\}$), but per $N_{\text{train}}$, the data processing matrix $\boldsymbol{A}$ is learned from 50,000 unlabeled samples using the algorithm described in the proof of Theorem 3. The corresponding results are shown in Figure 13, demonstrating the same trends as the theoretical efficiency. Moreover, as the number of unlabeled samples tends to infinity, the two curves coincide. To illustrate this, we also present results for the case that per $N_{\text{train}}$, the data processing matrix $\boldsymbol{A}$ is learned from 5,000,000 unlabeled samples in Figure 14. Notice that as the amount of unlabeled samples available grows, the gap between the theoretical efficiency and the empirical efficiency is reduced.

Finally, we visualize the action of $\boldsymbol{A} : \mathbb{R}^2 \to \mathbb{R}$ on the GMM data in Figure 15. While our analysis considers the regime $d > k \gg 1$, as is common in practice, we use small values of $d = 2$ and $k = 1$ to enable visualization.

Recall that the plug-in classifier, before and after the data processing, depends only on the distance of a test sample from each of the empirical means. Without processing, these empirical means are given by $\widehat{\boldsymbol{\mu}}_j = \frac{1}{N_j} \sum_{i=1}^{N_j} \boldsymbol{x}_{i,j}$ and hence distributed as $\widehat{\boldsymbol{\mu}}_j \sim \mathcal{N}\left(\boldsymbol{\mu}_j, \frac{\sigma^2}{N_j} \boldsymbol{I}_d\right)$. Similarly, after

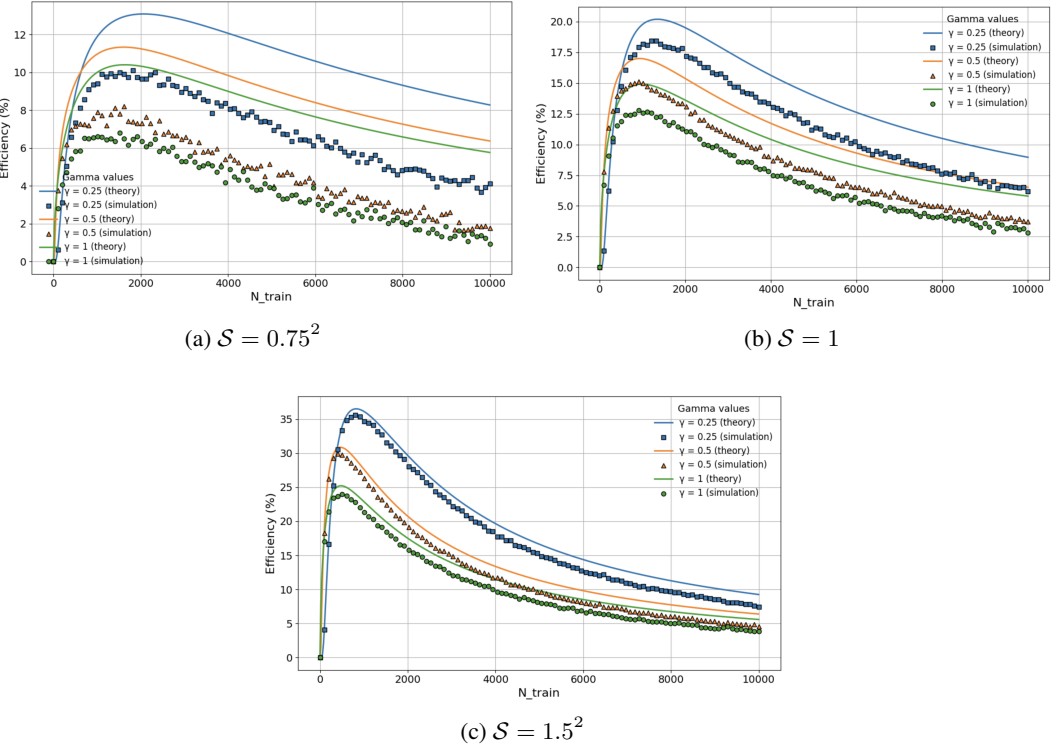

Figure 13: Extended empirical verification - per $N_{\text{train}}$, $\boldsymbol{A}$ is learned from 50,000 unlabeled examples. Presented for (a) $\mathcal{S} = 0.75^2$, (b) $\mathcal{S} = 1$ and (c) $\mathcal{S} = 1.5^2$.

processing by $\boldsymbol{A} : \mathbb{R}^d \to \mathbb{R}^k$, the empirical means obey $\boldsymbol{A}\widehat{\boldsymbol{\mu}}_j \sim \mathcal{N}\left(\boldsymbol{A}\boldsymbol{\mu}_j, \frac{\sigma^2}{N_j}\boldsymbol{I}_k\right)$, where the semi-orthonormality $\boldsymbol{A}\boldsymbol{A}^\top = \boldsymbol{I}_k$ is used. Consequently,

$$\mathbb{E}\left[\|\widehat{\boldsymbol{\mu}}_j - \boldsymbol{\mu}_j\|^2\right] = \sum_{i=1}^d \mathbb{E}\left[([\widehat{\boldsymbol{\mu}}_j]_i - [\boldsymbol{\mu}_j]_i)^2\right] = \sum_{i=1}^d \frac{\sigma^2}{N_j} = \frac{\sigma^2}{N_j}d$$

where we used $[\widehat{\boldsymbol{\mu}}_j]_i - [\boldsymbol{\mu}_j]_i \sim \mathcal{N}\left(0, \frac{\sigma^2}{N_j}\right)$. Similarly, $\mathbb{E}\left[\|\boldsymbol{A}\widehat{\boldsymbol{\mu}}_j - \boldsymbol{A}\boldsymbol{\mu}_j\|^2\right] = \frac{\sigma^2}{N_j}k$.

The data processing lowers the dimension from $d$ to $k$, and thus improves the average squared error of the mean estimator by

$$\frac{\sigma^2}{N}(d - k) > 0.$$

Since the classifier, before and after the data processing, depends only on the distance of the test sample from each of the empirical means, its accuracy increases when the accuracy of the empirical means improves while the distance between the means of the difference classes does not significantly reduce (i.e., $\|\boldsymbol{A}\widehat{\boldsymbol{\mu}}_2 - \boldsymbol{A}\widehat{\boldsymbol{\mu}}_1\| \approx \|\widehat{\boldsymbol{\mu}}_2 - \widehat{\boldsymbol{\mu}}_1\|$). The latter is accounted for by the property $\|\boldsymbol{A}\boldsymbol{\mu}\| = \|\boldsymbol{\mu}\|$ of the operator.

Note that this behavior is observed in Figure 15:

- In the red class, the distance between the empirical mean and the real mean is 0.4632 before applying $\boldsymbol{A}$ and 0.39 after applying $\boldsymbol{A}$.

- In the blue class, the distance between the empirical mean and the real mean is 0.439 before applying $\boldsymbol{A}$ and 0.38 after applying $\boldsymbol{A}$.

- The distance between the empirical means of the different classes is 4.0374 before applying $\boldsymbol{A}$ and 4.01 after applying $\boldsymbol{A}$. Indeed, both are close to the distance between the real means, which is 4.

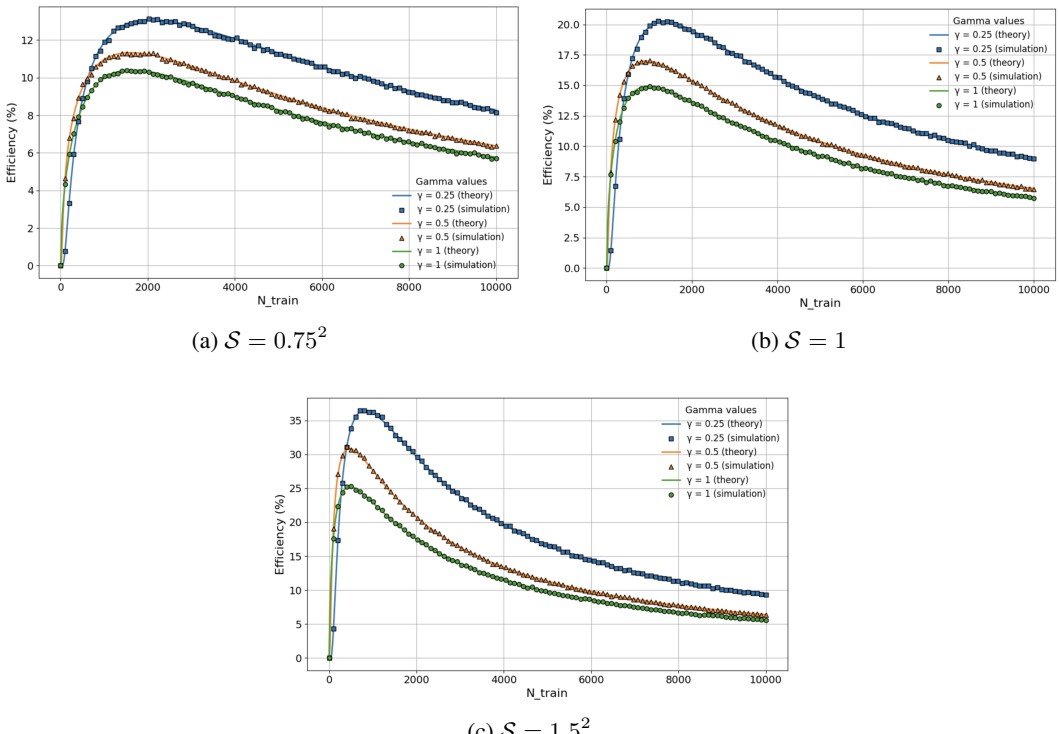

(a) $\mathcal{S} = 0.75^2$

(b) $\mathcal{S} = 1$

(c) $\mathcal{S} = 1.5^2$

Figure 14: Extended empirical verification - per $N_{\text{train}}$, $\boldsymbol{A}$ is learned from 5,000,000 unlabeled examples. Presented for (a) $\mathcal{S} = 0.75^2$, (b) $\mathcal{S} = 1$ and (c) $\mathcal{S} = 1.5^2$.

That is, $\boldsymbol{A}$ preserves the separation quality of the classes, while improving the estimation quality of $\boldsymbol{\mu}$.

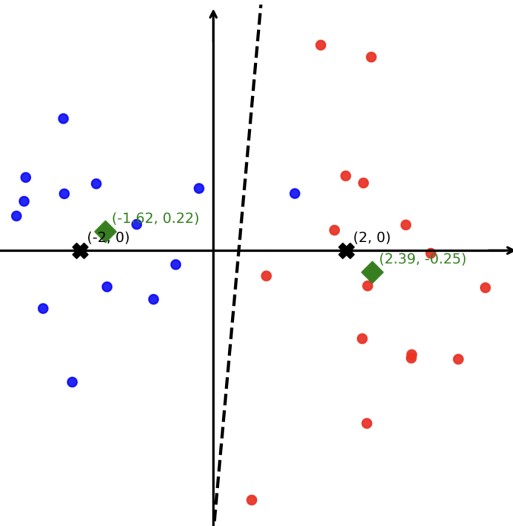

(a) The original data, sampled from GMM in $\mathbb{R}^2$ with $\boldsymbol{\mu}_1 = -\boldsymbol{\mu}_2 = (2,0)^\top$, $\sigma = 1$, and $N_{\text{train}} = 30$. The true means $\{\boldsymbol{\mu}_1, \boldsymbol{\mu}_2\}$ are marked by black 'X's and the empirical means $\{\hat{\boldsymbol{\mu}}_1, \hat{\boldsymbol{\mu}}_2\}$ are marked by green diamonds. The learned decision boundary is marked by the dashed line (determined by the distance to the empirical means).

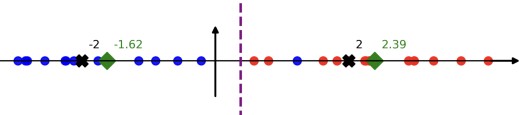

(b) The data after applying $\boldsymbol{A} \in \mathbb{R}^{1 \times 2}$ (in this case, projection onto the x-axis). As before, the true means $\{\boldsymbol{A}\boldsymbol{\mu}_1, \boldsymbol{A}\boldsymbol{\mu}_2\}$ are marked by black 'X's and the empirical means $\{\boldsymbol{A}\hat{\boldsymbol{\mu}}_1, \boldsymbol{A}\hat{\boldsymbol{\mu}}_2\}$ are marked by green diamonds.

Figure 15: Visualization of the effect of $\boldsymbol{A} : \mathbb{R}^2 \to \mathbb{R}$. Note that the empirical means (green diamonds) are closer to the true means (black 'X's) after the operation $\boldsymbol{A}$. The distance between the empirical means of the different classes remains similar.

