# OpenReview forum: "Does the Data Processing Inequality Reflect Practice? On the Utility of Low-Level Tasks"
_ICLR.cc/2026/Conference — ICLR 2026 Poster_

### Official Review · Reviewer_PqbN · 2025-10-29

**Soundness:** 3
**Presentation:** 2
**Contribution:** 3
**Rating:** 6
**Confidence:** 4

**Summary:**

This paper investigates when and why low-level data preprocessing can be beneficial for a strong classifier (e.g., one that converges to the optimal Bayes classifier). The authors first prove the existence of a preprocessing (dimension reduction) procedure that can improve classification performance. They then establish theoretical results illustrating the influence of various factors, such as the number of training samples and the degree of class separation. The analysis is further supported by both simulation studies and real-data experiments.

**Strengths:**

1. The problem studied in this paper is interesting and important.
2. The theoretical analysis is solid.
3. The authors consider various scenarios and influencing factors in their theoretical derivations.

**Weaknesses:**

1. The writing can be improved. For example, Section 3.2 discusses multiple scenarios; it would be helpful to outline them at the beginning of the section. In addition, providing some intuition behind the theorems would improve readability and accessibility.
2. Some theorems could potentially be extended (see the question below).
3. The data processing analyzed in this paper can restrictive (see the question below).

**Questions:**

1. Is the data processing step restricted to the dimension reduction procedure defined in Equation (7)?
2. In Theorems 5 and 6, is it possible to characterize how $\hat{p}_x$ compares to $\hat{p}_z$? Specifically, does the difference between them vary with certain parameters?

---

> ### Author Response · Authors · 2025-11-24
>
> We thank the reviewer for their feedback and are pleased that they appreciate the motivation of our work and the potential of the paper to open a new line of research on this topic. We address their concerns below.
>
> &nbsp;
>
> **W1 (Writing):**
>
> We thank the reviewer for this comment and have followed their suggestions to improve the readability of Section 3.2 in the revision.
> Specifically, we split Section 3.2 into subsection 3.2.1, where we prove that the error probability decreases due to the data processing, and subsection 3.2.2, where we provide a fine-grained analysis of the factors that affect the margin and maximal gain due to the data processing.
> We begin the section with an outline of the setup and achieved results. We also polished and extended discussions below several theorems, aiming to provide more intuition than in the original version of the paper.
>
> &nbsp;
>
> **W2, Q2 (Extension of theorems):**
>
> Please note that our Theorem 7 already analyzes the effects of different parameters on the relative margin between $\hat{p}_x$ and $\hat{p}_z$ (i.e., on $\eta=(\hat{p}_x - \hat{p}_z)/\hat{p}_x \times 100$.
>
> Following the reviewer's comment we add Appendix B in the revision, where we present and prove Theorem 10, which analyzes the effects of the different factors on the direct difference $\Delta = \hat{p}_x - \hat{p}_z$ for large $N$.
> The insights are aligned with those that are gained from Theorem 7.
>
> &nbsp;
>
> **W3, Q1 (Regarding the theoretical data processing):**
>
> The data processing considered in our theoretical setup is adopted because it makes the problem tractable, allowing us to establish rigorous theory and gain mathematically-backed insights.
> As can be seen in the proofs in the appendix, even with this choice of data processing the analysis remains challenging.
>
> Note that the linear low-level processing in the theoretical setup has proved to be sufficient for establishing multiple novel results, among them: (1) formally proving, for the first time, a gap between data processing inequality and practice: the existence of a low-level processing that improves classification performance for any finite number of training samples; and (2) establishing insights and surprising behaviors (e.g., maximal relative gain for high SNR) that are empirically observed in practical deep learning setups.
>
> Regarding (1), obtaining this fundamental result by a relatively simple and learnable dimension reduction procedure implies potential gain from more sophisticated procedures. Thus, this is not a limitation of our theory.
>
> Regarding (2), the alignment between the behavior in practical deep learning setups, with two different data processing techniques: denoising and encoding, demonstrates that the insights predicted under our theoretical setup generalize beyond the data processing considered in our theory.
>
> Please note that in the revision we add experiments for another practical setup (specifically, noisy CIFAR-10 setup with pre-classification encoding in Subsection 4.3 and Appendix E), where the behavior aligns with the theoretical insights.
>
> &nbsp;
>
> We hope that our response addresses all your concerns, and we would be happy to further clarify open issues.

---

> ### Author Response · Authors · 2025-11-28
>
> Dear Reviewer, we sincerely appreciate your time and valuable comments. Due to the limited discussion time, we would be grateful for any additional questions or confirmation that our response has addressed your concerns. Please note that, along with our response, we have also updated the paper according to the comments, with the changes marked in blue.

---

### Official Review · Reviewer_yFZ5 · 2025-10-29

**Soundness:** 3
**Presentation:** 2
**Contribution:** 2
**Rating:** 4
**Confidence:** 2

**Summary:**

Consider the binary classification problem.
It is a general belief that preprocessing data makes the performance of the Bayes classifier worse assuming that we know the ground truth distribution.
We would like to ask a question: If we use a practical classifier and we do not know the ground truth distribution, does the conclusion of the performance of the practical classifier getting worse still hold?
In this paper, the authors study this question and give a negative answer.
The authors consider the following setting.
Suppose the underlying distribution is the mixture of two equally weighted standard Gaussians with symmetric means.
Each sample from the training set is drawn from the distribution with a label indicating which Gaussian it is drawn from and the estimated mean for each Gaussian is the simple averaging of the corresponding samples.
Each sample from the test set is also drawn from the distribution without a label.
The classifier is to classify the sample to be the Gaussian whose estimated mean is closer to the sample.
The preprocessing is a linear transformation whose matrix is semi-orthogonal and norm preserving.
The result shows that the error for the preprocessing data can be smaller than the error for the original data.

**Strengths:**

- The problem is well-motivated.
The result may open a new line of work on this topic.

**Weaknesses:**

- Though the authors provide some experimental results, the theoretical explanation is based on a simplified setup.
In other words, the theory-practice gap is large that there may be a limited interpretation when extending to more realistic settings.

**Questions:**

.

---

> ### Author Response · Authors · 2025-11-24
>
> We thank the reviewer for their feedback and are pleased that they appreciate the motivation of our work and the potential of the paper to open a new line of research on this topic. We address their concerns below.
>
> &nbsp;
>
> **W1 (Theory-practice gap):**
>
> Please note that currently there are no available mathematical tools to rigorously analyze practical deep learning models and real-world datasets. Therefore, restricting the theoretical analysis to simplified models is inevitable.
>
> Regarding the data distribution model, as discussed in our paper, the Gaussian Mixture Model (GMM) is a standard and widely used data model in theoretical works on classification. See, e.g., the works that are cited in our paper (Cao et al., 2021; Deng et al., 2022; Wang \& Thrampoulidis, 2022; Kothapalli \& Tirer, 2025). Moreover, unlike some prior works, we do not restrict the SNR (the ratio between the squared norm of the mean and the variance).
>
> Regarding the classifier model, please note the motivation stated in the paper below Eq. 6.
> We aim to explore whether data processing can be beneficial even for a "strong" classifier. And the considered classifier is indeed strong: not only does it converge to the optimal Bayes classifier as the number of training samples grows, but for any finite number of training samples the estimated means used by this classifier are optimal estimators (attain the Cramer–Rao lower bound on the variance) of the true means that are used in the optimal Bayes classifier.
> Hence, demonstrating the benefit of low-level processing for such a classifier, which is "almost optimal" for the considered setup, underscores the potential advantages for weaker classifiers.
>
> Regarding the consideration of a simplified processing model (a linear operation rather than, e.g., a complex neural network), note that any existing deep learning theory work (which attempts to analyze phenomena that are observed in deep learning), is based on simplified models.
> Just to name a couple of examples out of many: the analysis of ``neural collapse'' in [1] is essentially based on a matrix factorization model that ignores the data, and the analysis of overparameterization in [2] omits all the nonlinear activation functions from the model (so their model is linear in the input).
>
> [1] Zhu et al., "A Geometric Analysis of Neural Collapse with Unconstrained Features", NeurIPS 2021.
> [2] Arora et al., "On the Optimization of Deep Networks: Implicit Acceleration by Overparameterization", ICML 2018.
>
> Therefore, it is reasonable to assume linear data processing for establishing rigorous theory and gain mathematically-backed insights.
> As can be seen in the proofs in the appendix, even for this linear data processing the analysis is challenging.
>
> Importantly,
> the linear low-level processing in the theoretical setup has proved to be sufficient for establishing multiple novel results, among them: (1) formally proving, for the first time, a gap between data processing inequality and practice: the existence of a learnable low-level processing that improves classification performance for any finite number of training samples; and (2) establishing insights and surprising behaviors (e.g., maximal relative gain for high SNR) that are empirically observed in practical deep learning setups.
>
> Regarding (1), obtaining this fundamental result by a relatively simple and learnable dimension reduction procedure is not a limitation of our theory, as it implies that more sophisticated procedures can also possess this advantage.
>
> Regarding (2), the alignment between the behavior in practical deep learning setups, with two different DNN-based data processing techniques: denoising and encoding, demonstrates that the insights predicted under our theoretical setup generalize beyond the data processing considered in our theory to practical deep learning settings.
>
> Please note that in the revision we add experiments for another practical setup (specifically, noisy CIFAR-10 setup with pre-classification encoding in Subsection 4.3 and Appendix E), where the behavior aligns with the theoretical insights.
>
> &nbsp;
>
> We hope that our response addresses your concerns, and we would be happy to further clarify open issues.
> If there are no additional concerns, and given your statement that: "the result may open a new line of work on this topic", we sincerely hope that you will reconsider your evaluation and possibly increase the score.

---

> ### Author Response · Authors · 2025-11-28
>
> Dear Reviewer, we sincerely appreciate your time and valuable comments. Due to the limited discussion time, we would be grateful for any additional questions or confirmation that our response has addressed your concerns. Please note that, along with our response, we have also updated the paper according to the comments, with the changes marked in blue.

---

### Official Review · Reviewer_p7cK · 2025-11-06

**Soundness:** 3
**Presentation:** 4
**Contribution:** 2
**Rating:** 6
**Confidence:** 3

**Summary:**

This paper investigates theoretical perspectives on the data processing inequality in classification, suggesting that appropriate preprocessing can improve finite-sample performance.

**Strengths:**

1. The paper studies an interesting question: whether the data processing inequality reflects practical deep learning behavior and connects a foundational information-theoretic concept to empirical DNN practice.

2. The theoretical derivation is mathematically sound under its assumptions, offering a clean proof for the potential benefit of preprocessing under finite-sample conditions.

3. The paper is well-written and visually clear, with carefully structured derivations and supporting empirical examples.

**Weaknesses:**

1. The theoretical model y→x→z implies a generative process, but deep models for classification follow a discriminative direction
x→z→ $\hat{y} $. This mismatch weakens the claimed alignment between the theoretical framework and modern deep learning models.

2. Eq. (3) assumes a Gaussian Mixture Model where data 𝑥 is generated conditional on label y. This assumption is conceptually inverted from real-world classification, where 𝑥 precedes 𝑦.

3. The theory defines 𝐴 as a linear transformation matrix, while experiments use nonlinear modules (DnCNN, Lu et al., 2025). The inconsistency makes it unclear whether the empirical findings genuinely validate the theory.

4. Although 𝐴 is constructively derived, the paper lacks intuitive explanation of how 𝐴 preserves or amplifies discriminative information. Visualizing its action in feature space or relating it to known transformations would strengthen the conceptual link between theory and practice.

5. The preprocessing step 𝐴𝑥 resembles data augmentation. If its primary function is to improve robustness via representation smoothing, the novelty may be overstated. Clarification is needed on whether the proposed framework represents a fundamentally new mechanism.

6. The experiments inject noise into data and then apply denoising, creating a circular setup that demonstrates improvement. This does not convincingly validate the theoretical claims or show robustness in realistic settings.

7. CIFAR-10 uses denoising (DnCNN), while miniImageNet uses encoding (Lu et al., 2025). Why not keep the processing method consistent, and how is the appropriate process determined for different datasets? The theory treats the processing step
𝐴 as a general transformation matrix and is not dataset-specific.

Minor:

1. The experiments on small datasets such as CIFAR-10 and miniImageNet, and ResNet only, are not enough to convincingly support the theoretical claims. Evaluating on larger-scale datasets and other architectures, such as ViT, is encouraged. Though the paper emphasizes theoretical contributions, demonstrating practical benefits is still necessary to substantiate its real-world relevance.

2. Can you provide the source code for Figure 1? This will help me better understand the connection between the theoretical derivation and its empirical illustration.

**Questions:**

Please see weakness.

---

> ### Author Response · Authors · 2025-11-24
>
> We thank the reviewer for their thorough review and valuable suggestions,
> and are pleased that they appreciate the motivation and thoroughness of the theoretical contribution, as well as the writing and the supporting experiments.
> We address their concerns below.
>
> &nbsp;
>
> **W1 (The Markov chain $y \rightarrow x \rightarrow z$):**
>
> Please note that the notation $y \rightarrow x \rightarrow z$ is a common notation for a Markov chain, equivalent to
> $p(z|x,y) = p(z|x)$.
> That is, given $x$ we have that $z$ is independent of $y$.
> We provided explanation on the notation in lines 44-45 of the original version, and following this comment we further explain it in the revision.
>
> Note that for $y$ being the true class (unknown at test time), $x$ being the sample, and $z$ being the processed sample, the Markov chain $y \rightarrow x \rightarrow z$ follows directly from the fact that the same processing ($z=\mathcal{A}(x)$) is applied on any $x$ regardless of its class. In other words, there is no assumption here that differs from practical low-level processing (e.g., denoising and encoding).
>
> The classifiers that we study indeed follow the direction: $x \rightarrow z \rightarrow \hat{y}$ if data processing $z=\mathcal{A}(x)$ is used, and $x \rightarrow \hat{y}$ if no data processing is used. Both flows are, of course, independent of the true unknown label $y$.
>
> To conclude, there is no mismatch here between the considered Markov chain and practical (deep) machine learning.
>
> &nbsp;
>
> **W2 (The GMM data model):**
>
> First, please note that in real-world classification, $x$ precedes $\hat{y}$ (the estimated label of $x$) and not $y$ (the true label of $x$ which is unknown at test time).
>
> Regarding Eq. (3), this is the standard way of defining a Gaussian Mixture Model (GMM) as the data distribution in theoretical works on classification. See, e.g., (Cao et al., 2021; Deng et al., 2022; Wang & Thrampoulidis, 2022; Kothapalli & Tirer, 2025), which we cited in our paper, and more.
>
> Recall that in GMMs, each component is a Gaussian with mean and covariance associated with the component index, i.e., conditioned on the component (Bishop, 2006).
> All the aforementioned theoretical works consider binary classification, where similar to us, the data distribution $p(x,y)$ is modeled by a GMM of order 2 where each class is associated with a Gaussian component and subsequently, the mean and variance are conditioned on the class.
>
> Of course, at test time, the classifier is given only the sample $x$, without the index of the associated GMM component (label). Moreover, unlike some prior works, we do not restrict the SNR (ratio between the squared norm of the mean and the variance) so our theory covers also low SNR regimes in which a value of $x$ can be associated with either of the labels with different probabilities.
>
> &nbsp;
>
> **W3 (Linearity of the theoretical low-level procedure):**
>
> Please note that currently there are no available mathematical tools to rigorously analyze practical deep learning models (such as DnCNN or deep encoders). Therefore, restricting the theoretical analysis to simplified models is inevitable.
>
> Indeed, any existing deep learning theory work (which attempts to analyze phenomena that are observed in deep learning) is based on simplified models.
> Just to name a couple of examples out of many: the analysis of "neural collapse" in [1] is essentially based on a matrix factorization model that ignores the data, and the analysis of overparameterization in [2] omits all the nonlinear activation functions from the model (so their model is linear in the input).
>
> [1] Zhu et al., "A Geometric Analysis of Neural Collapse with Unconstrained Features", NeurIPS 2021.
> [2] Arora et al., "On the Optimization of Deep Networks: Implicit Acceleration by Overparameterization", ICML 2018.
>
> Thus, it is reasonable to assume linear data processing for establishing rigorous theory and gain mathematically-backed insights.
> As can be seen in the proofs in the appendix, even for this linear data processing the analysis is challenging.
>
> Please note that the correctness of the theory is empirically validated (for the same linear data processing) in Section 3.3, and in Section 4, we empirically show that the trends established in the theoretical setting are observed also in practical deep learning experiments with sophisticated models --- hence demonstrating the usefulness and significance of our theory.
>
> In more detail, the linear low-level processing in the theoretical setup has proved to be sufficient for establishing multiple novel results, among them: (1) formally proving, for the first time, a gap between data processing inequality and practice: the existence of a learnable low-level processing that improves classification performance for any finite number of training samples; and (2) establishing insights and surprising behaviors (e.g., maximal relative gain for high SNR) that are empirically observed in practical deep learning setups.

---

> ### Author Response · Authors · 2025-11-24
>
> **W3 (cont.)**:
>
> Regarding (1), obtaining this fundamental result by a relatively simple and learnable dimension reduction procedure is not a limitation of our theory, as it implies that more sophisticated procedures can also possess this advantage.
>
> Regarding (2), the alignment between the behavior in practical deep learning setups, with two different DNN-based data processing techniques: denoising and encoding, demonstrates that the insights predicted under our theoretical setup generalize beyond the data processing considered in our theory.
>
> Nevertheless, we appreciate the reviewer's comment, as extending the analysis to nonlinear low-level processing is an interesting direction for future research, and we added it to the conclusion section of the revision.
>
> &nbsp;
>
> **W4 (Intuition and visualization the effect of $A$):**
>
> We thank the reviewer for this comment.
> Please note that below Theorem 3 in the original version we already mentioned the core reason for the discriminative gain from $A$:
> "We note that the semi-orthonormality of $A$ implies that it cannot increase the norm of any vector. The property $\| A \mu \| = \| \mu \|$ ensures that the separation quality remains unchanged (equal to Eq. 5) and is not reduced after the processing."
>
> Essentially, this implies that when applying $Ax$, the class-dependent (discriminative) component of $x$ (i.e., the projection of $x$ onto $\pm \mu$) is not attenuated. In contrast, the complementary component of $x$, which corresponds to within-class variability, is attenuated as the overall dimension is reduced and the semi-orthonormality of $A$ prevents amplification. Taken together, this is expected to facilitate classification, as we rigorously prove in our theorems.
>
> We add this explanation to the discussion below Theorem 3 in the revision, and also add discussion and a visualization that demonstrates it at the end of Appendix F in the revision.
>
> &nbsp;
>
> **W5 (Data processing vs augmentation):**
>
> There are key differences between pre-classification data processing (low-level tasks) and standard data augmentation:
>
> * The goal of data augmentation is to enlarge the training set by generating multiple samples from a single example. However, in low-level tasks, such as the processing considered in our theory, each sample $x_i$ results in a single processed version $z_i=Ax_i$.
>
> * The theoretical processing step $A:\mathbb{R}^d \to \mathbb{R}^k$ is a dimensionality reduction step, which modifies the sample dimension. As a result, the classifier that is trained on $\{z_i = A x_i \}$ cannot be applied to $x \in \mathbb{R}^d$ at test-time (due to mismatched dimension) but rather only to $Ax \in \mathbb{R}^k$. In other words, our processing step is performed during both training and testing.
> On the other hand, data augmentation is typically applied only during training and at test-time the trained model is applied on the original sample (without augmentations).
>
> Please note that our work does not claim to introduce new data processing techniques but rather aims to provide a theoretical explanation for the usefulness of such low-level tasks and to analyze their effect on a high-level classification task. To the best of our knowledge, despite the wide practical utilization of this pipeline, no such analysis or insights exist, which constitutes the novelty and significance of our work.
>
> &nbsp;
>
> **W6 (Regarding using noise injection in the experiments):**
>
> Please note that adding Gaussian noise to images in order to study the classification of corrupted images is a widely used setup that has been considered in many works.
> Just to name a few examples, here are works that use this setup and are already cited in our paper:
>
> (Liu et al., 2018): "When image denoising meets high-level vision tasks: A deep learning approach", IJCAI 2018.
> (Hendrycks \& Dietterich, 2019): "Benchmarking neural network robustness to common corruptions and perturbations", ICLR 2019.
> (Lu et al., 2025): "Ditch the denoiser: Emergence of noise robustness in self-supervised learning from data curriculum", NeurIPS 2025.
>
> Note also that in the image processing community, additive Gaussian noise is a fundamental and widely used model for real-world noise in natural images, and serves as the standard benchmark for evaluating denoising methods.
>
> Modifying the level of Gaussian noise in the experiments corresponds to modifying the standard deviation $\sigma$ in our theoretical model. Indeed, as the level of Gaussian noise increases, more details are masked by noise and samples from different classes become harder to distinguish. This behavior mirrors the effect of increasing $\sigma$ (reducing the SNR) in our theoretical setup. Lastly, please note that we also apply encoding as the data processing (in this case, it cannot be said that the setup is circular).

---

> > ### Author Response · Authors · 2025-11-24
> >
> > **W7 (Regarding using different processing methods in the experiments):**
> >
> > We thank the reviewer for this comment. Please note that the fact that we show that the theoretical trends also appear in practical models across different data-processing methods (*both* denoising and encoding) strengthens the significance of our theoretical contributions.
> >
> > Following the reviewer's comment, we add Subsection 4.3 and Appendix E in the revision, where we present results for the CIFAR-10 setup in which the data processing is based on encoding (similar to the Mini-ImageNet experiments). The trends predicted by our theory still hold.
> >
> > Regarding which data processing procedure to use, the new empirical results on CIFAR-10 show that encoding can yield higher efficiency than denoising for the classification task. Yet, we believe that this may not be the case for other high-level tasks, which may require preserving the spatial information in the image (e.g., object detection). We believe that our theoretical work may pave the way for future studies on questions such as: given a high-level task, what is the optimal low-level processing?
> > This discussion is added in the revision.
> >
> > &nbsp;
> >
> > **Minor 1 (Experiments on larger datasets):**
> >
> > We thank the reviewer for this suggestion.
> > We are aware of the significance of supporting theoretical findings with practical deep learning experiments and thus already presented results for 2 different practical low-level modules, practical DNN classifiers, and 2 benchmark datasets, which is beyond the empirical coverage of many theoretical works. Note that each point in Figures 2 and 3 (and many other figures in the appendix) is the result of training 3 models (a processing module and two classifiers) across multiple random seeds.
> > These empirical results support our theoretical findings.
> >
> > We used our compute resources to add another setup in the revision based on your comment W7, and we will do our best to add another, larger-scale setup in the final version.
> >
> > &nbsp;
> >
> > **Minor 2 (Source code for Figure 1):**
> >
> > Following the reviewer's request, we include source code for producing Figure 1 in the zip file of the supplementary material.
> >
> > &nbsp;
> >
> > We hope that our response addresses all your concerns, and we would be happy to further clarify open issues.

---

> ### Author Response · Authors · 2025-11-28
>
> Dear Reviewer, we sincerely appreciate your time and valuable comments. Due to the limited discussion time, we would be grateful for any additional questions or confirmation that our response has addressed your concerns. Please note that, along with our response, we have also updated the paper according to the comments, with the changes marked in blue.

---

### Author Response · Authors · 2025-12-02

Dear AC,

We thank you for handling our manuscript and sincerely thank the reviewers for their positive evaluation of our work, valuable suggestions, and insightful questions.

We are pleased that all the reviewers recognized the novelty and importance of our work, which includes significant contributions such as: presenting a comprehensive theoretical study showing that, despite concepts like the data processing inequality, even for a classifier that is tightly connected to the optimal Bayes classifier, there exists a pre-classification processing that improves its accuracy for any finite number of training samples; rigorously analyzing the different factors that affect the performance gain (including identifying a non-intuitive relation between the maximal gain and the SNR); and demonstrating that the trends observed in four practical deep learning setups are consistent with our theoretical results.

Specifically, **Reviewer p7cK** appreciated the motivation and thoroughness of the theoretical contributions, as well as the writing and the supporting experiments. **Reviewer yFZ5** stated that the problem is well-motivated and also that: "The result may open a new line of work on this topic."
**Reviewer PqbN** recognized the problem as "interesting and important" and appreciated the solidness and thoroughness of the theoretical contributions.

We provided a point-to-point response to the comments of each of the reviewers and **addressed all their concerns**.

Along with our response, we also updated the paper according to the comments, with the changes marked in blue, which include:

* Following a comment of Reviewer p7cK, we added Subsection 4.3 and Appendix E, where we present a new set of experiments for the CIFAR-10 setup in which the data processing is based on encoding (similar to the Mini-ImageNet experiments).
The trends predicted by our theory hold also in this setup.
Overall, despite the theoretical focus of our paper, it includes **four different practical setups**: CIFAR-10 with MSE-based denoising, CIFAR-10 with SURE-based denoising, CIFAR-10 with encoding, and Mini-ImageNet with encoding. Note also that each point in an empirical figure is the result of training three different models over multiple random seeds.

* Following a comment of Reviewer PqbN, we added Appendix B, where we present and prove Theorem 10, which analyzes the effect of the different factors on the direct difference $\Delta=\hat{p}_x-\hat{p}_z$ for large $N$.
The insights are aligned with those that are gained from Theorem 7  (which considers the relative difference $\eta=(\hat{p}_x - \hat{p}_z)/\hat{p}_x \times 100$).

* We polished and extended the discussions below several theorems, aiming to provide more intuition than in the original version of the paper.
We also improved the readability of Section 3.2, as suggested by Reviewer PqbN.


Lastly, we would like to reiterate the justification for the data processing considered in our theoretical setup.
(Note that while this choice allowed us to establish rigorous theory, it still required a technically demanding analysis, as can be seen in our proofs).


Note that the linear low-level processing in the theoretical setup has proved to be sufficient for establishing multiple novel results, among them: (1) formally proving, for the first time, a gap between data processing inequality and practice: the existence of a low-level processing that improves classification performance for any finite number of training samples; and (2) establishing insights and surprising behaviors (e.g., maximal relative gain for high SNR) that are empirically observed in practical deep learning setups.

Regarding (1), obtaining this fundamental result by a relatively simple and learnable dimension reduction procedure implies potential gain from more sophisticated procedures. Thus, this is in fact **a strength of our theory**.

Regarding (2), the alignment between the behavior observed in practical deep learning setups, with two different data processing techniques: denoising and encoding, demonstrates that **the insights and predictions of our theory generalize well beyond the theoretical setup**.

We hope that the novelty of our work and the significance of our contributions (as acknowledged by the reviewers), together with our detailed response, which addresses all the points raised by the reviewers, will lead to accepting our paper.

---

### Meta-Review · Area_Chair_KdSG · 2026-01-06

**Summary:**

This paper asks when “low-level” preprocessing can help classification, despite the data processing inequality. It provides a clean theoretical result showing that, for finite samples, there exists a learnable preprocessing (dimension reduction) that can improve a strong classifier’s accuracy, and backs this up with simulations plus practical experiments using denoising/encoding.

Across the submitted reviews, the paper is generally seen as well-motivated and technically solid, with the main skepticism coming from how far the simplified theory and experimental setups can be taken as evidence for real-world deep learning practice.

**Reviewer Concerns:**

Concerns addressed by the rebuttal:
1) Clarity/positioning: the authors clarified the modeling assumptions and better explained why the setup is appropriate for the question being asked.
2) Evidence: the rebuttal adds supporting experiments and additional intuition to connect the theoretical phenomenon to practical pipelines.
3) Presentation: several requests for clearer structure and explanation were handled reasonably well.

Concerns still outstanding:
1) Theory–practice gap: the strengthened experiments help, but the empirical scope is still limited relative to the “reflects practice” framing, so some skepticism about broad generality remains reasonable.
2) Experimental representativeness: even with added results, parts of the evaluation can still be read as somewhat stylized, and it is not fully settled how often the claimed effect is a dominant explanation in modern systems.

**Reviewer Scores:**

1) Reviewer p7cK (score 6): likely unchanged at 6 (possibly a slight increase) since the main requests were addressed and added evidence reduces uncertainty.
2) Reviewer PqbN (score 6): likely increase to 7 because the rebuttal targets the concrete asks (clearer structure, more intuition, and added supporting material).
3) Reviewer yFZ5 (score 4): likely increase to 5; the response improves motivation and evidence, but the broader generalization concern likely remains.

---

### Decision · Program_Chairs · 2026-01-26

Accept (Poster)